# Multi-lens ultrasound arrays enable large scale three-dimensional micro-vascularization characterization over whole organs

Nabil Haidour[1], Hugues Favre [1], Philippe Mateo[1], Juliette Reydet[1], Alain Bizé[2], Lucien Sambin[2], Jianping Dai[2], Paul-Matthieu Chiaroni[2,3], Bijan Ghaleh [2], Mathieu Pernot [1], Mickael Tanter [1] & Clement Papadacci [1] ✉

Mapping microcirculation at the whole-organ scale in 3D is crucial for understanding vascular pathologies and improving diagnostics. Although 3D ultrasound localization microscopy (ULM) enables microscopic resolution by localizing intravenously injected microbubbles in small animal models, visualizing entire organs in large animals or humans remains challenging due to limited field of view, low sensitivity, and probe technological complexity. Here, we demonstrate how a multi-lens array method overcomes these limitations. Combined with 3D ULM, it maps and quantifies large vascular volumes (up to $120 \times 100 \times 82$ mm³) at high spatial resolution (125–200 μm) with a volumetric acquisition rate of 312 Hz, using low-cost technology. This approach enables deeper insights into hemodynamics from large vessels to pre-capillary arterioles, by providing vast and rich datasets of whole-organ vascularization. It could also facilitate diagnosis of microcirculation disorders and monitoring of small-vessel disease treatments by addressing key limitations of current imaging modalities.

Microcirculation plays a vital role in sustaining tissue health by regulating blood flow and facilitating the exchange of gases, nutrients, and waste products. In key organs such as the heart, kidneys, and liver, proper microcirculation is essential for their optimal function. Any structural or functional alteration in these networks can lead to severe clinical outcomes, including heart failure, renal failure, and various chronic diseases. By evaluating the integrity of the circulatory system from the largest arteries down to small arterioles, clinicians could identify and treat dysfunctions early, potentially preventing irreversible damage[1]. Nevertheless, the amount of information to deal with in order to reach such a goal is massive. Building a $100 \times 100 \times 100$ mm³ raw data volume with a 100 μm voxel size at a classical 300 Hz frame

rate for hemodynamic estimations leads to a typical acquisition movie of 300 billion of voxels per second.

CT angiography (CTA) is the clinical gold standard to noninvasively visualize blood vessels at the whole-organ scale by injecting ionizing contrast agents, providing detailed anatomical insights into vascular structures in clinic. The recent advancement of photon-counting computed tomography (PCCT) has further improved CTA by offering enhanced spatial resolution and better tissue contrast[2]. However, CTA, even with PCCT, has limited capacity to assess flow dynamics due to its static nature and temporal constraints. Magnetic Resonance Imaging (MRI), particularly with 4D flow imaging, enables detailed whole-organ or whole-body visualization of flow dynamics by capturing blood flow patterns over time in three dimensions[3]. This

[1]Institute Physics for Medicine Paris – INSERM, ESPCI Paris-PSL, CNRS, Paris, France. [2]INSERM U955-IMRB, University Paris-Est Creteil, National Veterinary School of Alfort, Paris, France. [3]Interventional Cardiology Department, Henri Mondor Hospital, Creteil, France. ✉e-mail: clement.papadacci@espci.fr

capability makes it highly effective for hemodynamic assessments and functional imaging. However, MRI faces limitations, including a spatial resolution limited to a few tenths of a millimetre, high costs, and limited accessibility, which can hinder its widespread clinical use. On the other hand, ultrasound is being routinely used to assess blood flow and its dynamics in the clinic, but it usually suffers from low resolution and sensitivity. Nevertheless, ultrasound imaging is an active research field, and major technical advances achieved over the past decades open perspectives for developing high-performance imaging modes that could rapidly be translated to clinics, given the relatively low cost and portability of ultrasound technologies[4–6].

In the past decade, ultrasound localization microscopy (ULM) was developed to map microcirculation at the microscopic scale in pre-clinical organ models[7,8]. This technique has progressed to the point of demonstrating 2D proof-of-concept in patients to image organs in super-resolution, such as the brain[9], heart[10], and kidney[11]. However, 2D imaging has inherent limitations, including its inability to provide a complete view of the organ under study. Improper positioning of the imaging plane can lead to increased risks of diagnostic mis-interpretations, highlighting the importance of developing three-dimensional approaches to overcome these constraints[12]. Recently, three-dimensional (3D) ULM has been applied to map the whole organ microcirculation in small animals[13–18], providing valuable insights into organ function and enabling a more comprehensive understanding of their physiological processes. However, physiology, anatomy, and size of small animal organs are hardly comparable to human organs and translation of 3D ULM to large animal models has been limited by current technology. State-of-the-art matrix array probes are associated with high complexity, expensive, and cumbersomeness hardware, primarily due to the necessity to connect each individual element of the probe to a receive channel. Thousands of elements and receive channels are thus required to cover the entire volume of a human organ, which limits their current applications.

To alleviate these issues, we have recently introduced the concept of multi-lens arrays both in simulations[19] and in in vitro experiments with a simplified prototype[20]. It consists in using a large aperture probe filled with large elements combined to individual acoustic diverging lens. Large elements provide increased sensitivity while covering a wide probe surface. Diverging lenses are used to restore a low directivity, compensating for the reduced angular aperture typically associated with large elements. This design improves the focusing quality and enhances image resolution. Comprising of a few hundred elements, the probe is compatible with ultrasonic scanners commonly available in clinical settings. When combined to 3D ULM methods, these arrays can assess the blood flows across several spatial scales, down to the micron scale across the whole organ. In this manuscript, we introduce a first large multi-lens array probe and a full post-processing framework to enable the imaging and quantification of microcirculation of living organs in unprecedented large volumes. Applied to map organs in a large animal swine model, we demonstrate ULM imaging over massive volumes (up to $120 \times 100 \times 82$ mm$^3$), including ex-vivo heart, in-vivo kidney and liver with an effective temporal resolution depending on the accumulation time required for ULM reconstruction. It provides vast and rich mdatasets of whole organ vascularization and hemodynamics that could prove invaluable for advancing AI-driven medical and comprehensive medicine.

## Results

In this article, we presented the results of a multi-lens matrix array probe with a large aperture of $104 \times 82$ mm$^2$ and a reduced number of elements (252 elements), as illustrated in Fig. 1a. A schematic diagram of a single large element coupled to a compound diverging lens is shown in Fig. 1b, highlighting the principle of this composite material. It consists of a plano-convex lens combined to plano-concave lens which according to Snell's law enables to create a strong diverging lens

(part Principles and definitions in the Supplementary Methods section).

The first red line inside the plano-convex lens represents the beam originating from the transducer. The second red line shows the beam refracting due to the difference in sound velocity between the plano-convex lens ($c_1 = 2570$ m s$^{-1}$), and the flat lens layer ($c_2 = 1015$ m s$^{-1}$). The third red line illustrates additional refraction as the beam passes from the plano-concave lens into the biological soft tissue ($c_3 = 1540$ m s$^{-1}$). The Fig. 1c further illustrates the refraction principles of the two diverging lenses in the compound design, showing how both lenses contribute to beam divergence.

Figure 1d compares the simulated acoustic field emitted by a large element with a compound lens to a large element with no lens and a small element used in a conventional dense matrix. The pressure fields produced by the large element with the compound lens (Fig. 1d. i) and the small element (Fig. 1d. iii) show low directivity and a wider field of view. Large element with and without the compound lens show high energy transmission (Fig. 1d. ii). The large element with the compound lens is the only approach that combines low directivity and high transmitted energy. The angular directivity at −6 dB, measured at a depth of 35 mm in the pressure field, is presented in the Fig. 1d. iv. The angular directivity for the large element with compound lens, the large element and the small element were found to be 33.17°, 10.27°, and 51.13°, respectively.

The sensitivity response of the multi-lens array probe was compared to a dense matrix array and a sparse array with the same number of elements, as shown in Fig. 1e. Sensitivity was defined as normalized intensities received from a transmitter placed in the medium of interest. Qualitative observations in Fig. 1e.i illustrate the sensitivity response of the multi-lens array probe, revealing high sensitivity across a large simulated volume ($105 \times 100 \times 100$ mm$^3$). In contrast, the dense matrix array demonstrates a confined sensitivity below the aperture (Fig. 1e. ii), while the sparse array shows low sensitivity, especially in the far field (Fig. 1e. iii). These qualitative observations were quantified and presented in Fig. 1e. iv, which shows that the multi-lens array achieves significantly higher sensitivity for an intensity above −18 dB within the simulated volume and covering up to 1050 cm$^3$. In comparison, the dense matrix and sparse array showed lower sensitivity, covering 443 cm$^3$ and 580 cm$^3$ of the total volume, respectively.

The simulated points spread functions (PSF) of the multi-lens array probe was performed and quantified, revealing a grating lobe level of approximately −25.50 dB, −24.57 dB, and −25.36 dB, and lateral resolution was found to be 1.15 mm, 1.17 mm, and 1.26 mm at −6 dB for imaging depths of 30 mm, 45 mm, and 60 mm, respectively. These results were obtained using synthetic aperture imaging with 252 transmit sources. Additionally, we evaluated image quality under ultrafast imaging conditions using a reduced number of transmit events (16 sources), to reflect the practical configuration used during acquisitions. The corresponding results are presented in the Supplementary Fig. 2.

The tip of a needle was imaged in vitro to study the PSF of the multi-lens array probe, as illustrated in Fig. 2b. Grating lobes with an intensity of -22.40 dB and a lateral resolution of 1.06 mm at −6 dB were quantified for an ultrafast imaging sequence using 16 sources in transmit (Fig. 2c). The measured -6 dB width of the PSF was consistent with the theoretical resolution calculated based on the ultrasonic wavelength, focal distance, and array aperture.

3D ULM was performed on a large flow phantom containing a tube with injected microbubbles (MBs). The results presented in Fig. 2d, showed the reconstructed MB density volume of the tube, revealing a large imaging volume of $100 \times 93 \times 30$ mm$^3$. Grating lobes were filtered out by the tracking algorithm. Absolute flow velocity was measured in the tube (Fig. 2e), revealing a Poiseuille flow profile (Fig. 2f). A statistical Student's $t$-test showed significant differences between neighbouring voxels along the velocity profile ($P$ values from left to right are $P_1 = 0.0004$, $P_2 = 3.23 \times 10^{-11}$, $P_3 = 0.0022$, $P_4 = 4.22 \times 10^{-05}$, $P_5 = 0.0059$, $P_6 = 8.03 \times 10^{-05}$), indicating a resolution of 75 μm.

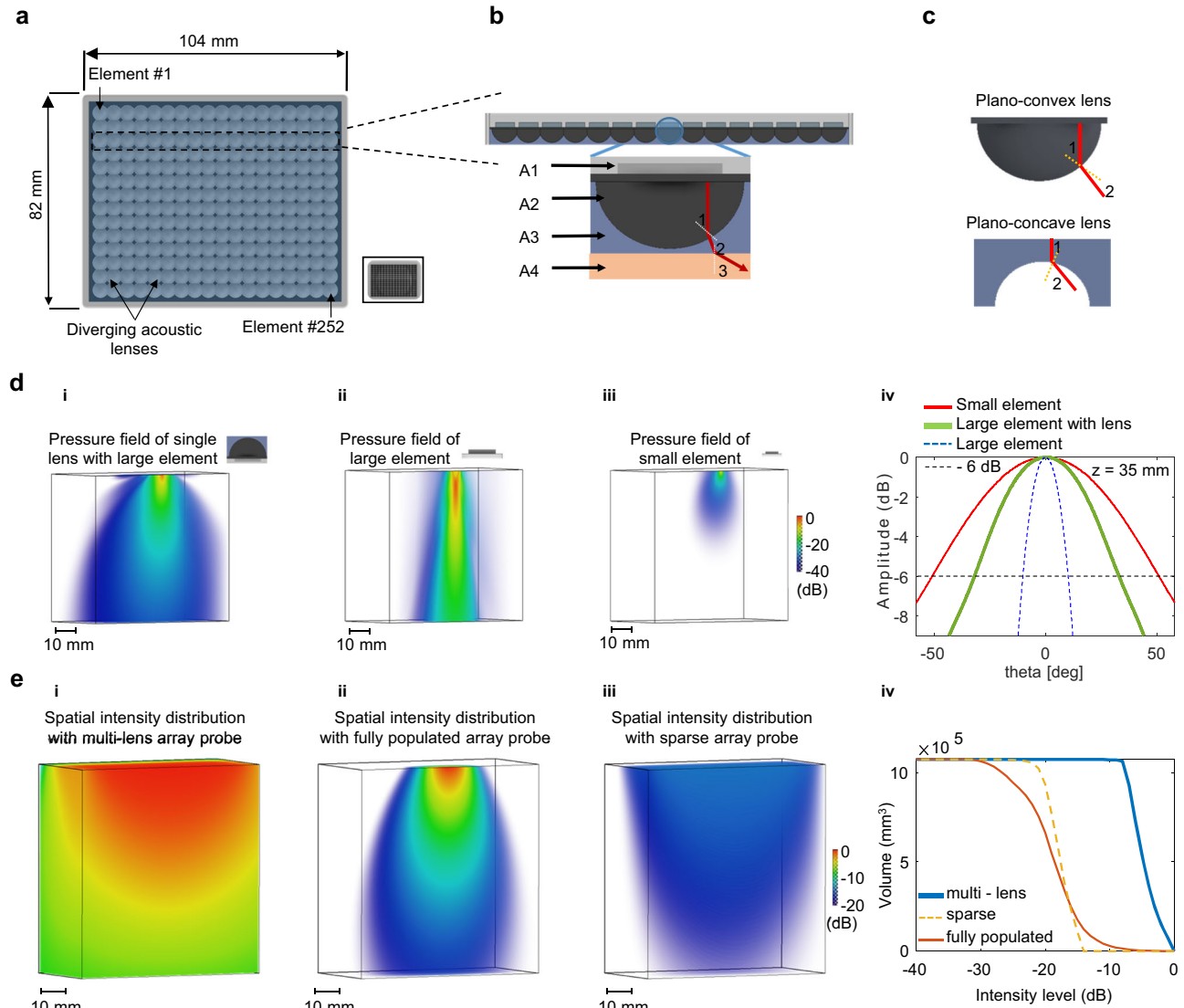

**Fig. 1 | Definition, simulation study of the multi-lens array probe and comparison to conventional approaches. a** Schematic representation of the multi-lens array probe with a large aperture (-104 × 82 mm²) composed of 252 elements, compared to a conventional fully populated probe with the same number of elements (aperture-14 × 11 mm²) depicted within the black box. **b** Schematic diagram of the compound diverging lens placed in front of a large element; red lines indicating the beam path the compound lens and the dashed white line represents the normal; **A1** represents the large transducer (3λ, where λ ≈ 1.5 mm), **A2** is the plano-convex lens, **A3** is the plano-concave lens, and **A4** is the medium of interest. **c** A schematic diagram of the plano-convex diverging lens is shown in black, with red lines indicating the beam path within the lens. The beam path through the blue plano-concave lens is also illustrated. (**a**–**c** created using Autodesk Inventor Professional 2025.0.1). **d** Schematic and 3D maps of maximum transmitted acoustic pressure field, illustrating directivity performances over a volume of 60 × 60 × 60 mm³: **d.i** the compound lens material coupled to a single large element of an (3λ × 3λ) compared to **d.ii** a single large element without lens (3λ × 3λ), and **d.iii** a single small element (λ/2 × λ/2). **d.iv** Graphs show the angular directivity [degrees] as a function of intensity threshold level, measured at a depth (z) of 35 mm from the maximum pressure field shown in (**d**). for three configurations: single lens with large element (solid green), large element without lens (blue dashed), and small element (red curve). **e** Spatial distribution of the intensity field received by the multi-lens probe (**e.i**) over a large volume (-105 × 100 × 100 mm³) compared to conventional probes, fully populated (**e.ii**) and sparse matrix arrays (**e.iii**). (**e.iv**) Graph showing the intensity threshold level as a function of the covering volume for the three probes: multi-lens array (solid blue curve), fully populated array (red curve), and sparse array (orange dashed curve).

The inner tube diameter was measured using skeletonization: $916 \pm 38 \, \mu m$ which was in a good agreement with the manufacturer tube diameter (870 μm). The maximum absolute velocities at the centre of the tube were measured for the four flow rates: $62 \pm 5 \, mm \, s^{-1}$, $86 \pm 4 \, mm \, s^{-1}$, $122 \pm 5 \, mm \, s^{-1}$, and $177 \pm 4 \, mm \, s^{-1}$. Flow rates were quantified: $73 \pm 6 \, mL \, h^{-1}$, $101 \pm 3 \, mL \, h^{-1}$, $143 \pm 5 \, mL \, h^{-1}$, and $209 \pm 5 \, mL \, h^{-1}$, respectively. The measured flow rates were in a good agreement with the imposed flow rates ($R^2 = 0.98$), Fig. 2g.

For more complex spatial and flow patterns, the twisted-tube configuration was imaged. The 3D microbubble (MB) density map (Supplementary Fig. 3c). reveals a detailed reconstruction of the tubes over a large volume, clearly illustrating the separation between the intertwined tube segments (Supplementary Fig. 3e and Supplementary Fig. 3f). Flow direction mapping demonstrated both upward and downward flow within the structure (Supplementary Fig. 3d). Quantification of flow velocity revealed a Poiseuille profile, with high spatial resolution (Supplementary Fig. 3g).

The multi-lens array probe provided the 3D visualization of the whole coronary microcirculation network of the isolated swine heart (Fig. 3a) in a large imaging volume of 120 × 100 × 82 mm³ as illustrated by the 3D MB density volume in Fig. 3b.

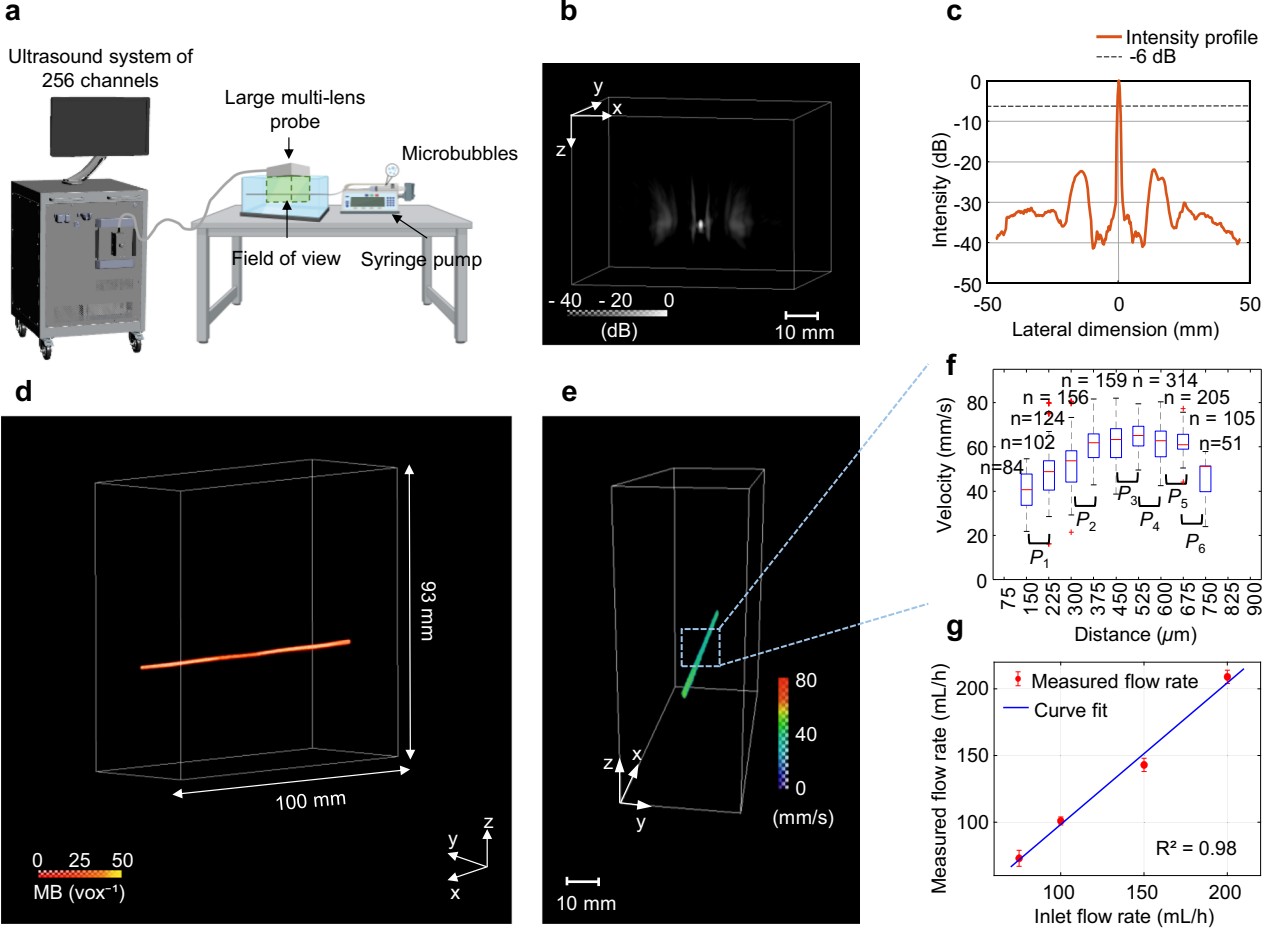

**Fig. 2 | In vitro validation of the multi-lens array. a** Schematic representation of the experimental flow phantom setup. Microbubbles (MB) were injected into an 870 μm inner diameter tube and 3D Ultrasound Localization Microscopy (ULM) was performed using the multi-lens array. Created in BioRender. Papadacci, C. (2025) https://BioRender.com/e2rxni8 and Autodesk Inventor Professional 2025.0.1 **b** Point spread function in 3D (PSF) of the multi-lens. **c** Quantification of the grating lobes and the lateral resolution at -6dB (horizontal dashed line); the red graph represents the maximum intensity projection profile in lateral dimension of the PSF using 16 sources in transmit to perform ultrafast ultrasound imaging. **d** 3D MB density of the tube (imaging volume ~100 × 93 × 30 mm³). **e** 3D absolute flow map of the MB velocity distribution through the tube. **f** Cross section of the tube velocities in the blue dashed box in (**e**) highlights a Poiseuille profile. Significant differences between the neighbouring voxels (~75 μm) within the velocity profile were

demonstrated using an unpaired two-sided Student's $t$-test. $P$ values from left to right are $P_1 = 0.0004$, $P_2 = 3.23 \times 10^{-11}$, $P_3 = 0.0022$, $P_4 = 4.22 \times 10^{-05}$, $P_5 = 0.0059$, $P_6 = 8.03 \times 10^{-05}$. The red horizontal line indicates the median; boxes denote the 25th and 75th percentiles; Whiskers extend to the most extreme non-outlier data points. n samples corresponds to the number of MB passing through each voxel/vox (84, 102, 124, 156, 159, 314, 205, 105 and 51 from left to right). other points, outliers. Measurements are technical replicates. **g** Flow rate measurements as function of the inlet flow rate set by the syringe pump, showing a linear relationship as shown by blue curve fit ($y = 1.06x - 7.83$; $y$: measured flow rate and $x$: inlet flow rate, coefficient of determination ($R^2$) = 0.98); Red dots represent the mean values of the data points ($n = 10$ samples for each measurement) and the horizontal red lines indicate the error bars as ± SD: 73 ± 6, 101 ± 3, 143 ± 5 and 209 ± 5 (from left to right); Measurements are technical replicates.

Coronary network can be visualized in different directions as illustrated by the YZ and XY projections of the 3D density (Fig. 3c). The Fig. 3d shows the maximum intensity projection (XZ view) of the Power Doppler image of the ex vivo porcine heart. The high temporal resolution achieved by the 3D ultrafast ultrasound imaging allowed for the estimation of absolute flow velocities via tracked MBs, as shown in Fig. 3e, with velocity ranges from 10 mm s⁻¹ in small vessels to over 300 mm s⁻¹ in the largest vessels. A Poiseuille flow profile was observed by analysing the velocity profile in a large vessel (Vessel 1 in the magnified view in Fig. 3f). A statistical Student's $t$-test revealed significant differences of velocity values along the profile between neighbouring voxels (voxel size = 150 μm) containing tracked MB velocities as shown in Fig. 3g. A magnified view of 3D density in Fig. 3h highlights the coronary trees, with the skeletonization in Fig. 3i to estimate vessel radii, revealing a range from approximately 75 μm for the smallest vessels to 600 μm for the largest vessels. Figure 3j shows the flow direction in the coronary network, where red (downward)

can be associated with the arterial flow system and blue (upward) the venous flow system. Fitting to a power law, the flow rate radius relationship provides an exponent of 2.3 ($R^2$ = 0.82), as shown in Fig. 3k, which is consistent with Murray's law[21]. The number of tracked MBs within the coronary network is shown as function of their estimated velocity in Fig. 3l, with a maximum velocity of 300 mm s⁻¹. The Fourier Shell Correlation (FSC) analysis was performed to estimate spatial resolution for the 3D ULM coronary mapping, achieving a resolution of approximately 125 μm at the intersection of the FSC and ½ bit curves, as shown in Fig. 3m.

The multi-lens probe was positioned on the right lateral side of the swine, in front of the kidney, and held using an articulated arm. Ultrasound acquisitions were synchronized with the Electrocardiogram (ECG) R-peak wave, during periods of minimal respiratory motion, as illustrated in Fig. 4a. The Fig. 4b shows a B-mode image of the porcine kidney obtained with a portable ultrasound device. 3D ULM of the kidney was performed over a large volume of 60 × 80 ×

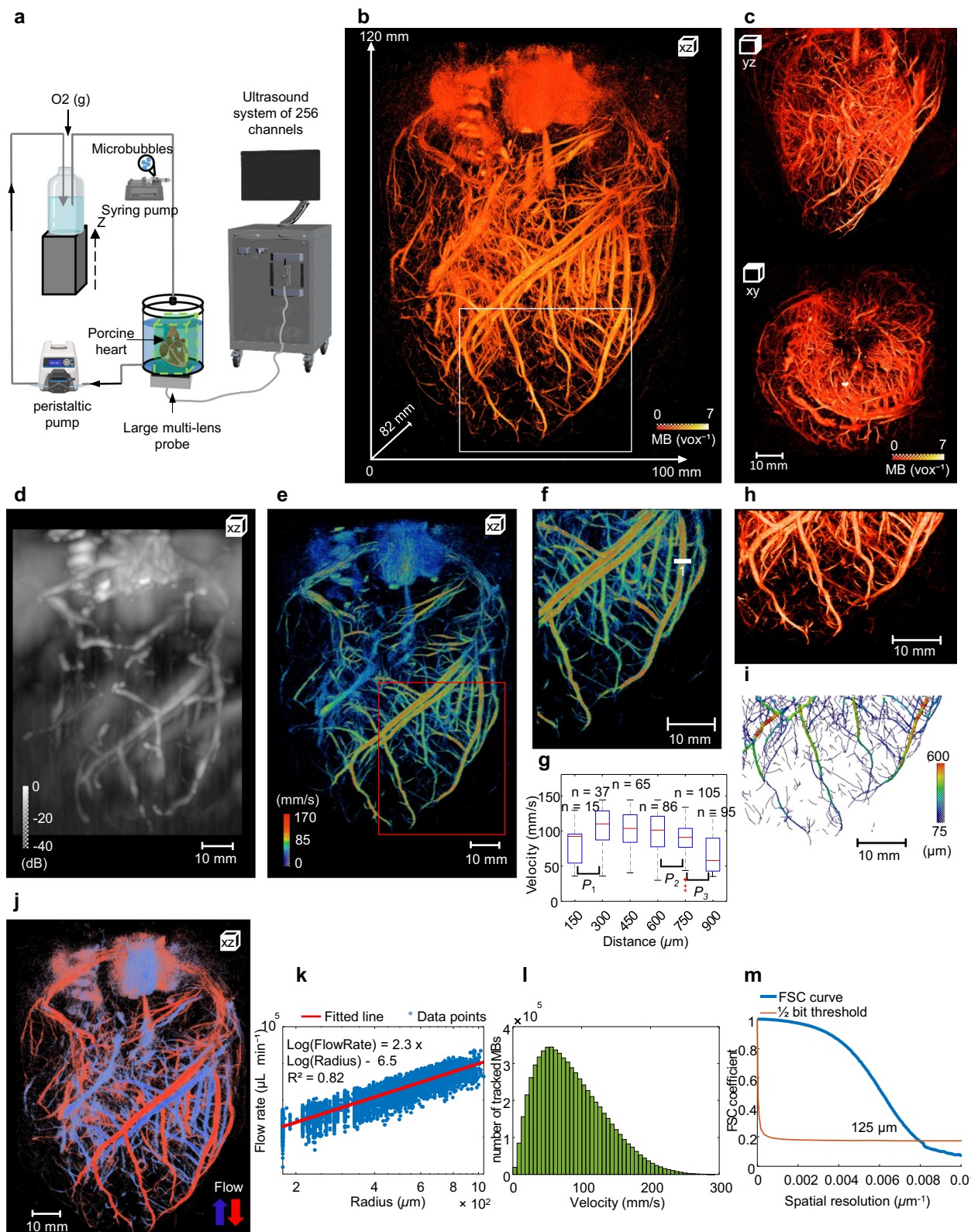

40 mm³, and the results is presented as 3D MB density maps in Fig. 4c and a YZ projection view in Fig. 4d. The 3D absolute flow velocity shown in Fig. 4e, was evaluated, reaching a maximum velocity of approximately 280 mm s⁻¹ in the large vessels. 3D directional flow velocities reveal the arterial flow system in red and the venous flow system in blue (Fig. 4f. i and Fig. 4f. ii for XZ and YZ projections view, respectively). A maximum intensity projection (YZ view) of the Power

Doppler image of the in vivo porcine kidney is shown in Fig. 4g. A skeletonization algorithm was applied on the volume showing in white box in Fig. 4c, to estimate vessel radii, revealing a range of 70 μm to 400 μm as illustrated in Fig. 4h. A spatial resolution of 147 μm was obtained by applying the FSC, as shown in Fig. 4i. Fitting to a power law, the flow-rate radius relationship provides an exponent of 2.13 ($R^2 = 0.70$), which is in good agreement with Murray's law, as shown in

**Fig. 3 | 3D Coronary mapping of an isolated porcine heart. a** Experimental setup diagram: 3D Ultrasound localization Microscopy (ULM) imaging using the multi-lens array probe driven by an ultrasound system of 256 channels to image the heart with a retrograde and microbubble perfusion. Created in BioRender. Papadacci, C. (2025) https://BioRender.com/hyvll4l and Autodesk Inventor Professional 2025.0.1. **b** 3D Microbubbles (MB) density map of the coronary network of the porcine heart illustrating the large imaging volume (-120 × 100 × 82 mm³). **c** Views of the MB density map (yz and xy views) of the coronary porcine heart. **d** Maximum Intensity Projection (MIP) of Power Doppler data acquired from the isolated porcine heart using the multi-lens probe. **e** 3D flows velocity map showing the MB velocity through the coronary networks. **f** Magnified view of the red box in (**e**) volume (53 × 41 × 38 mm³). **g** Analysis of the Poiseuille flow profile in a large vessel 1 in (**f**). Significant differences between the neighbouring voxels (-150 μm) within the

velocity profile were demonstrated using an unpaired two-sided Student's *t*-test. P values from left to right are P1 = 0.0116, P2 = 0.0088, P3 = 7.25 × 10−08. The red horizontal line indicates the median; boxes denote the 25th and 75th percentiles; Whiskers extend to the extreme non-outlier data points. *n* samples correspond to the number of MB passing through each voxel (15, 37, 65, 86, 105 and 95 from left to right). other points, outliers. Measurements are technical replicates. **h** Magnified view of the region in the white box in (**b**) (30 × 43 × 45 mm volume). **i** Quantification of the vessel's radii by applying skeletonization algorithm on the density in (**f**). **j** 3D flow map showing arterial (red) and venous (blue) flow; arrows indicate upward and downward directions. **k** Flow rate as a function of radius (blue stars) provides an exponent of 2.3 which is in good agreement with Murray's law. **l** number of MB distribution as a function of MBs velocity. **m** Fourier shell correlation (FSC; solid blue curve) to estimate the resolution of 3D ULM coronary mapping (-125 μm).

Fig. 4j. The number of tracked Mbs within the kidney vasculature is shown as function of their estimated velocity in Fig. 4k. The Fig. 4.l shows a Poiseuille flow profile by analysing the velocity profile in porcine kidney within the vessel (i) illustrated in the Fig. 4e. A statistical Student's *t*-test revealed significant differences of velocity values along the profile between neighbouring voxels (voxel size = 128 μm) containing tracked MB velocities. Dynamic ULM was performed (Fig. 4m), which highlights higher velocities during systolic phase compared to the diastolic phase. This observation is confirmed quantitatively for the velocity's measurements and resistivity indices $RI_1$ and $RI_2$ within the artery vessels 1 and 2 (V1 and V2), as shown in Fig. 4n.

Figure 5a illustrates the implemented 3D rigid inter-block motion correction pipeline. For each acquisition block at a low respiratory phase, a Power Doppler volume reconstruction was generated. An intensity-based registration between these volumetric Power Doppler blocks was then performed to compute a transformation, which was subsequently applied to the corresponding tracked microbubble points for each block.

As a result, Fig. 5b presents a qualitative comparison of the cortical density map of the kidney before (Fig. 5b. 1) and after (Fig.b.2) motion correction. The Fig. 5c represents a magnified view of the region highlighted by the white box in Fig. 5b to appreciate the effect of the correction. Additionally, a quantitative comparison of vessel density measurements (vessels i and ii) shows an increase in density after motion correction, indicating improved vessel visualization and alignment.

The multi-lens probe was positioned above the swine, placed in front of the liver, and held using an articulated arm. Similarly to the kidney experiment, ultrasound acquisitions were synchronized with the ECG R-peak trigger and during periods of minimal respiratory motion as illustrated in Fig. 6a. Real-time B-mode imaging using a portable ultrasound device allowed 2D visualization of the liver (Fig. 6b). The multi-lens array probe enabled wide-field imaging, enabling high-depth visualization of the liver's vasculature and capturing a substantial imaging volume of 65 × 100 × 82 mm³, as demonstrated by the 3D density maps in Fig. 6c. The Fig. 6d shows the Power Doppler volume corresponding to the in vivo acquisition of the porcine liver. The Fig. 6e illustrates the 3D absolute flow velocity within the liver, reaching a maximum velocity value of 300 mm s⁻¹. The direction of MBs flow through the liver's vasculature is shown in Fig. 6f, with upward flow in red and downward flow in blue. Dynamic velocity measurement was performed for arterial (pulsatile), vein (pulsatile/retrograde) and portal (no pulsatility) vein vessels, representing in V1, V2 and V3 respectively in the Fig. 6g, which highlighted higher velocities during systolic phase compared to the diastolic phase in case of an arterial vessel in Fig. 6g. i. The Fig. 6g. ii, shows the velocity profile within a hepatic vein in the liver, reflecting the characteristic pulsatile flow of the hepatic venous system. The Fig. 6g. iii suggests that the flow velocity is uniform in the portal vein. The Fig. 6h presents a Poiseuille flow profile by analysing the velocity distribution within the vessel (i) in porcine liver flow velocity illustrated in the Fig. 6e. A statistical

Student's *t*-test revealed significant differences of velocity values along the profile between neighbouring voxels (voxel size = 159 μm) containing tracked MB velocities. Fitting the flow-rate-to-radius relationship to a power law yielded an exponent of 2.36 (R² = 0.61), which is consistent with Murray's law (Fig. 6i). Spatial resolution of 200 μm was measured using Fourier Shell Correlation (FSC), as demonstrated in Fig. 6j.

## Discussion

In this study, an ultrasonic multi-lens array approach was developed to map and quantify micro-scale vascular network and flow dynamics. This method overcomes the limitations of conventional ultrasound imaging by enabling high-resolution volumetric imaging of large organs.

This innovative ultrasonic imaging method was initially conceived and validated through numerical simulations, followed by in vitro experiments on large phantoms. It was then applied to an explanted swine heart and in vivo swine kidney and liver, demonstrating whole-organ volumetric imaging capabilities. It enabled a complete visualization of the vascular network with a spatial resolution of a few tenths of micrometre (down to 125 μm) with a very high temporal resolution (312 Hz frame rate), providing anatomical details with high precision. Fine vascular quantification was achieved by measuring the vessel radius at every voxel, ensuring structural mapping across the entire vascular tree. Beyond structural imaging, the method offered advanced flow dynamic assessment across all vascular scales, capturing both macro and microcirculatory hemodynamic. This detailed analysis allowed for the differentiation of arteries and veins based on axial flow direction and enabled the measurement of absolute flow velocity along with its Poiseuille profile, which is a key indicator of laminar flow behaviour. These precise flow measurements pave the way for advanced physiological evaluations, including velocity distribution mapping, flow rate calculation, pulsatility index, and the assessment of Murray's law, a principle that links vessel diameter to optimal flow efficiency. This level of anatomical and functional measurements details beyond those of the conventional imaging techniques, allowing assessment of vascular structures and flow.

The combination of whole-organ volumetric imaging with high-resolution vascular quantification effectively addresses key limitations of existing modalities, such as ultrasound Doppler imaging, CT angiography, and 4D flow MRI. The high spatial resolution achieved with this technique is due to the implementation of 3D Ultrasound Localization Microscopy (ULM) and 3D dynamic ULM[22,23], going beyond the diffraction limit that traditionally constrains standard ultrasound resolution. The method demonstrated a spatial resolution approximately ten times finer than conventional ultrasound, enabling the visualization of small vascular structures. Additionally, the limited field of view commonly associated with classical matrix arrays was resolved through the use of a low-frequency, and large-aperture matrix array. This innovative approach allowed the reconstruction of large volumetric datasets, both laterally and in depth, enabling the

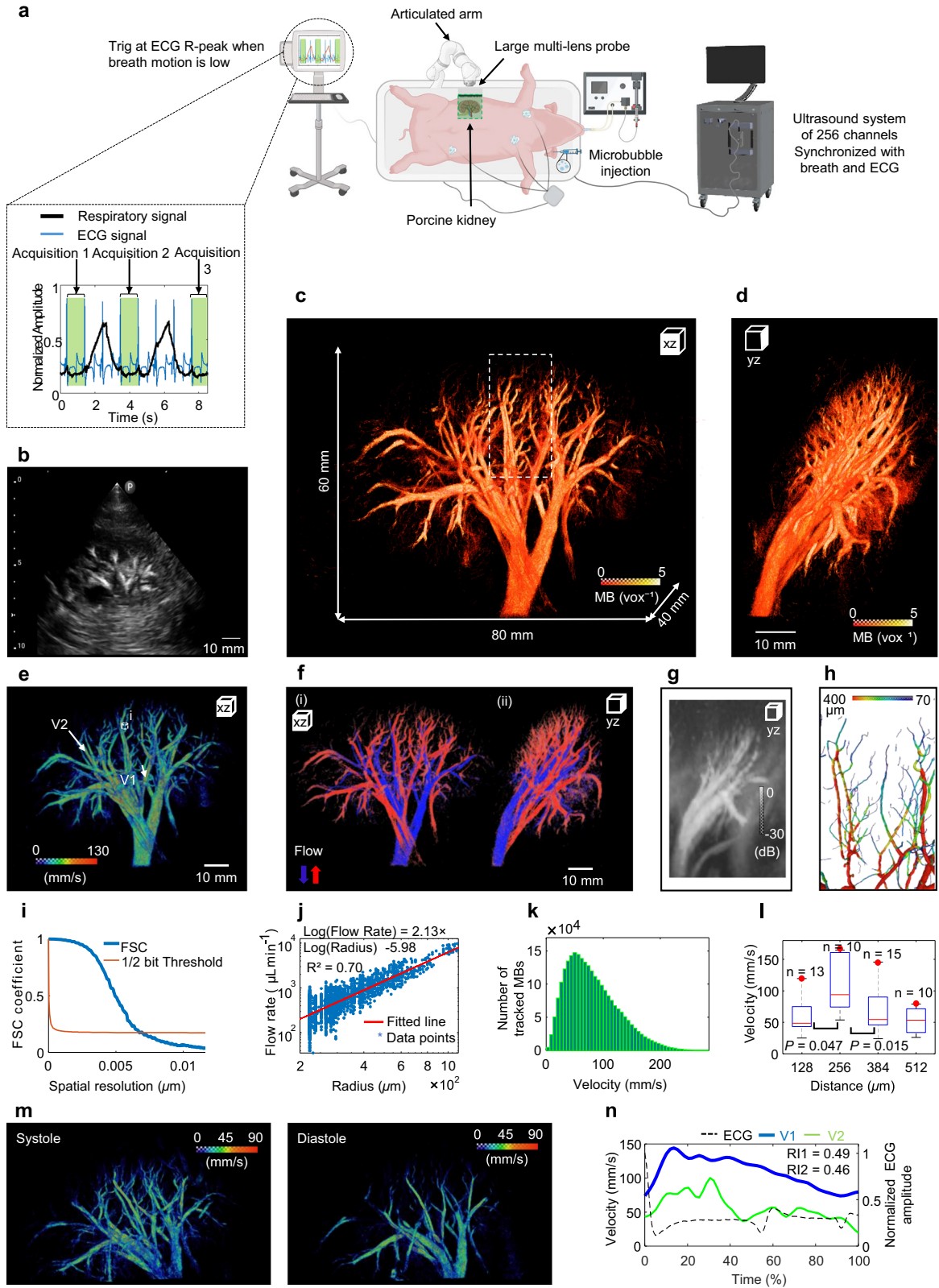

visualization of entire organs without compromising image resolution or frame rate.

The enhanced resolution allows precise quantification at a small scale, significantly improving anatomical visualization compared to standard ultrasound and CT angiography. Beyond structural imaging, the method also offers flow dynamic assessment, overcoming the temporal limitations of CT angiography. Assessing flow dynamics across all vascular scales also surpasses 4D flow MR,I which is often limited to the flow assessment of large vessels. Furthermore, the non-invasive nature and real-time capabilities make it a cost-effective and accessible alternative, reducing the financial and logistical barriers often associated with MRI.

For the explanted swine heart, a retrograde perfusion was set to perfuse the coronary network[10,14,24], which was imaged and quantified

**Fig. 4 | In vivo porcine kidney. a** Experimental setup diagram: 3D ultrasound localization microscopy (ULM) of in vivo kidney using multi-lens. Created in BioRender. Papadacci, C. (2025) https://BioRender.com/i6ehye3 and Autodesk Inventor 2025.0.1. The acquisition windows (green) triggered between two respiratory phases (solid black curve) and synchronized with the R-peak of the electrocardiogram (ECG) signal (blue curve). **b** B-mode image of the kidney captured by Philips device. **c** 3D microbubbles (MB) density map of the kidney showing the size of imaging volume ($60 \times 80 \times 40$ mm³). **d** YZ projection of kidney MB density. **e** 3D flows map of MB velocity distribution. **f** 3D flow velocity maps (XZ and YZ views) showing arterial (red) and venous (blue) flow; arrows indicate upward (red), downward (blue) directions. **g** Maximum Intensity Projection of Power Doppler kidney using the multi-lens. **h** Magnified view of the region in the white box in (**c**) ($28 \times 16 \times 38$ mm³); and his related Skeletonization to quantify vessel's radii. **i** Fourier shell correlation (FSC; solid blue curve) to estimate resolution within the 3D ULM of kidney (-147 μm). **j** Flow rate as a function of radius (blue stars) provides an exponent of 2.13 which is in a good agreement with Murray's law. **k** number of MB distribution as function of velocity. **l**, Analysis of Poiseuille flow profile of vessel (**i**) in (**e**). Significant differences between the neighbouring voxels (-128 μm) within the velocity profile were demonstrated using an unpaired two-sided Student's t-test. P values from left to right are 0.0469 and 0.0153. The red horizontal line indicates the median; boxes denote the 25th and 75th percentiles; Whiskers extend to the extreme non-outlier data points. Red dots indicate regions of peak velocity. n samples are the number of MB passing through voxels (13, 10, 15,10 from left to right). other points, outliers. Measurements are technical replicates. **m** Maps showing the MB velocity in systole and diastole phases. **n** Velocity variation over time (in percentage according to ECG) for vessels (V1 in solid blue and V2 in green) and resistivity index RI₁ and RI₂ for V1 and V2, respectively.

over the entire heart. The swine model is considered an ideal experimental model to study the coronary artery network because cardiac, vascular anatomies and dimensions are comparable to those of humans. It is often used to study coronary artery diseases because swine models have negligible collateral circulation, so each coronary artery supplies a specific cardiac region, which is ideal to induce myocardial infarction. The method would be particularly suited for preclinical cardiovascular research, particularly in the study of physiological processes, drug development and ex vivo investigations of ischaemia-reperfusion injury[25]. Note that Modulating temperature to increase motion could provide an interesting experimental model for evaluating the accuracy and robustness of motion correction algorithms. Moreover, being able to map the complex network of a swine heart, envision the capability of mapping the coronary network in humans. Direct applications include the characterization of human grafts for assessing graft quality, addressing the critical challenge of determining the donor heart's viability for transplantation[26].

The swine's kidney network was mapped over a large volume ($60 \times 80 \times 40$ mm³). The large arterial network visualized through Doppler imaging in Supplementary Fig. 5 demonstrated qualitative similarity to X-ray scan, with comparable arteries clearly identified in both imaging modalities. The entire network, including smaller vessels, were evaluated. Translation to human applications is anticipated in the near future, as no fundamental barriers exist to prevent its realization. Imaging kidney microcirculation in patients is particularly important in chronic kidney disease. In most cases, the peripheral kidney microvascular is damaged, yet there are currently no non-invasive and quantitative diagnostic tools to effectively assess this condition. Swine models of chronic kidney disease could also be a natural continuation of this work[27], where quantitative description of flow distribution could be used to detect microvascular impairment.

Liver imaging in ULM is more challenging due to respiratory, cardiac, stomach motions[28] and most of the liver is placed behind the ribs, next to the lungs, where strong reflection can occur. Still, we were able to map a large volume of liver ($65 \times 100 \times 82$ mm³) and to retrieve and quantify 3D maps. Translation to humans is envisioned with the current technology and could benefit to patients with chronic liver disease. Early detection of chronic liver diseases is particularly important, but current non-invasive diagnostic methods are insensitive to early lesions. 3D ULM has the potential to detect early liver fibrosis as well as early development of hepatic tumour.

Accurately capturing the dynamics and structure of microcirculation across entire organs enables the creation of detailed datasets that cover a broad range of physiological and pathological conditions. It would enable the development of sophisticated algorithms in artificial intelligence and machine learning capable of detecting patterns and anomalies in medical images with remarkable precision, often surpassing human capabilities. Moreover, the concept of digital twins, a virtual replica of a patient's physical characteristics, marks a significant advancement in personalized medicine. These digital models allow for simulations and predictions of disease progression and treatment outcomes. Leveraging extensive datasets, healthcare providers can customize treatments based on individual patient profiles, leading to more effective and efficient care. Thus, by mapping and analysing microcirculation across whole organs, researchers can generate vast and rich datasets that are invaluable for advancing AI-driven medical analysis, digital twins, and comprehensive medicine.

The method relies on the development of a custom matrix array and post-processing algorithms. The matrix array aperture is large compared to conventional ultrasonic matrix arrays, allowing its footprint to cover entire swine organs like the heart and the kidney. The piezoelectric elements are large, three times larger than the wavelength, allowing only 252 piezoelectric elements to cover the entire aperture. As a matter of comparison, to cover the same aperture with half-wavelength-size piezoelectric elements used in conventional arrays, a total of 15042 elements would have been needed, which is simply not achievable with current technologies. Large elements also present the advantage of delivering more energy in the medium of interest and being more sensitive in receiving compared to half-wavelength elements. This is particularly important when comparing to sparse array techniques, often limited by their low signal-to-noise ratio[29]. Large elements must be combined into a diverging lens, which has the ability to lower the directivity of large elements, which is required to maximize the antenna gain of the array. A diverging lens can be of different geometry: plano-convex, plano-concave or compound, depending on the speed of sound of the lens material. The speed of sound in the lens rules the beam divergence according to Snell's law and the material can be tuned to achieve the desired directivity. For an equal number of elements, the multi-lens array was shown to be more sensitive than the small-aperture probe and sparse array. Sensitivity of small aperture probe suffered on the side of the volume mainly due to the limited angular aperture associated to small aperture. Sensitivity of the sparse array was particularly low in the far field due to the small surface area of the small elements, which limits the amount of energy transmitted and received.

Row-column addressable (RCA) arrays have emerged as a promising alternative to fully populated matrix probes for 3D ultrasound imaging[30–32], including applications such as ultrafast imaging[33,34] and Ultrasound Localization Microscopy (ULM)[35]. By independently addressing rows and columns, RCA arrays drastically reduce the number of active channels, simplifying hardware architecture. However, the design of RCA arrays imposes certain limitations. Specifically, achieving a large aperture suitable for imaging entire organs is challenging due to the inherently large elevational dimension, which can lead to signal decorrelation and degraded image quality. To address this, recent developments have introduced RCA arrays equipped with acoustic lenses[36,37] or mechanically curved configurations[24] to enable diverging wave transmission and

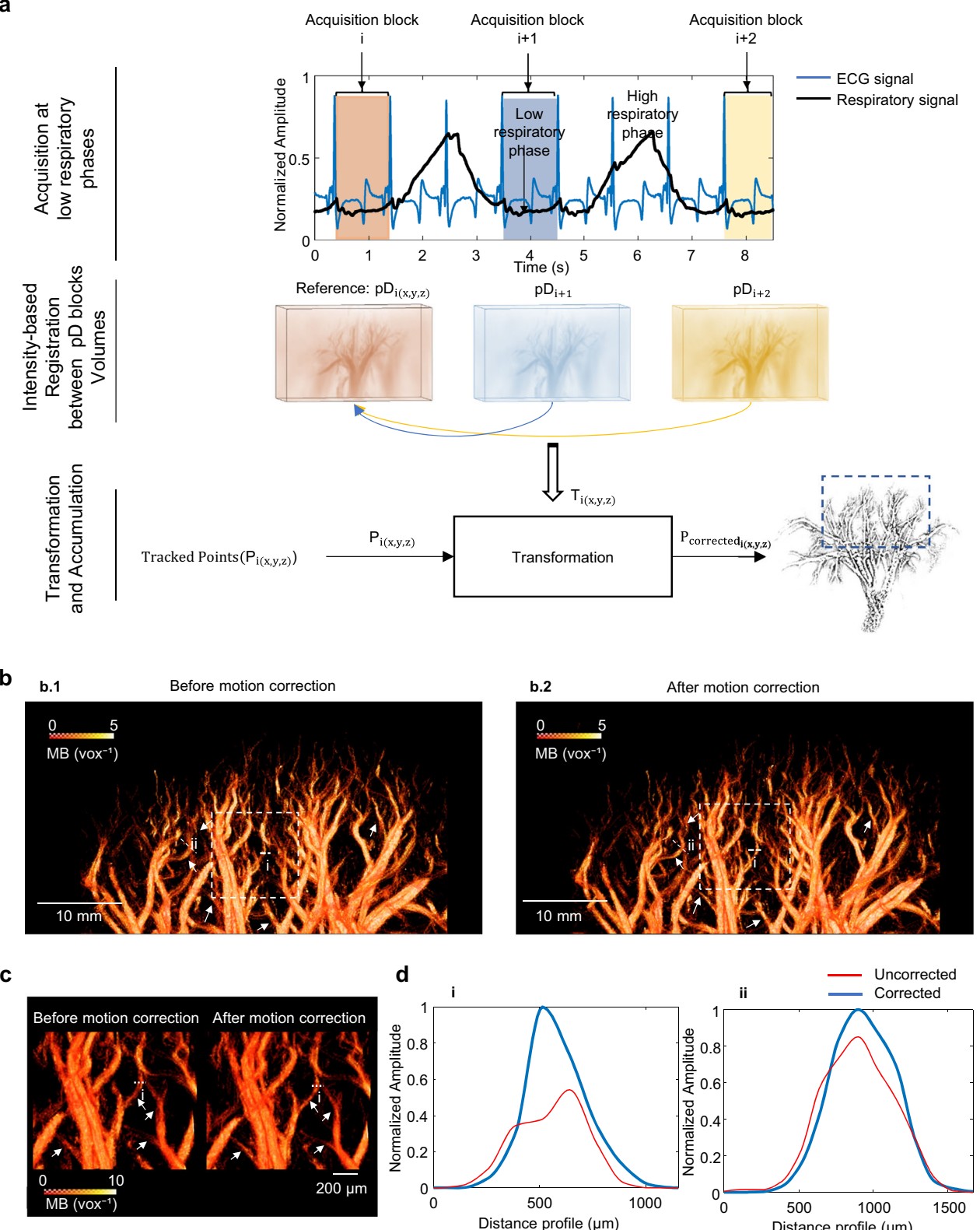

**Fig. 5 | Inter-block motion correction pipeline using 3D power Doppler intensity rigid registration. a** Illustration of acquisition blocks obtained during low respiratory phases and synchronized with the cardiac cycle. The power Doppler volumes are colour-coded according to each corresponding acquisition. $pD_{i_{(x,y,z)}}$ denotes the power Doppler block used as the reference for 3D intensity-based rigid registration. $T_{i_{(x,y,z)}}$ represents the transformation parameters resulting from intensity-based rigid registration between the reference power Doppler volume and the volume to be aligned. $P_{i_{(x,y,z)}}$ corresponds to the microbubble (MB) tracked positions within each block to be spatially corrected. $P_{corrected_{i_{(x,y,z)}}}$ represents the MB positions within each block after inter-block motion correction. **b** Motion correction results: **b.1**, density map of the kidney cortex (blue box; ~ 28 × 45 × 33 mm³) before motion correction; **b.2**, density map after correction. **c**, Magnified views corresponding to the white boxes (-23 × 13 × 18 mm³) in (**b**), highlighting detailed changes before and after correction. **d** Cross-sectional profiles of two representative vessels: **d.i**, distance profile across vessel i from (**c**), showing the normalized amplitude as a function of distance before (red curve) and after (solid blue curve) correction; **d.ii**, corresponding profile for vessel ii from (**b**).

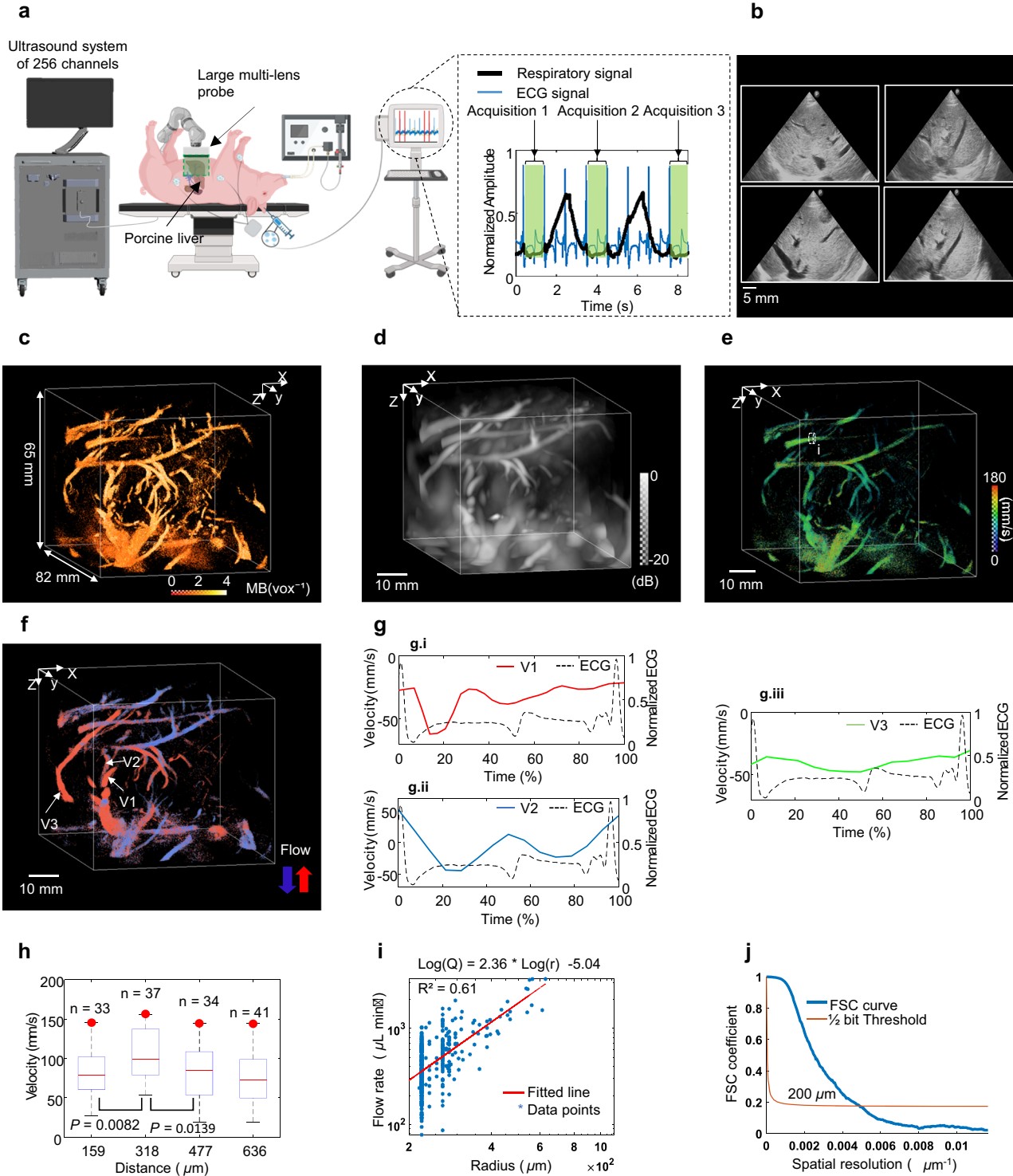

**Fig. 6 | In vivo porcine liver. a** Experimental setup diagram: Ultrasound Localization Microscopy (ULM) imaging was performed in vivo on liver using a multi-lens probe positioned above the liver and driven by a 256-channel ultrasound system. Created in BioRender. Papadacci, C. (2025) https://BioRender.com/bp11wr0 and Autodesk Inventor Professional 2025.0.1. The acquisition windows (green) were triggered between two respiratory phases (solid black curve) and synchronized with the R-wave of the electrocardiogram (ECG) signal (blue curve). **b** B-mode images of the porcine liver captured by portable device. **c** 3D MB density map of the liver, illustrating the dimensions of the imaging volume (65 × 100 × 82 mm³). **d** Maximum Intensity Projection of Power Doppler of liver using the multi-lens probe. **e** flow velocity map of tracked microbubbles (MBs). **f** 3D directional flow velocity map of MBs. Red; upward flow; Blue: downward flow. arrows indicate upward (red), downward (blue) directions. **g** Dynamic velocity variation over time (in percentage) for vessels (V1, V2 and V3) according to ECG signal in graph with dashed line: **g.i** Dynamic

velocity over time in an arterial vessel 1 (red curve). **g.ii** Dynamic velocity over time in a vein vessel 2 (blue curve). **g.iii** Dynamic velocity over time in a portal vein (green curve). **h** Analysis of the Poiseuille flow profile in vessel (i) in (**e**). Significant differences between the neighbouring voxels (-159 μm) within the velocity profile were demonstrated using an unpaired two-sided Student's t-test. P values from left to right are 0.0082, 0.0139. The red horizontal line indicates the median; boxes denote the 25th and 75th percentiles; Whiskers extend to the extreme non-outlier data points. n samples correspond to the number of microbubbles (MBs) passing through voxels (33, 37, 34 and 41 from left to right). other points, outliers. Measurements are technical replicates. Red dots indicate the regions of peak velocity. **i**, Flow rate (Q) as a function of radius (r) (blue stars; data points) provides an exponent of 2.36 which is in a good agreement with Murray's law. **j** Fourier shell correlation (FSC; solid blue curve) to estimate resolution of 3D ULM of liver (-200 μm).

increase the field of view. While the row-column approach provides higher frequencies and improved spatial resolution with a reduced number of active channels, it is better suited for superficial or medium-depth imaging. In contrast, the proposed multi-lens array was specifically designed to achieve a larger 3D field of view and greater penetration depth, which are essential for imaging deep and voluminous whole abdominal organs such as the liver, kidney, and heart. The two systems are thus complementary, each optimized for distinct anatomical targets and imaging requirements.

While fully populated matrix arrays have demonstrated volumetric ULM in localized brain regions, their field of view and application remain organ-specific[38,39]. Our custom multi-lens array approach offers scalable, whole-organ imaging across diverse deep tissues, though adaptation for transcranial human use will require dedicated redesign. The 1 MHz central frequency of the multi-lens array could be an important parameter to enable ultrasonic waves to travel through the human skull, by reducing high attenuation and aberration. Thus, it could have the potential to image the whole brain vasculature in large animal models or humans through the skull, offering promising capabilities for the early diagnosis of neurovascular pathologies[40,41]. The multi-lens array probe was characterized in vitro. The 3D Velocity map was quantified and a laminar flow was characterized by the Poiseuille profile. The profile was used to calculate a spatial resolution using dynamic information[9] and a resolution of 75 μm was found, corresponding to a resolution of λ/20, which is in agreement with the literature for ULM methods[42,43].

Cost of the system (less than €200k) was considered low in comparison to the investment usually required for fully populated matrix arrays (~€1 M).

Like any other approach, the multi-lens array approach has some limitations. In this study, 252 elements were chosen to fit the research ultrasound device composed of 256 channels. A higher number of elements could have been chosen to achieve an even larger field of view. For the same configurations, an array of 1008 elements driven with multiplexed approaches[44,45] could achieve a field of view of 200 mm by 160 mm. The current spatial resolution, estimated at $75 \times 75 \times 75 \ \mu m^3$ could be not enough to visualize the smallest capillaries (less than 10 μm). It could be improved by developing multi-lens arrays operating at higher frequency (2 to 3 MHz), while maintaining a good penetration in the far field. Thanks to 3D ULM, grating lobes do not appear to be a limitation and did not appear on the density and velocity maps due to the two thresholds applied during the localization process. The first is based on signal amplitudes which allows to suppress high random noise and grating lobes that have lower amplitudes than the main lobe of microbubbles. The second threshold is based on the value of the correlation coefficient calculated between the local maxima neighbourhood and a modelled Gaussian PSF. Since the shape of the grating lobes differs significantly from the shape of the main lobe, the remaining grating lobes were efficiently filtered out. This processing step could be optimized by modelling a more realistic PSF, which takes into account grating lobes instead of a Gaussian PSF, which only fits the main lobe. Machine learning approaches could also be an interesting solution to maximize the number of detected microbubbles[46].

Quasi real-time contrast-enhanced Doppler imaging (0.3 vol s⁻¹ at a grid reconstruction resolution of $\lambda \approx 1.5$ mm) with the multi-lens probe was employed as a guidance tool to facilitate precise probe positioning and to confirm effective acoustic coupling throughout the acquisition process. To further improve and accelerate experimental procedures, the development of a faster algorithm for real-time contrast-enhanced Doppler imaging could be investigated. In addition, for real-time B-mode imaging, B-mode-based intra-block motion correction, colour Doppler or elastography modes, grating lobes could remain an issue. To address this, the probe design could be further optimized by reducing the interelement pitch at the cost of the

aperture array size. Alternatively, introducing a randomized distribution of element positions within the aperture could break the array symmetry and suppress grating lobes. In addition to hardware-level improvements, several post-processing strategies could be employed to mitigate image artifacts and enhance resolution, including deconvolution techniques[47,48], multiply and sum beamforming techniques[49,50], spatiotemporal matrix beamforming[51], or short-lag spatial coherence of backscattered echoes[52] could be used. Finally, advanced machine learning based approaches may offer further enhancement of image quality by learning to suppress noise and artifacts adaptively[53]. Regarding the microbubble injection protocol employed in this study, it was not optimized for clinical practice. Future work will focus on improving and adapting this protocol for clinical applications.

The size of the array may cause patient discomfort when pressure is applied to ensure acoustic coupling. To alleviate this issue, an acoustically transparent pad could be placed in front of the probe to distribute pressure more evenly and enhance patient comfort[54]. Thanks to its low central frequency, the multi-lens probe is not too impacted by attenuation and imaging depth could be increased in the presence of the pad. Furthermore, probe design could be adapted and optimized to accommodate specific organ features, such as their size and vascular complexity.

Reduced sensitivity can be observed at the edges of the density volumes (coronary network, Fig. 3b). This effect may come from the central positioning of the transmit sources on the probe, resulting in lower energy delivery to the side regions of the imaged volume. To alleviate this issue, different approaches could be considered, such as increasing intersources pitch or transmitting successively with sources located on the sides of the probe. The in vivo 3D ULM images could be enhanced by applying the SVD beamformer to correct phase aberration through different tissues[55]. Furthermore, the detection of slowly moving MBs through the small vessels could be improved using adaptive spatiotemporal SVD[56], or amplitude-modulated SVD for ultrafast imaging[57].

Motion artifacts were minimized through an optimized experimental setup and acquisition strategy designed to reduce physiological motion in the data. An inter-block motion correction algorithm was applied, resulting in improved MB density and alignment of vascular structures. While B-mode–based motion correction is a common alternative in ULM literature[10,58], its effectiveness requires a high B-mode image quality, which could not be achieved by the current multi-lens probe. To assess intra-block motion, each block was subdivided into three shorter segments to compute intra-block power Doppler volumes. This analysis, detailed in the Supplemental Materials, demonstrated the presence of residual motion within blocks and the benefit of increased temporal segmentation. Further improvement could be achieved by implementing frame-by-frame intra-block motion correction, enabling more precise compensation for physiological motion such as pulsatility. SVD on short temporal sliding windows and advancing methods such as Iterative Closest Point (ICP)[59], could also be used to improve the localization of the slowest bubbles.

While individual glomeruli were not explicitly segmented in this study, their size (~150 μm) falls within the resolution range of our imaging system. Previous studies have demonstrated glomerular detection using Ultrasound Localization Microscopy[11,60], but required extensive postprocessing methods. In contrast, our approach prioritizes whole-organ 3D vascular imaging with flow dynamics, enabling global assessment of renal perfusion and microvascular integrity. Future work may incorporate dedicated segmentation strategies to extract glomerular features from these volumetric datasets.

The liver imaging results presented in this study are preliminary and currently show limited vascular detail compared to other organs. This can be attributed to the inherent complexity of hepatic microvasculature, as well as suboptimal acquisition parameters not

specifically optimized for liver imaging. While major vessels were captured in 3D with flow information, further improvements, such as optimized microbubble dilution, and longer acquisition durations, will be necessary to fully demonstrate the super-resolution potential of the system in the liver. Future work will focus on addressing these factors to enhance vascular detail and clinical relevance. While CT angiography was used to validate kidney imaging, it was not performed for the liver due to the complexity of contrast timing and breath-hold synchronization in porcine models. As liver angiography is not standard clinical or preclinical practice, the liver results presented here serve as a feasibility demonstration, with future studies planned to include anatomical validation.

In this study, we introduced a large multi-lens array probe combined with a robust post-processing pipeline to assess vascular networks and dynamics at the micro-scale across large volumes. The method was rigorously validated through simulations as well as in vitro, ex vivo, and in vivo experiments on the kidney and liver. This approach holds significant potential to become a pivotal tool for preclinical research and clinical applications, particularly when evaluating complex microcirculation networks in 3D over large fields of view. Its potential is further underscored by the simplicity of the underlying technology, making it highly accessible for widespread adoption in research laboratories and hospitals.

## Methods

### Animal ethics and management

The experiments were approved by the animal ethics committee and were carried out according to project authorization [#28654-2020121509545743]. The pigs are housed in group enclosures with solid floors, providing a surface area exceeding the standard requirements (minimum $0.7 \, m^2$ per animal). The enclosures are equipped with alternating day and night lighting, maintained at a temperature of 18 to 22 °C, with humidity levels ranging from 45% to 65%, and an air exchange rate of 10 volumes per hour. The animals are fed daily with appropriate pellets at a rate of $20 \, g \, kg^{-1}$ body weight, have continuous access to fresh drinking water, and are provided with sawdust bedding and toys to promote their well-being.

### Multi-lens array

The same multi-lens array probe was used in simulation, for in vitro validation and during ex vivo and in vivo experiments. The total aperture size of the probe was $104 \times 82 \, mm^2$. The probe was composed of three acoustic principal layers. The first layer consists of 252 large transducer matrix piezocomposite elements ($14 \times 18$). The size of an individual element was set to $4.5 \times 4.5 \, mm^2$ representing 3 times the acoustic wavelength ($3\lambda^2 \approx 4.5 \, mm^2$, wavelength; $\lambda \approx 1.5 \, mm$). The pitch between elements was set to 5.48 mm (corresponding $3.65\lambda$) in both directions, this choice represents a compromise design to avoid large lens overlap while limiting grating lobe artefacts and preserving pressure field uniformity. The transducer layer was aligned with a second layer composed of diverging acoustic lenses (plano-convex lenses), also arranged in $14 \times 18$ configuration. The two layers were assembled with structural adhesive and an acoustic matching layer. A third flat lens layer (silicon plano-concave lens) was moulded above the diverging lens layer, completing the assembly of a compound diverging lens. The compound diverging lens was made of an epoxy plano-convex lens of a maximum thickness of 3.2 mm, corresponding to a radius of curvature of 3.2 mm, a speed of sound $c_1$ of $2570 \, m \, s^{-1}$ and an acoustic impedance of 3 MRayl. A silicon plano-concave lens was set on top with a maximum thickness of 0.375 mm and a radius of curvature of 3.2 mm, a speed of sound $c_2$ of $1015 \, m \, s^{-1}$ and an acoustic impedance of 1.1 MRayl. Other layers as backing and mechanical stiffener were added behind the piezocomposite. Technical schematic details of the custom multi-lens probe are shown in Supplementary Fig. 1. The central frequency was set to 1 MHz with a 67% bandwidth to minimize

attenuation. The array used in the experiments was manufactured by Vermon®, France.

### Simulation

The multi-lens array probe performance was assessed through simulation using k-Wave toolbox[61]. The transmit pressure field of a two-cycle pulse from an individual element of the array was simulated in a volume of $60 \times 60 \times 60 \, mm^3$ within a grid spacing of $\lambda/10$ ($\lambda = 1.54$ mm) and a time step of 30 ns. It was compared to the transmit pressure field of a large element of the same size without a lens and a small element ($\lambda/2 \times \lambda/2$) used in conventional array probes. The speed of sound and density of the simulated medium were set to $1540 \, m \, s^{-1}$ and $1000 \, kg \, m^{-3}$, respectively.

The maximum acoustic pressure during propagation was recorded in the volume of interest and presented in decibels (dB) for the three elements, as shown in Fig. 1d. The amplitude profile of the pressure field was used to quantify directivity as the angle at −6 dB at a 35 mm depth along a circular arc. Comparison results for the three elements are plotted in Fig. 1d.iv.

A synthetic phantom was designed consisting in three scatterers at depths of 30, 45, 60 mm, distributed at the centre of a volume of $70 \times 104 \times 100 \, mm^3$. The multi-lens probe and a synthetic aperture imaging sequence[62] were simulated to obtain Radio Frequency (RF) signals of the scatterers. RF signals were beamformed (detailed in the section Beamforming processing) and 3D B-mode volumes were displayed to visualize numerical points spread functions as a function of depth (Supplementary Fig. 2). The simulation was executed in CUDA language, provided by the k-Wave toolbox, and processed on GPU units of NVIDIA®, model RTX A6000 (GPU Memory = 48 giga bytes).

The points spread functions were quantified by applying maximum intensity projection in the lateral direction to assess the grating lobes and the lateral resolution.

Sensitivity simulations in receive were conducted using the Field II toolbox[63,64]. The multi-lens array probe was compared to two arrays with the same frequency (1 MHz) and number of elements (252) but with small elements size ($\lambda/2 \times \lambda/2$). Sensitivity of a dense matrix probe and a large sparse array probe (with the same aperture size as the multi-lens probe) was studied.

Sensitivity can be defined as the array capability to detect weak acoustic signals. In this study, it was defined as the intensity of received signals from a small transducer by the array probes across various spatial locations.

The sensitivity responses of the three probes were evaluated through the following steps: a small transmitter, with a size of $\lambda/2$, was modelled within a $105 \times 100 \times 100 \, mm^3$ volume to emit an acoustic wave toward the matrix array probes at different locations on a grid spaced of $\lambda$ in all spatial directions. The arrays received the ultrasonic waves and simulated RF signals were stored in memory for each point location of the transmitter. The received intensity distribution over the volume was calculated from the RF signals amplitude as:

$$I(x, y, z) = \sum_{i=1}^{n} \int_{t=t_0}^{t=T} \left| RF_i(x, y, z, t) \right|^2 dt \qquad (1)$$

where $RF_i(x, y, z, t)$ is the Radio-frequency signal as a function of time t, recorded by the element $i$ for the transmitter located at the (x,y,z) coordinates. $t_0$ and $T$ are the start and end recording time, respectively. $n$ is the number of elements in the array.

Intensity spatial distributions were normalized with the maximum absolute intensity value obtained by the multi-lens array probe and presented in decibels scale (dB), to observe regions of highest and lowest intensity within the field of view.

The sensitivity response of the three probes was compared and assessed and by presenting the intensity threshold level as a function

of the number of voxels within the volume of $105 \times 100 \times 100$ mm$^3$ as shown in Fig. 1e. iv.

## Experiments

Several experimental studies were performed using the multi-lens array, including point spread function characterization, resolution and flow rate validations on a large flow phantom, ex vivo assessment of an isolated porcine heart, and in vivo proof of concept on a porcine kidney and liver.

For the in vitro studies, a flow phantom consisting of a polyethylene tube immersed in a water tank was imaged (inlet diameter ~870 μm, depth = 60 mm). A solution of 100 μL of microbubbles (Sonovue® Braco, Italy) mixed with 100 mL of water was continuously injected into the tube using a syringe pump (KDS Legato 130, KD Scientific, Holliston, USA) at a rate of 75 mL h$^{-1}$ during acquisition. Four different flow rates were studied (75 mL h$^{-1}$, 100 mL h$^{-1}$, 150 mL h$^{-1}$, and 200 mL h$^{-1}$). Additionally, imaging was performed on a twisted-tube configuration to assess more complex flow patterns. This setup used a polyethylene tube with an inner diameter of 870 μm. The tube was inclined, extending from 42 mm to 80 mm of imaging depth. A microbubble solution diluted at a 1:1000 ratio was injected at a flow rate of 200 mL h$^{-1}$.

For the ex vivo experiment on the isolated perfused heart, a porcine heart from a $30 \pm 1$ kg pig was anesthetized, explanted, cannulated at the aorta. Retrograde perfusion was established through the aorta according to the Langendorff method[10,14,24], with an oxygenated Krebs-Henseleit solution and a perfusion rate of approximately 0.25 L min$^{-1}$. The heart was immersed in a box made of a TPX polymer, chosen for its low acoustic attenuation and impedance mismatch. Heart motion was reduced by setting the Krebs solution temperature to 3 C°. A 100% concentration of MBs solution was added to the perfusion flow at 0.25 mL min$^{-1}$ and injected into the aorta using a syringe pump set to a rate of 15 mL h$^{-1}$, corresponding to a 1:1000 dilution of the initial MB solution. The multi-lens array was positioned in the apical view orientation (Fig. 3a).

For the in vivo experiments, the studies were performed on three swines (12 weeks old and weighing 30 to 36 kg). The swines were anesthetized with sedation with Zoletil 0.1 mL kg$^{-1}$ IM, then Comfortan 1 mL IV, then isoflurane 2% anaesthesia with air-oxygen mixture, after tracheal intubation and the use of a Minerve Alpha 100® respirator. A trained physician performed a renal angiography in porcine 1. X-ray dye (iodixanol, Visipaque™ 320, GE Healthcare) was injected through the catheter and GE® (model OEC 9900 Elite) X-ray generator was used to image the kidney artery network. The images of renal angiography of porcine 1 were stored in memory and illustrated in the Supplementary Fig. 5. Real-time B-mode imaging was performed by a trained physician with a linear phased array connected to portable ultrasound scanner (Philips® Lumify, probe S4-1U) to acquire B-mode images for initial anatomical localization and alignment. The images were stored in memory. The multi-lens array probe was positioned in front of the kidney and fixed using an articulated arm. Subsequently, quasi-real-time contrast-enhanced Doppler imaging with the multi-lens probe was employed as a guidance tool to facilitate precise probe positioning and to confirm effective acoustic coupling throughout the acquisition process. An initial bolus of 1 mL of microbubble solution was injected through the intravenous system into the porcine ear. This was followed by additional intermittent bolus injections of 0.5 mL every minute for the rest of the acquisition. The ultrasound acquisitions were triggered between two respiratory phases to minimize motions artefacts. Similarly, real-time B-mode imaging was performed by a trained physician to image the liver network after MBs injection. The multi-lens array probe was positioned in front of the liver, fixed using an articulated arm, and an initial bolus of 1 mL of microbubble solution was injected through the intravenous system into the porcine ear. This was followed by additional intermittent bolus injections of 0.5 mL every minute for

the rest of the acquisition. The ultrasound acquisitions were triggered between two respiratory phases to minimize motion artifacts. Physiological parameters during the experiment (Electrocardiogram, respiratory signals) were monitored and acquired with a PowerLab device (ADInstruments®, Dunedin, New Zealand), equipped with spirometers and differential biological potential amplifiers, and managed by LabChart software.

To ensure efficient acoustic coupling and avoid air bubbles between the probe surface and the tissue, specifically, the skin in in vivo experiments and the heart container in ex vivo settings, a custom-degassed gel was employed for both ex vivo and in vivo experiments. The gel consisted of a 1:1 mixture of water and commercial ultrasound gel, which was degassed via centrifugation for 4 min. Following gel application to the probe surface, the probe was carefully angled and gradually applied to the tissue to eliminate air bubbles and ensure optimal acoustic coupling.

## Ultrasound sequences

The ultrasound acquisition sequences were performed using the multi-lens array probe described in section (Multi-lens array) driven by a programmable ultrasound device with 256 transmit-receive channels (Vantage, Verasonics®, USA). Transmit schemes were programmed to perform 3D ultrafast ultrasound imaging in order to preserve a sufficient temporal resolution. The selected configuration consisted of 16 sequential transmissions, each using a single element of the probe to emit a diverging wave. The spatial arrangement of these 16 sources is positioned in the centre of the probe. For each transmission, the source was located at the physical position of the active element on the transducer surface as illustrated in the Supplementary Fig. 2b. Back-scattered echoes were recorded with all elements after each transmission and RF data were stored in memory, digitalized at 100% bandwidth (two samples per wavelength). Diverging wave emissions consisted in short pulses of two cycles centred at a frequency of 1 MHz. Given the imaging depth, the pulse repetition frequency was set to 5000 Hz and associated volume rate was 312.5 Hz. The acquisitions were divided into blocks of 830 ms, corresponding to 256 volumes for the in vitro and ex vivo experiments, and into blocks of 1.2 s for the in vivo kidney and liver experiments. The number of blocks in one acquisition varied depending on the application: 1000 blocks were acquired for the tube phantom, 400 blocks for the ex vivo heart, 290 blocks for the kidney and 133 blocks for the liver porcine. Each block was repeated every 300 ms, allowing the RF data block to be stored on the memory disk. The acquisition durations were as follows: 13 min for the flow phantom imaging shown in Fig. 2, 5 minutes for the isolated porcine heart in Fig. 3, 5 minutes for the in vivo kidney in Fig. 4, and 162 s for the in vivo liver in Fig. 6. The total amount of data per acquisition block varied from 1.1 gigabytes to 2.8 gigabytes.

## 3D motion correction

3D rigid motion correction was performed between blocks using a Power Doppler-based registration workflow[14]. Specifically, both the 3D Power Doppler volumes and the corresponding microbubble (MB) tracked positions for each block were used as inputs. Intensity information in 3D power Doppler images was used to compute the transformation that maximized the Advanced Normalized Correlation between a selected reference Power Doppler volume (chosen as the one with the highest number of tracked MBs) and each of the other 3D Power Doppler volumes to be aligned.

The registration algorithm was implemented using Elastix 5.0 (https://elastix.lumc.nl/) and performed in full 3D space. A rigid transformation comprising both rotation and translation components was estimated and subsequently applied to the microbubble (MB) tracks to spatially align all datasets across blocks. The transformation matrix was computed using Elastix's rigid parameter configuration, which employs a multi-resolution registration framework based on the

*MultiResolutionRegistration* method and uses a B-spline interpolator (*BSplineInterpolator*) in combination with the Euler transform model.

For intra-block motion correction presented in the Supplementary Fig. 4, we implemented a 3D registration strategy within each acquisition block. Each block was arbitrary divided into three temporal sub-blocks, for which 3D Power Doppler volumes were computed and rigidly registered using the same framework as inter-block correction. The corrected sub-blocks were then concatenated to form a motion-compensated volume for the full block. This corrected Power Doppler volume, along with the corrected microbubble tracks, was then used as input for the standard inter-block motion correction.

## Reporting summary

Further information on research design is available in the Nature Portfolio Reporting Summary linked to this article.

## Data availability

Data availability statement: All 3D volumes (tubes, heart, kidney and liver, provided in both density and speed) are available as NIfTI files at https://doi.org/10.5281/zenodo.16965337 (ref. 65) enabling independent analysis and quantification.

## Code availability

All codes used to generate the results and figures in this study are available upon request.

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

## Acknowledgements

This work was supported by Inserm research accelerator (Inserm ART) in Biomedical Ultrasound and was funded by the European Union, European Research Council (ERC) grant agreement number 101042470 to C.P. The GPU RTX A6000 used in this work was awarded by the NVIDIA Academic Hardware Grant Programme to C.P. We gratefully acknowledge Vermon® for manufacturing the custom multi-lens ultrasound probe. We thank Alexandre Dizeux (Physics for Medicine) for his great assistance with data visualization and for creating the cover image.

## Author contributions

C.P., M.T., M.P. and N.H. conceived the study. N.H., H.F. and J.R. designed the simulation pipeline. P.M. designed the Langendorff ex vivo setup. A.B., L.S., J.D., P.C. and B.G. designed the in vivo setup. N.H., H.F. and C.P. developed the acquisition sequence. N.H., C.P. acquired in vitro, ex vivo, in vivo data. N.H. and C.P. processed the data. N.H. and C.P. wrote the first draft. All authors edited and approved the final version of the manuscript.

## Competing interests

The authors declare no competing interests.
