## [Transparent Peer Review file · Nature Communications]

Multi-lens ultrasound arrays enable large scale three-dimensional micro-vascularization characterization over whole organs

Corresponding Author: Dr Clement Papadacci

Version 0:

Reviewer comments:

Reviewer #1

(Remarks to the Author)

The manuscript presents an effort to implement volumetric Ultrasound Localization Microscopy (ULM) in large animal models using a custom-built 252-element probe combined with diverging lens. While the topic is of clear interest for advancing super-resolution ultrasound imaging, the study lacks sufficient methodological clarity, technical rigor, and validation to support its claims. Key experimental steps are insufficiently documented, several core results are questionable or inconsistent, and the overall evidence presented does not convincingly establish the feasibility of reliable 3D ULM in complex in vivo organs. In its current form, the manuscript does not meet the standards for publication. Major and minor concerns are detailed below.

Major Comments

1. The manuscript does not provide sufficient technical details regarding the custom 252-element probe design or its fabrication. For the field to meaningfully assess and reproduce the findings, more comprehensive specifications, schematics, and performance benchmarks are required.
2. The in vitro demonstration uses only a single microtube, which limits the assessment of the system's ability to resolve complex 3D vascular structures. To truly demonstrate 3D ULM capabilities, the system should be challenged with multiple adjacent vessels—ideally closely spaced or intertwined structures. Previous studies have used such validation strategies, including twisted tubes (Favre et al., Phys. Med. Biol., 2023), star-shaped geometries (Coudert et al., IEEE TUFFC, 2024), or tightly packed parallel tubes with sharp edges (Heiles et al., IEEE TMI, 2019). A similar experimental setup is necessary to confirm that the proposed system can resolve fine spatial features and vessel separation in three dimensions.
3. The microbubble concentration used in the in vitro experiments is too high to allow for proper Ultrasound Localization Microscopy. At such densities, individual microbubbles cannot be resolved, making super-resolution imaging fundamentally unfeasible. The authors should significantly dilute the contrast agent — typically around 1/8000 of the original concentration — to enable effective sparse localization and proper ULM processing.
4. The manuscript briefly mentions the use of a two-threshold method for localization — one based on signal amplitude to suppress noise and grating lobes, and another based on a correlation coefficient with a modeled Gaussian PSF. However, this explanation remains too vague to understand the actual implementation and performance of the localization pipeline. Given the importance of this step in any ULM workflow, a detailed, step-by-step description is necessary. The authors should provide a supplementary figure clearly illustrating the localization process, including example images at each stage of thresholding and correlation filtering.
5. The authors claim a resolution of 75 μm , yet the Fourier Shell Correlation (FSC) measurements reported are 125 μm in the heart, 147 μm in the kidney, and 200 μm in the liver. This discrepancy needs to be addressed, and resolution claims should be revised accordingly.

6. While the authors correctly refer to the isolated perfused heart as "ex vivo" in the main text, the manuscript title misleadingly describes this as an "in vivo" demonstration. This inconsistency could mislead readers about the actual scope of the study. Furthermore, the ex vivo heart imaging is not a novel contribution, as similar experiments using a Row-Column array were already published by the same group (Caudoux et al., IEEE TMI, 2024). This significantly reduces the originality of the work and should be explicitly acknowledged and discussed.

7. The claim of whole-organ kidney imaging is compromised by the inability to visualize glomeruli, which are essential for assessing kidney function. Successful glomerular imaging has been demonstrated in both rat models and humans using ultrasound localization microscopy (Denis et al., 2023, EBioMedicine; Bodard et al., 2023, Investigative Radiology). The absence of glomerular detection and segmentation in this study raises concerns about the depth and clinical relevance of the kidney imaging presented.

8. The liver imaging results presented do not convincingly demonstrate volumetric super-resolution capabilities. The visualizations are limited, with sparse vascular detail, and fail to reach the standard expected for a meaningful in vivo liver application.

9. Motion compensation is crucial in 3D ULM, especially for in vivo imaging. While the authors mention using a 3D Power Doppler rigid registration workflow between blocks, they fail to demonstrate or validate the effectiveness of this correction. Notably, intra-block motion, which can be significant, is not addressed. Furthermore, there is no visual or quantitative evidence showing the impact of motion correction on ULM images, such as before-and-after comparisons. The manuscript lacks a detailed explanation of the cross-correlation used in the registration process — is it applied in 3D? What function(s) is/are employed? A figure illustrating the correction steps and a plot showing how motion compensation was assessed are essential for clarity and transparency.

10. The authors do not present any comparative imaging, such as contrast-enhanced Doppler, to highlight the added value of 3D ULM. This comparison is particularly important to justify the complexity of the technique over existing modalities.

11. While CT angiography is shown for the kidney, no such comparison is presented for the liver. Including this data is necessary to validate the ULM imaging in the liver and to assess anatomical accuracy.

12. The authors suggest that their custom 252-element probe is necessary to enable volumetric ULM in large organs. However, fully populated matrix arrays have already been used successfully for transcranial 3D ULM in large brains, such as in nonhuman primates (Xing et al., 2025, EBioMedicine) and sheep (Coudert et al., 2024, IEEE TUFFC). Moreover, the physical size of the proposed array appears too large for practical use in human imaging, particularly through the temporal bone, which is a common acoustic window for cerebral applications. This limitation further challenges the clinical translatability of the proposed system.

Minor comments

1. L.63 "Diverging lenses are used to restore high imaging performances" is unclear. It would benefit from a brief explanation of how the lenses contribute to performance improvement and in what specific way.

2. Line 68, the phrase "whole heart, kidney and liver ULM imaging" may give the impression that all organs were imaged in vivo. Since the heart experiment was performed ex vivo, it would be helpful to specify this clearly to avoid confusion.

3. L.136: "Grating lobes were filtered out by the tracking algorithm" would benefit from a clearer explanation of the method used. As noted in the major comments, more detail on the localization and filtering steps—ideally with a visual example—would improve transparency.

4. Line 139: "indicating a resolution of 75 μm ." This statement is not supported by appropriate resolution metrics. Resolution should ideally be assessed by the system's ability to distinguish two closely spaced structures rather than based on the size of a single reconstructed tube. Please refer to Major Comments regarding more rigorous in vitro validation strategies.

5. Line 464: "A bolus of 1 mL MBs solution was injected... followed by a bolus injection of 0.5 mL/min..." This phrasing is unclear — is the 0.5 mL/min delivery a continuous infusion, or a repeated bolus every minute? Clarifying whether this is a continuous infusion or intermittent dosing would help readers better understand the microbubble injection protocol.

6. Line 477: "16 sources positioned on elements at the probe center were selected..."

It would be helpful to provide more details about the source positions and their characteristics — for example, the angular range, direction of emission, and how the sources were distributed.

7. Line 484: "400 blocks for the ex vivo heart, 290 blocks for the kidney and 133 blocks for the liver porcine..." It would be useful to explain why different numbers of blocks were used for each organ. Is the variation due to the organ's size, complexity, or other technical limitations? A more consistent approach, such as using 400 blocks for all organs, might improve comparison and reproducibility.

8. Line 504: "For the tracking, a simpletracker algorithm has been used based on the Hungarian algorithm..." Please add the maximum linking distance used for the tracking algorithm and clarify how it was chosen for each experiment.

Reviewer #2

(Remarks to the Author)

In this paper, the authors aim to improve 3D microcirculation mapping using Ultrasound Localization Microscopy (ULM) by extending the 3D field of view with a newly proposed probe. The initial version of the probe was previously introduced by the same group in earlier publications, which focused on simulations and in-vitro acquisitions (with only 16 elements). In this paper, a more finalized version of the probe is presented, incorporating more elements and used in in-vitro, ex-vivo, and in-vivo acquisitions.

The manuscript is well-written and easy to follow. The ULM results are impressive and demonstrate the strong potential of the proposed probe for future large-organ acquisitions. However, as with any 2D or 3D ULM imaging, motion correction is necessary to achieve optimal image quality. In this study, motion correction is applied only between acquisition blocks and not within each block. I believe this issue should be addressed or at least investigated further.

Additionally, I would be interested in a discussion or comparison between the proposed probe and another probe developed by the same group, described in the paper: "Curved toroidal row-column addressed transducer for 3D ultrafast ultrasound imaging." Both probes feature a low number of elements while allowing for a large 3D field of view. Although the row-column array operates at a higher frequency, its 90% bandwidth should allow it to transmit pulses similar to those of the proposed probe. A discussion of the benefits of the new probe in comparison with other probes would enhance the manuscript, as there is currently little discussion or no discussion at all on row-column array probes, which also offer a larger field of view than matrix arrays.

Overall, I find the proposed work to be excellent, with a wide range of potential applications for the new probe.

INTRODUCTION

•Lines 31–32: Could you briefly explain the motivation behind choosing a volume of $100 \times 100 \times 100 \text{ mm}^3$ at such a frame rate? Is it based on the average size of the organs you are targeting? While I understand physical limitations exist for deep imaging, a short explanation would be helpful.

•Lines 43–44: Please provide references for the statement: "...ultrasound imaging is an extremely active research field, and major technical advances throughout the past decades...". I assume this refers to all areas of medical ultrasound imaging, not just flow imaging or ULM.

RESULTS

•Would it be possible to include a picture of the proposed probe? Since this is a new design, it would be informative. There is an image in the previous paper with only 16 elements, but how does the current version look with more elements? Are there any limitations in the setup? What is the probe's weight—can it be handheld? Since you use a mechanical arm for positioning, is it only usable in this configuration? What about its real-time imaging capabilities?

•Figure 1 Legend: Please provide the volume size for part d. It is included for part e, but missing for d. Also, d.iv is not described in the legend.

•Line 98: "Fig.1.d.ii" appears to refer to the small element, but it should probably be "Fig.1.d.iii".

•Line 99: "Fig 1.d.iii" — should this be referring to both d.ii and d.iii?

•Line 101: Why is the depth of 35 mm of particular interest? Is this where all elements reach peak pressure? Please include this depth information in either the Fig. 1 legend or in the plot of d.iv.

•Lines 114–116: Why did you use a SAI (Synthetic Aperture Imaging) transmission here, while in all in-vitro/ex-vivo/in-vivo acquisitions, a different transmission is used? Could the same evaluation be done using the same transmission scheme as the experimental acquisitions?

•Lines 117–121: Why is only the needle tip visible? Are you thresholding the rest of the image? Also, why are 16 sources used in this acquisition instead of SAI? Please clarify this choice in the experimental methods and explain how the 16 sources were selected (e.g., location of virtual point sources, number of elements used in sub-aperture transmissions, etc.).

•Lines 145–147: Fig. 3a looks impressive. Would the authors consider sharing a .vtk file (or other format) for improved 3D visualization of the heart?

•Lines 145–147: In the 3D motion correction section, it is mentioned that correction is applied only between blocks. Is any correction done within blocks? How much impact could intra-block motion have on image quality? Fig. 3 shows a nearly motionless acquisition (with temperature reduced to 3°C), but the stated goal of the paper is to "enable imaging and quantification of microcirculation in living organs over unprecedented large volumes." Many ULM studies use B-mode images for motion correction—could you use yours in the same way? What do your B-mode images look like?

•Lines 154 & 156 – Fig. 3f and Fig. 3h: The magnified views appear to be the same size as Fig. 3b and Fig. 3e. Could you further zoom in to provide clearer detail?

•Lines 177–180: Could you provide a B-mode image from your system for Fig. 4, and for the other in-vivo figure as well? Currently, B-mode images are shown using a different system. Does this imply that the proposed probe cannot be used alone in real-time and requires another system to find the right position? Also, given the probe's size, how well does it couple to the skin? Are you using ultrasound gel, and how confident are you that no air is trapped? How much pressure is applied to ensure good contact?

•Fig. 4g.i: This image could potentially be omitted, as g.ii provides more detail.

•Fig. 5b: As with previous figures, it would be helpful to include a B-mode image from the proposed probe.

DISCUSSION

•Extended Fig. 2: Why does your power Doppler image show more vessels than ULM? For example, the vessel indicated by the blue arrow is not visible in the ULM image. Is the threshold too high? Also, could you include similar arrows in the ULM image?

•Line 279: "Translation to human applications..." — Since most people associate ultrasound imaging with real-time, handheld probe, how complex is your setup compared to a conventional linear probe or existing 3D probes? This would be an important discussion point when presenting your B-mode images. You might include it around line 339, where you mention "To perform real-time B-mode imaging..." Have you tested any of these techniques?

•Line 302: "The matrix array aperture is extremely large..." — As mentioned earlier, discussing the practical setup on a patient (required pressure, patient comfort, etc.) would be valuable.

•Line 351: "...leave some residual motion within the block" — This is a key limitation that should be addressed. It is well

established that motion is one of the main challenges in ULM, and image quality can suffer without appropriate correction.

SIMULATION

•Line 411: Why was a synthetic aperture imaging sequence used? Is this the optimal transmission for this probe? How do you account for rapid motion in such scenarios? Why weren't simulations done using 16-source transmissions, as in the experimental setup? Could you show intensity and other evaluation results using that configuration?

EXPERIMENTS

•Line 451: You chose to reduce heart motion for the acquisition. Have you considered comparing results with and without motion to evaluate the effectiveness of your motion correction methods? Especially given that you do not do motion correction inside each block.

3D ULM DATA PROCESSING

•Line 497: Can you specify the average MB (microbubble) velocities that are filtered out by removing between 12 and 55 of the first eigenvectors? Filtering out slow-moving MBs may reduce the number of visible small capillaries. Is that why fewer small vessels are seen in the kidney?

3D MOTION CORRECTION

•Line 523: Is power Doppler-based motion correction the optimal approach? Are there prior studies using this method? Do you have references? Have you explored B-mode-based motion correction? You mention correction between blocks—what about motion within each block? If no intra-block correction is applied, does that mean the method worked for the heart only because it was stopped?

3D VISUALIZATION

•Would it be possible to share NIfTI files for 3D dataset visualization? Alternatively, could you provide a 3D PDF with predefined views of key regions?

Reviewer #3

(Remarks to the Author)

Reviewer #4

(Remarks to the Author)

Version 1:

Reviewer comments:

Reviewer #1

(Remarks to the Author)

The authors have addressed all of my previous comments. I only have a few minor revisions listed below. I recommend the paper for acceptance once these are addressed.

Minor points:

6- Line 477: "16 sources positioned on elements at the probe center were selected..."

Authors need to specify the position of the virtual source.

2- For the Extended Fig. 5], authors need to specify if a) is the Power Doppler mode with or without MBs injection. Is it Contrast enhanced PD? Please clarify this in the text as well

Reviewer #2

(Remarks to the Author)

In this second version, the authors have provided extended information about motion compensation, the transmission scheme used, and additional details about the probe and system. Based on the feedback from the other reviewers, it appears we all had similar comments regarding these sections. I am pleased that the authors have included sufficient material on their motion compensation approach to clarify its limitations and justify their choices. The paper has been significantly improved. I still have a few minor questions and comments regarding motion compensation, but overall, I am satisfied with the current version of the manuscript. The authors have addressed all my concerns, and I look forward to seeing this probe—and similar ones based on the same principles—become available to other research groups.

Motion Compensation:

• When selecting the three shorter segments to compute intra-block power Doppler volumes, why did you choose three? How did you determine the number of frames used to generate the sub-blocks? An explanation should be provided. In your reviewer response, you wrote: "Specifically, each acquisition block was divided into three sub-blocks based on time," but no further justification or rationale was given.

- At the final stage of your motion correction process, when applying the transformation matrices, do you retain all datasets? Do you perform a secondary check to assess whether motion correction was sufficient for all tracks?
- Extended Figure 4: Panels (b), (c), and (d) should match the region shown in Figure 5(a). Using the same region would allow for a fairer comparison. Additionally, in Extended Figure 4(d), please show the line profile for all motion compensation methods, including the uncorrected version. This will help evaluate whether the two-step method provides a significant improvement over the one-step method.

Quasi Real-Time:

- “Subsequently, our prototype allowed for quasi real-time imaging using Doppler mode after injecting contrast agents, which enabled to adjust fine probe placement.”

In several places—both in your response and the revised manuscript—you refer to “quasi real-time.” Could you please define what you mean by “quasi real-time”? (e.g., frame rate, latency, image quality?)

- In the final text, you do not mention that ultrasound contrast agents are necessary for quasi real-time Doppler imaging. For example, you wrote:

“Quasi-real-time Doppler imaging with the multi-lens probe was employed as a guidance tool to facilitate precise probe positioning and to confirm effective acoustic coupling throughout the acquisition process.” “Subsequently, quasi-real-time Doppler imaging with the multi-lens probe was employed as a guidance tool to facilitate precise probe positioning and to confirm effective acoustic coupling throughout the acquisition process.”

This could be misleading, as Doppler imaging can be performed without contrast agents. Please clarify in the final manuscript whether contrast agents are required to achieve quasi real-time Doppler imaging with your setup.

- Furthermore, if your ultimate goal is to use this approach in human subjects, injecting microbubbles to aid with positioning could limit the total allowable dosage for localization. Can you perform contrast-free Doppler imaging for guidance instead? Have you investigated contrast-free transmission strategies for this probe? I know you have mentioned it in your revised manuscript but please provide further information as it is a limitation of your probe if it is mandatory to inject microbubbles for Doppler imaging.

Ex Vivo and In Vivo Experiments:

- Can you provide the total amount of MBs injected during your experiments? In your updated text, you state:

“An initial bolus of 1 mL of microbubble solution was injected through the intravenous system into the porcine ear. This was followed by additional intermittent bolus injections of 0.5 mL every minute for the rest of the acquisition.”

Can you discuss whether this injection protocol would be feasible for translation to human subjects?

Other Comments:

- For all newly provided MIP Power Doppler images, please ensure the views match those of your SR images. For example, the view for the heart appear too short, and the liver image is shown from a different angle.
- Extended Figure 3: The image of the setup is incorrectly positioned—it overlaps with panel (b) rather than panel (c).
- Extended Figure 3(e): When showing the cross-section of the twisted tube, please overlay the Doppler imaging. Use a fair and consistent dynamic range when displaying the Doppler image.

Reviewer #4

(Remarks to the Author)

Reviewer #1 (Remarks to the Author):

The manuscript presents an effort to implement volumetric Ultrasound Localization Microscopy (ULM) in large animal models using a custom-built 252-element probe combined with diverging lens. While the topic is of clear interest for advancing super-resolution ultrasound imaging, the study lacks sufficient methodological clarity, technical rigor, and validation to support its claims. Key experimental steps are insufficiently documented, several core results are questionable or inconsistent, and the overall evidence presented does not convincingly establish the feasibility of reliable 3D ULM in complex in vivo organs. In its current form, the manuscript does not meet the standards for publication. Major and minor concerns are detailed below.

We would like to thank Reviewer #1 for recognizing the clear interest of our manuscript and for providing constructive feedback. We have addressed all comments with a detailed point-by-point response and revised the manuscript accordingly. We believe these revisions have significantly improved the quality and clarity of the manuscript.

Major Comments

1. The manuscript does not provide sufficient technical details regarding the custom 252-element probe design or its fabrication. For the field to meaningfully assess and reproduce the findings, more comprehensive specifications, schematics, and performance benchmarks are required.

As recommended, we added more technical details regarding the custom 252-element probe on its design and fabrication in the Methods section (sub-section Multi-lens array) as follows;

“The total aperture size of the probe was $104 \times 82 \text{ mm}^2$. The probe was composed of three acoustic principal layers. The first layer consists of 252 large transducer matrix piezocomposite elements (14×18). The size of an individual element was set to $4.5 \times 4.5 \text{ mm}^2$ representing 3 times the acoustic wavelength ($3\lambda^2 \approx 4.5 \text{ mm}^2$, wavelength; $\lambda \approx 1.5 \text{ mm}$). The pitch between elements was set to 5.48 mm (corresponding 3.65λ) in both directions, this choice represents a compromise design to avoid large lens overlap while limiting grating lobe artefacts and preserving pressure field uniformity. The transducer layer was aligned with a second layer composed of diverging acoustic lenses (plano-convex lenses), also arranged in 14×18 configuration. The two layers were assembled with structural adhesive and an acoustic matching layer. A third flat lens layer (silicon plano-concave lens) was molded above the diverging lens layer, completing the assembly of a compound diverging lens. The compound diverging lens was made of an epoxy plano-convex lens of a maximum thickness of 3.2 mm, a speed of sound c_1 of 2570 m/s and an acoustic impedance of 3 MRayl. A silicon plano-concave lens was set on top with a maximum thickness of 0.375 mm, a speed of sound c_2 of 1015 m/s and an acoustic impedance of 1.1 MRayl. Other layers as backing and mechanical stiffener were added behind the piezocomposite. Technical schematic details of the custom multi-lens probe are shown in Extended Fig.1”

As a matter of clarity, specification and schematics details of probe layers were shown in the **Extended Fig. 1** and attached below as sub-figure;

Technical schematics of the custom 252-element multi-lens probe. a. Transducer specifications and dimensions: (a.i) Elevation cross section view of the probe showing the layers containing: 1 represents the composite of piezoelectric elements, 2 represents the convex lens matrix layer, 3 flat lens layer which is the silicon concave lens, 4 backing layer, 5 mechanical stiffener layer. (a.ii) Magnified view of a single compound lens shown the convex lens with a thickness of 3.2 mm and the concave lens on top with a maximum thickness of 0.375 mm. (a.iii) View of the probe's aperture plane, illustrating the total aperture size (104 mm × 82 mm) and the number of transducer elements (252). (a.iv) Magnified view of the probe's aperture plane, showing the geometry of a single transducer elements (side length: 4.5 mm), pitch (5.48 mm), and lens base diameter (6.37 mm).

2. The in vitro demonstration uses only a single microtube, which limits the assessment of the system's ability to resolve complex 3D vascular structures. To truly demonstrate 3D ULM capabilities, the system should be challenged with multiple adjacent vessels—ideally closely spaced or intertwined structures. Previous studies have used such validation strategies, including twisted tubes (Favre et al., Phys. Med. Biol., 2023), star-shaped geometries (Coudert et al., IEEE TUFFC, 2024), or tightly packed parallel tubes with sharp edges (Heiles et al., IEEE TMI, 2019). A similar experimental setup is necessary to confirm that the proposed system can resolve fine spatial features and vessel separation in three dimensions.

As recommended by the reviewer, a study on a twisted-tube was performed. The results have been added as an **Extended Fig. 1** and described in the Methods (sub-section Experiments) and the Results section, respectively;

Methods. Experiments: “Additionally, imaging was performed on a twisted-tube configuration to assess more complex flow patterns. This setup used a polyethylene tube with an inner diameter of 870 μm. The tube was inclined, extending from 42 mm to 80 mm of imaging depth. A microbubble solution diluted at a 1:1000 ratio was injected at a flow rate of 200 mL/h.”

Results: “For more complex spatial and flow patterns, the twisted-tube configuration was imaged. The 3D microbubble (MB) density map reveals a detailed reconstruction of the tubes over a large volume, clearly illustrating the separation between the intertwined tube segments. Flow direction mapping demonstrated both upward and downward flow within the structure. Quantification of flow velocity revealed a Poiseuille profile, with high spatial resolution (see Extended Fig. 2).”

Validation on twisted-tube phantom. **b**, Volumetric enhanced-contrast doppler of the twisted tube. **c**, 3D MB density of the twisted tube for an imaging volume of $(38 \times 93 \times 16 \text{ mm}^3)$. Top-left of (c): experimental setup showing the multilens probe and twisted tube in a water tank. **d**, 3D directional flow velocity map, blue indicates upward flow while red represents downward flow. **e**, Cross section of the twisted tube (white box in c) based on MB density, reveals the separation between the intertwined tube segments. **f**, Distance profile across the cross-section of the twisted tube shown in panel (e). **g**, Cross-sectional velocities in the twisted tube reveal a characteristic Poiseuille flow profile, with downward flow on the left and upward flow on the right in panel (g). Significant differences between the neighbouring voxels (voxel size = $75 \mu\text{m}$) of the velocity profiles were calculated using Student's t-test; * $P < 0.05$, ** $P < 0.01$, *** $P < 0.001$, **** $P < 0.0001$; The red horizontal line indicates the median; the boxes show the 25th and 75th percentiles; whiskers extend to the most extreme data points that are not considered outliers; The red bullet points mark the regions of peak velocity.

3. The microbubble concentration used in the in vitro experiments is too high to allow for proper Ultrasound Localization Microscopy. At such densities, individual microbubbles cannot be resolved, making super-resolution imaging fundamentally unfeasible. The authors should significantly dilute the contrast agent — typically around 1/8000 of the original concentration — to enable effective sparse localization and proper ULM processing.

We agree that appropriate microbubble dilution is crucial for effective Ultrasound Localization Microscopy (ULM). We ensured that bubble density remained sufficiently sparse to allow for reliable localization and super-resolution imaging. We successfully performed ULM on simple tube and twisted-tube phantoms using a dilution of 1/1000, which is indeed higher than the 1/8000 dilution cited by the reviewer. However, it is important to note that successful ULM validations have been reported across a broad range of microbubble concentrations by various research groups. For instance, comparable or even higher concentrations have been employed in previous studies with successful ULM reconstructions in tube phantoms:

- 1/100 dilution in (Caudoux, M. et al.2024, 10.1109/TMI.2024.3391689).
- 1/500 dilution in (Qian, X. et al.2022, 10.1109/TBME.2021.3120368).
- 1/500 dilution in (Favre, H. et al.2023, 10.1088/1361-6560/acbde3).
- 1/1000 dilution in (Lok, U.-W. et al.2022, 10.1007/s40846-022-00755-y).
- 1/2000 dilution in (Harput, S. et al.2020, 10.1109/TUFFC.2019.2943646).

The dilution used in our current study falls within this validated range.

4. The manuscript briefly mentions the use of a two-threshold method for localization — one based on signal amplitude to suppress noise and grating lobes, and another based on a correlation coefficient with a modeled Gaussian PSF. However, this explanation remains too vague to understand the actual implementation and performance of the localization pipeline. Given the importance of this step in any ULM workflow, a detailed, step-by-step description is necessary. The authors should provide a supplementary figure clearly illustrating the localization process, including example images at each stage of thresholding and correlation filtering.

As recommended by the reviewer, we have added a supplementary figure (**Extended Fig. 3.a**) which illustrates the localization and tracking pipeline in detail. This figure provides a step-by-step visualization of the 3D localization and tracking process including example images at each stage.

Localization and tracking process pipeline. a. Schematic illustration of the ULM processing pipeline, starting from IQ beamformed data, followed by SVD-based clutter filtering, microbubble (MB) localization, MB tracking, and final 3D rendering of the flow phantom.

A more detailed description of the localization pipeline has been added to the revised manuscript in the **Methods** section **3D ULM data processing**. As follows;

“Microbubble identification was performed by detecting local maxima that were cross-correlated with a Gaussian point spread function (PSF) previously characterized from in vitro measurements. Local maxima were retained as microbubbles if their intensity was above the 99th percentile of the absolute values of the filtered IQ volume (Extended Fig. 3.a.i) and their cross-correlation coefficient exceeded a threshold of 0.6 (Extended Fig. 3.a.ii). These dual thresholds effectively suppressed grating lobes by leveraging their lower intensity and distinct shape compared to the main lobe corresponding to the microbubble. MB center positions were stored over time to be tracked.”

5. The authors claim a resolution of 75 μm , yet the Fourier Shell Correlation (FSC) measurements reported are 125 μm in the heart, 147 μm in the kidney, and 200 μm in the liver. This discrepancy needs to be addressed, and resolution claims should be revised accordingly.

We thank the reviewer for this observation. The 75 μm value refers to the dynamic resolution, as defined and calculated in (Demené, C. et al.2021, 10.1038/s41551-021-00697-x), and was obtained in phantom experiments. This approach remains a robust and recognized method for evaluating the system’s intrinsic dynamic resolution.

However, as recommended by the reviewer, we used Fourier Shell Correlation (FSC) to avoid any confusion on the assessment of resolution. Consequently, we have removed the previously reported 75 μm resolution from both the abstract and the main discussion. Instead, we now report FSC-derived resolution values obtained in the organs.

It was added as:

“Combined to 3D ULM, it can map and quantify large vascular volumes (up to $120 \times 100 \times 82 \text{ mm}^3$) at high spatial resolution (from $125 \mu\text{m}$ to $200 \mu\text{m}$ depending on the organ of interest).”

6. While the authors correctly refer to the isolated perfused heart as "ex vivo" in the main text, the manuscript title misleadingly describes this as an "in vivo" demonstration. This inconsistency could mislead readers about the actual scope of the study. Furthermore, the ex vivo heart imaging is not a novel contribution, as similar experiments using a Row-Column array were already published by the same group (Caudoux et al., IEEE TMI, 2024). This significantly reduces the originality of the work and should be explicitly acknowledged and discussed.

As recommended by the reviewer, the term "*in vivo*" was removed from the manuscript title to avoid any potential misunderstanding regarding the experimental setup.

We agree that the isolated perfused heart model is not novel and has been employed in previous studies, including by our group [(Demeulenaere, O. et al.2022, 10.1016/j.jcmg.2022.02.008); (Caudoux, M. et al.2024, 10.1109/TMI.2024.3391689)], and others in 2D ULM imaging (Yan, J. et al.2024, 10.1038/s41551-024-01206-6). However, our goal in this work is not to introduce a new biological model, but rather to demonstrate the capability of our proposed multi-lens ultrasound system to image highly complex vascular structures within a substantially larger 3D imaging volume than previously reported.

To avoid any misconception that the isolated perfused heart model is novel, we have added appropriate references in both the Methods and Discussion sections.

7. The claim of whole-organ kidney imaging is compromised by the inability to visualize glomeruli, which are essential for assessing kidney function. Successful glomerular imaging has been demonstrated in both rat models and humans using ultrasound localization microscopy (Denis et al., 2023, EBioMedicine; Bodard et al., 2023, Investigative Radiology). The absence of glomerular detection and segmentation in this study raises concerns about the depth and clinical relevance of the kidney imaging presented.

The term “whole-organ” was used in reference to the unprecedented size of the field of view achieved with our approach, which enables volumetric imaging of entire organs such as the kidney.

We thank the reviewer for this interesting comment regarding glomerular imaging. We acknowledge that glomeruli are important for assessing kidney function and appreciate the opportunity to clarify our position.

Given that the typical size of glomeruli in humans is approximately $150 \mu\text{m}$, they fall within the theoretical resolution range of our imaging system (FSC of $148 \mu\text{m}$). In (Denis, L. et al.2023, 10.1016/j.ebiom.2023.104578) glomeruli were reported with an average size of $\sim 380 \mu\text{m}$ as measured by ULM, likely reflecting not only anatomical dimensions but also local vascular density and microbubble accumulation patterns. This suggests that glomerular structures or at least their “ULM signatures”, should be present in our 3D datasets.

However, as demonstrated in (Denis, L. et al.2023, 10.1016/j.ebiom.2023.104578), glomerular identification required dedicated and intensive postprocessing tools, including spatial filtering, cluster detection, and statistical validation. Similar postprocessing techniques applied to our data could possibly enable detection of glomeruli, this effort falls outside the scope of the current work.

Our study instead focuses on demonstrating the capability of a non-invasive, scalable system to perform whole-organ 3D kidney imaging with access to flow dynamics. Our method enables comprehensive mapping of the renal vasculature, including segmental and interlobular arteries and capillary zones

which are critical for evaluating perfusion heterogeneity and microvascular function. This global and functional perspective supports important clinical applications such as early detection of microvascular dysfunction, monitoring of chronic kidney disease, and non-invasive assessment of transplant viability.

We have clarified this point in the revised Discussion section:

“While individual glomeruli were not explicitly segmented in this study, their size (~150 μm) falls within the resolution range of our imaging system. Previous studies [e.g.,(Denis, L. et al.2023, 10.1016/j.ebiom.2023.104578), (Bodard, S. et al.2023, 10.1016/j.kint.2023.01.027)] have demonstrated glomerular detection using Ultrasound Localization Microscopy, but required extensive postprocessing methods. In contrast, our approach prioritizes whole-organ 3D vascular imaging with flow dynamics, enabling global assessment of renal perfusion and microvascular integrity. Future work may incorporate dedicated segmentation strategies to extract glomerular features from these volumetric datasets.”

8. The liver imaging results presented do not convincingly demonstrate volumetric super-resolution capabilities. The visualizations are limited, with sparse vascular detail, and fail to reach the standard expected for a meaningful in vivo liver application.

We thank the reviewer for this valuable comment regarding the liver imaging results. We acknowledge that, compared to the kidney and heart datasets, the liver images show sparser vascular detail and may not fully illustrate the potential of volumetric super-resolution.

Several factors may have contributed to this outcome. First, the liver’s dense microvascular architecture and high perfusion rates pose particular challenges for ULM. Additionally, imaging conditions in this experiment were not optimized specifically for liver imaging, as the primary goal of this study was to demonstrate the feasibility of large-field, multi-organ volumetric imaging.

Despite these limitations, the presented data still demonstrate the ability to capture major hepatic vessels (portal, arteries and veins) in 3D with flow dynamics. We have changed the manuscript to clarify that these liver results are preliminary, and we acknowledge that further optimization, particularly in microbubble dilution, motion correction, and acquisition duration, is needed to achieve higher-resolution reconstructions comparable to dedicated liver ULM studies.

We have also revised the Discussion to reflect these limitations and to position liver imaging as a promising direction for future refinement of the method.

“The liver imaging results presented in this study are preliminary and currently show limited vascular detail compared to other organs. This can be attributed to the inherent complexity of hepatic microvasculature, as well as suboptimal acquisition parameters not specifically optimized for liver imaging. While major vessels were captured in 3D with flow information, further improvements, such as optimized microbubble dilution and longer acquisition durations, will be necessary to fully demonstrate the super-resolution potential of the system in the liver. Future work will focus on addressing these factors to enhance vascular detail and clinical relevance.”

9. Motion compensation is crucial in 3D ULM, especially for in vivo imaging. While the authors mention using a 3D Power Doppler rigid registration workflow between blocks, they fail to demonstrate or validate the effectiveness of this correction. Notably, intra-block motion, which can be significant, is not addressed. Furthermore, there is no visual or quantitative evidence showing the impact of motion correction on ULM images, such as before-and-after comparisons. The manuscript lacks a detailed explanation of the cross-correlation used in the registration process — is it applied in 3D? What function(s) is/are employed? A figure illustrating the correction steps and a plot showing how motion compensation was assessed are essential for clarity and transparency.

We agree that motion correction is critical for in vivo volumetric ULM. In our study, a 3D rigid motion correction strategy was implemented between acquisition blocks using a Power Doppler-based

registration workflow. Specifically, each block's 3D Power Doppler volume and its associated microbubble track positions were used. The registration was performed in full 3D space using Elastix 5.0 (<https://elastix.lumc.nl/>), a well-established medical image registration toolbox. To compute the transformations, we used intensity information from the Power Doppler volumes and applied Advanced Normalized Correlation as the similarity metric. Each Power Doppler volume was registered to a reference volume, selected as the one with the highest number of tracked MBs, by estimating a rigid transformation (rotation and translation) using Elastix's Euler transform model, combined with a multi-resolution framework and B-spline interpolation. The resulting transformation matrices were then applied to the corresponding MB positions to spatially align all blocks.

A detailed explanation has been incorporated into the revised manuscript under the "3D Motion Correction" subsection of the Methods section as.

“3D rigid motion correction was performed between blocks using a Power Doppler-based registration workflow. Specifically, both the 3D Power Doppler volumes and the corresponding microbubble (MB) tracked positions for each block were used as inputs. An intensity information in 3D power Doppler images was used to compute the transformation that maximized the Advanced Normalized Correlation between a selected reference Power Doppler volume (chosen as the one with the highest number of tracked MBs) and each of the other 3D Power Doppler volumes to be aligned.

The registration algorithm was implemented using Elastix 5.0 (<https://elastix.lumc.nl/>) and performed in full 3D space. A rigid transformation comprising both rotation and translation components was estimated and subsequently applied to the microbubble (MB) tracks to spatially align all datasets across blocks. The transformation matrix was computed using Elastix's rigid parameter configuration, which employs a multi-resolution registration framework based on the *MultiResolutionRegistration* method and uses a B-spline interpolator (*BSplineInterpolator*) in combination with the Euler transform model.”

As recommended by the reviewer, we added a new figure, (**Fig. 5**) to the revised manuscript that illustrates the full motion correction workflow along with a before-and-after visual and quantitative comparison of the ULM data on two vessels, demonstrating the effectiveness of the motion compensation method applied between acquisition blocks.

The Fig.5 was commented in the results section as:

“Fig.5.a illustrates the implemented 3D rigid inter-block motion correction pipeline. For each acquisition block at a low respiratory phase, a Power Doppler volume reconstruction was generated. An intensity-based registration between these volumetric Power Doppler blocks was then performed to compute a transformation, which was subsequently applied to the corresponding tracked microbubble points for each block.

As a result, Fig.5.b presents a qualitative comparison of the cortical density map of the kidney before (Fig5.b.1) and after (Fig.5.b.2) motion correction. The Fig.5.c represents a magnified view of the region highlighted by the white box in Fig.5.b to appreciate the effect of the correction. Additionally, a quantitative comparison of vessel density measurements (vessels i and ii) shows an increase in density after motion correction, indicating improved vessel visualization and alignment.”

Inter-block motion correction pipeline using 3D power Doppler intensity rigid registration. **a**, Illustration of acquisition blocks obtained during low respiratory phases and synchronized with the cardiac cycle. The power Doppler volumes color-coded according to each corresponding acquisition. $pD_{i(x,y,z)}$ denotes the power Doppler block used as the reference for 3D intensity-based rigid registration. $T_{i(x,y,z)}$ Represents the transformation parameters resulting from intensity-based rigid registration between the reference power Doppler volume and the volume to be aligned. $P_{i(x,y,z)}$ corresponds to the microbubble (MB) tracked positions within each block to be spatially corrected. $P_{corrected i(x,y,z)}$ represents the MB positions within each block after inter-block motion correction. **b**, Motion correction results: **b.1**, density map of the kidney cortex (blue box; $\sim 28 \times 45 \times 33 \text{ mm}^3$) before motion correction; **b.2**, density map after correction. **c**, Magnified views corresponding to the white boxes ($\sim 23 \times 13 \times 18 \text{ mm}^3$) in (b), highlighting detailed changes before and after correction. **d**, Cross-sectional profiles of two representative vessels: **d.i**, distance profile across vessel i from panel (c), showing the normalized amplitude as a function of distance before (red curve) and after (blue curve) correction; **d.ii**, corresponding profile for vessel ii from panel (b).

We acknowledge the reviewer's point that intra-block motion was not addressed in this version of the pipeline. However, intra-block motion was minimized through fixed probe with an articulated arm, respiratory triggering, short block durations which stay within a low respiratory phase and anaesthesia. Nevertheless, as suggested by the reviewer, we investigated intra-block motion capability and demonstrated both the methodology and feasibility of our correction approach. The results are presented in Extended Fig.4.

Intra-block motion correction pipeline using 3D power Doppler intensity rigid registration. **a**, Illustration of acquisition blocks obtained during low respiratory phases and synchronized with the cardiac cycle. Each acquisition is divided into three sub-phases, and the corresponding power Doppler volumes are reconstructed and color-coded by phase. $sub - pD_{i(x,y,z)}^j$ denotes the sub power doppler block used as the reference for 3D intensity-based rigid registration for the others sub power dopplers within one block. $T_{i(x,y,z)}$ represents the transformation parameters resulting from registration between the reference sub-volume and each sub-volume to be aligned. $P_{i(x,y,z)}$ corresponds to the microbubble (MB) tracked positions within each sub-block prior to correction, and $P_{T_i(x,y,z)}$ represents the MB positions after intra-block correction. The concatenated result of all sub-volumes after rigid transformation is denoted as $pD'_{T_i(x,y,z)}$ which constitutes the full power Doppler volume for one cardiac cycle. At the inter-block level, $T'_{i(x,y,z)}$ denotes the transformation parameters derived from intensity-based rigid registration between the reference power Doppler block and each subsequent block. $P_{corrected_{i(x,y,z)}}$ represents the MB positions within each block after inter-block motion correction. **b**, Density map of the kidney cortex (blue box; $\sim 39 \times 33 \times 43 \text{ mm}^3$) before motion correction. **c**, Density map of the kidney cortex (blue box; $\sim 39 \times 33 \times 43 \text{ mm}^3$) after motion correction. **d**, Comparison of Cross-sectional density profile of vessel (i) in both panel b and c, showing the normalized amplitude as a function of distance before (red curve) and after (blue curve) motion correction.

We added a paragraph in the discussion section:

“Motion artifacts were minimized through an optimized experimental setup and acquisition strategy designed to reduce physiological motion in the data. An inter-block motion correction algorithm was applied, resulting in improved MB density and alignment of vascular structures. While B-mode–based motion correction is a common alternative in ULM literature [e.g., (Yan, J. et al.2024, 10.1038/s41551-024-01206-6),(Dencks, S. et al.2025, 10.1109/TUFFC.2025.3543322)], its effectiveness requires a high B-mode image quality which could not be achieved by the current multi-lens probe. To assess intra-block motion, each block was subdivided into three shorter segments to compute intra-block power Doppler volumes. This analysis, detailed in the Supplemental Materials, demonstrated the presence of residual motion within blocks and the benefit of increased temporal segmentation. Further improvement could be achieved by implementing frame-by-frame intra-block motion correction, enabling more precise compensation for physiological motion such as pulsatility.”

10. The authors do not present any comparative imaging, such as contrast-enhanced Doppler, to highlight the added value of 3D ULM. This comparison is particularly important to justify the complexity of the technique over existing modalities.

We thank the reviewer for this important suggestion.

As recommended by the reviewer, contrast-enhanced Doppler images have been added as; Fig.3.d (ex vivo heart), Fig.4.g (in vivo kidney), and Fig.6.d (liver), as well to Extended Fig.3.b for the twisted-tube phantom validation.

d, Maximum Intensity Projection (MIP) of Power Doppler data acquired from the isolated porcine heart with the multi-lens probe.

g, Maximum Intensity Projection (yz view) of Power Doppler data acquired from a porcine kidney with the multi-lens probe.

d, Maximum Intensity Projection of Power Doppler data acquired from a porcine liver with the multi-lens probe.

b, Volumetric enhanced-contrast doppler of the twisted tube.

11. While CT angiography is shown for the kidney, no such comparison is presented for the liver. Including this data is necessary to validate the ULM imaging in the liver and to assess anatomical accuracy.

While CT angiography was used to validate kidney imaging, we did not perform it for the liver due to technical and logistical challenges. In large animal models such as pigs, liver CT angiography is not part of standard clinical or research protocols and requires complex contrast timing and respiratory control, making routine acquisition difficult.

We agree that CT would be valuable for anatomical validation of liver ULM and have clarified in the manuscript that liver imaging is presented here as a feasibility demonstration. Future studies focused on the liver will incorporate appropriate reference imaging to support anatomical validation. It was added to the discussion as:

“While CT angiography was used to validate kidney imaging, it was not performed for the liver due to the complexity of contrast timing and breath-hold synchronization in porcine models. As liver angiography is not standard clinical or preclinical practice, the liver results presented here serve as a feasibility demonstration, with future studies planned to include anatomical validation.”

12. The authors suggest that their custom 252-element probe is necessary to enable volumetric ULM in large organs. However, fully populated matrix arrays have already been used successfully for transcranial 3D ULM in large brains, such as in nonhuman primates (Xing et al., 2025, EBioMedicine) and sheep (Coudert et al., 2024, IEEE TUFFC). Moreover, the physical size of the proposed array appears too large for practical use in human imaging, particularly through the temporal bone, which is a common acoustic window for cerebral applications. This limitation further challenges the clinical translatability of the proposed system.

We thank the reviewer for highlighting recent advances in 3D ULM using fully populated matrix arrays in large organs such as the brain. We agree that studies such as (Xing, P. et al.2025, 10.1016/j.ebiom.2024.105457) and (Coudert, A. et al.2024, 10.1109/TUFFC.2024.3432998) represent significant progress in the field and demonstrate the potential of matrix arrays for volumetric imaging in large animal models.

However, our study addresses a different challenge: enabling large volumetric coverage across multiple deep organs (e.g., heart, liver, kidney) with an extended field of view. While previous referenced studies achieved high resolution in localized brain regions, they were constrained to smaller fields of view and organ-specific setups. In contrast, our approach aims to provide a versatile and adaptable platform capable of covering full-organ volumes.

We agree with the reviewer that the probe was not developed for transcranial applications. A redesign would be necessary for human transcranial imaging, to target the temporal bone. Further optimization in both size and design would be required. This optimization lies beyond the scope of the current study. It was added in the discussion section as:

“While fully populated matrix arrays have demonstrated high-resolution volumetric ULM in localized brain regions, their field of view and application remain organ-specific [(Xing, P. et al.2025, 10.1016/j.ebiom.2024.105457) and (Coudert, A. et al.2024, 10.1109/TUFFC.2024.3432998)]. Our custom multi-lens array approach offers scalable, whole-organ imaging across diverse deep tissues, though adaptation for transcranial human use will require dedicated redesign.”

Minor comments

1. L.63 “Diverging lenses are used to restore high imaging performances” is unclear. It would benefit from a brief explanation of how the lenses contribute to performance improvement and in what specific way.

A clearer explanation has been added to the manuscript as follows:

"Diverging lenses are used to restore a low directivity, compensating for the reduced angular aperture typically associated with large elements. This design improves the focusing quality and enhances image resolution."

2. Line 68, the phrase “whole heart, kidney and liver ULM imaging” may give the impression that all organs were imaged in vivo. Since the heart experiment was performed ex vivo, it would be helpful to specify this clearly to avoid confusion.

As recommended by the reviewer, to avoid any confusion regarding the scope of the imaging performed, we have clarified in the revised manuscript in the **Introduction** section as follows:

“Applied to map organs in a large animal swine model, we demonstrate ULM imaging over massive volumes (up to $120 \times 100 \times 82 \text{ mm}^3$), including ex-vivo heart, in-vivo kidney and liver”

3. L.136: “Grating lobes were filtered out by the tracking algorithm” would benefit from a clearer explanation of the method used. As noted in the major comments, more detail on the localization and filtering steps—ideally with a visual example—would improve transparency.

To improve clarity and transparency, we have added an Extended Fig.3.a to the revised manuscript illustrating the key steps of the processing pipeline, including how grating lobes were filtered out by the tracking algorithm.

4. Line 139: "indicating a resolution of $75 \mu\text{m}$." This statement is not supported by appropriate resolution metrics. Resolution should ideally be assessed by the system’s ability to distinguish two closely spaced

structures rather than based on the size of a single reconstructed tube. Please refer to Major Comments regarding more rigorous in vitro validation strategies.

As recommended by the reviewers, we performed additional in vitro validation using a more complex phantom consisting of a twisted-tube geometry.

This setup allowed us to demonstrate the system's ability to resolve closely spaced structures thus providing a more rigorous and representative assessment of the imaging performance. These new results are presented in Extended Fig.3 and discussed in the revised manuscript.

5. Line 464: "A bolus of 1 mL MBs solution was injected... followed by a bolus injection of 0.5 mL/min..." This phrasing is unclear — is the 0.5 mL/min delivery a continuous infusion, or a repeated bolus every minute? Clarifying whether this is a continuous infusion or intermittent dosing would help readers better understand the microbubble injection protocol.

The phrasing "0.5 mL/min" refers to an intermittent bolus injection, meaning that a 0.5 mL bolus of microbubble solution was injected **once** every minute, rather than as a continuous infusion.

As recommended, we have revised the manuscript to clearly describe the microbubble injection protocol as follows:

"An initial bolus of 1 mL of microbubble solution was injected through the intravenous system into the porcine ear. This was followed by additional intermittent bolus injections of 0.5 mL every minute for the rest of the acquisition."

This clarification has been added to the revised **Methods** section **Experiments**.

6. Line 477: "16 sources positioned on elements at the probe center were selected..."

It would be helpful to provide more details about the source positions and their characteristics — for example, the angular range, direction of emission, and how the sources were distributed.

We thank the reviewer for this important observation. We have clarified it in the revised manuscript in section "**Ultrasound sequences**." as follows:

"Transmit schemes were programmed to perform 3D ultrafast ultrasound imaging in order to preserve a sufficient temporal resolution. The selected configuration consisted of 16 sequential transmissions, each using a single element of the probe to emit a diverging wave. The spatial arrangement of these 16 sources is positioned in the center of the probe, and illustrated in the Extended Fig.2.b. Backscattered echoes were recorded with all element after each transmission"

For more clarification about the source positions, schematics of the positioning sources were added to the Extended Fig.2.b.

b, Schematic illustration of the ultrafast acquisition scheme using the multilens probe, showing 16 transmit apodizations (TX), where each source is activated individually on a single element during each transmission. During reception, all elements are activated after every transmit event.

7. Line 484: "400 blocks for the ex vivo heart, 290 blocks for the kidney and 133 blocks for the liver porcine..." It would be useful to explain why different numbers of blocks were used for each organ. Is the variation due to the organ's size, complexity, or other technical limitations? A more consistent approach, such as using 400 blocks for all organs, might improve comparison and reproducibility.

We agree with the reviewer that using 400 blocks for all acquisitions would likely improve image quality and consistency across datasets. However, the variation in the number of blocks was due to the specificity of each acquisition.

For the ex vivo heart, a continuous acquisition was feasible because the heart was stopped, allowing us to acquire 400 blocks without the need for synchronization or triggering with physiological parameters. In contrast, for the in-vivo kidney and liver imaging sessions, acquisitions were triggered to synchronize with physiological motion, (Respiration and heart motion), which introduced longer acquisition durations and practical limitations (e.g., anesthesia duration). Therefore, we limited a low number of blocks for the kidney the liver to ensure high-quality data while maintaining experimental feasibility.

8. Line 504: "For the tracking, a simpletracker algorithm has been used based on the Hungarian algorithm..." Please add the maximum linking distance used for the tracking algorithm and clarify how it was chosen for each experiment.

We have added the maximum linking distance used for the tracking algorithm in the revised manuscript as follows:

“For the tracking, the Simpletracker algorithm was used, which is based on the Hungarian algorithm (<https://github.com/tinevez/simpletracker>, Jean-Yves Tinevez, 2021). A maximum linking distance of 0.96 mm (approximately 2 pixels) between two subsequent positions was allowed. Tracks smaller than 5 frames were rejected.”

This distance was chosen as a compromise to allow the detection of high velocities in large vessels, while avoiding erroneous connections in smaller vessels that could degrade tracking accuracy.

Reviewer #2 (Remarks to the Author):

In this paper, the authors aim to improve 3D microcirculation mapping using Ultrasound Localization Microscopy (ULM) by extending the 3D field of view with a newly proposed probe. The initial version of the probe was previously introduced by the same group in earlier publications, which focused on simulations and in-vitro acquisitions (with only 16 elements). In this paper, a more finalized version of the probe is presented, incorporating more elements and used in in-vitro, ex-vivo, and in-vivo acquisitions.

The manuscript is well-written and easy to follow. The ULM results are impressive and demonstrate the strong potential of the proposed probe for future large-organ acquisitions. However, as with any 2D or 3D ULM imaging, motion correction is necessary to achieve optimal image quality. In this study, motion correction is applied only between acquisition blocks and not within each block. I believe this issue should be addressed or at least investigated further.

Additionally, I would be interested in a discussion or comparison between the proposed probe and another probe developed by the same group, described in the paper: “Curved toroidal row-column addressed transducer for 3D ultrafast ultrasound imaging.” Both probes feature a low number of elements while allowing for a large 3D field of view. Although the row-column array operates at a higher frequency, its 90% bandwidth should allow it to transmit pulses similar to those of the proposed probe. A discussion of the benefits of the new probe in comparison with other probes would enhance the manuscript, as there is currently little discussion or no discussion at all on row-column array probes, which also offer a larger field of view than matrix arrays.

Overall, I find the proposed work to be excellent, with a wide range of potential applications for the new probe.

We would like to thank the reviewer for their positive and constructive comments. We are glad that the manuscript was found to be clear and that the potential of the proposed probe was appreciated.

We fully agree that motion correction is a critical component of 3D ULM imaging. In this revised version, we have expanded the description of our motion compensation approach and clarified that the current implementation focuses on inter-block rigid motion correction using 3D Power Doppler registration. While intra-block motion correction remains a challenge and was not initially implemented in this study, we proposed a method in the revised manuscript in the supplemental materials, which leverages motion estimation from Doppler-based volumes and correction of microbubble positions within blocks. We also added an entire paragraph dedicated to motion correction in the discussion.

We also included a paragraph in the discussion on the RCA probes and its comparison to our approach. While the row-column approach provides higher frequencies and improved spatial resolution with a reduced number of active channels, it is better suited for superficial or medium-depth imaging. In contrast, the proposed multi-lens array was specifically designed to achieve a larger 3D field of view and greater penetration depth, which are essential for imaging deep and voluminous whole abdominal organs such as the liver, kidney, and heart. The two systems are thus complementary, each optimized for distinct anatomical targets and imaging requirements. This comparison was added in the discussion as;

“Row-column addressable (RCA) arrays have emerged as a promising alternative to fully populated matrix probes for 3D ultrasound imaging (Rasmussen, M. F. et al.2013, 10.1109/ULTSYM.2013.0370) (Rasmussen, M. F. et al.2015, 10.1109/TUFFC.2014.006531), (Christiansen, T. L. et al.2015, 10.1109/TUFFC.2014.006819), including in applications such as ultrafast imaging (Flesch, M. et al.2017, 10.1088/1361-6560/aa63d9), (Sauvage, J. et al.2020, 10.1109/TMI.2019.2959833) and Ultrasound Localization Microscopy (ULM) (Tan, Q. et al.2024, 10.1109/TBME.2024.3426487). By independently addressing rows and columns, RCA arrays drastically reduce the number of active channels, simplifying hardware architecture. However, the design of RCA arrays imposes certain limitations. Specifically, achieving a large aperture suitable for imaging entire organs is challenging due to the inherently large elevational dimension, which can lead to signal decorrelation and degraded image quality. To address this, recent developments have introduced RCA arrays equipped with acoustic lenses (Audoin, M. et al.2023, 10.1109/IUS51837.2023.10306503), (Salari, A. et al.2025, 10.1109/TUFFC.2025.3526523) or mechanically curved configurations (Caudoux, M. et al.2024, 10.1109/TMI.2024.3391689) to enable diverging wave transmission and increase field of view. While the row-column approach provides higher frequencies and improved spatial resolution with a reduced number of active channels, it is better suited for superficial or medium-depth imaging. In contrast, the proposed multi-lens array was specifically designed to achieve a larger 3D field of view and greater penetration depth, which are essential for imaging deep and voluminous whole abdominal organs such as the liver, kidney, and heart. The two systems are thus complementary, each optimized for distinct anatomical targets and imaging requirements.”

We believe these additions improve the manuscript and provide a clearer perspective on both the system's capabilities and future directions.

INTRODUCTION

•Lines 31–32: Could you briefly explain the motivation behind choosing a volume of $100 \times 100 \times 100$ mm³ at such a frame rate? Is it based on the average size of the organs you are targeting? While I understand physical limitations exist for deep imaging, a short explanation would be helpful.

We thank the reviewer for the comment. The choice of a $100 \times 100 \times 100 \text{ mm}^3$ volume is indeed motivated by the average size of large organs such as the liver, kidney and the heart.

The frame rate is a result of a trade-off between imaging depth and the number of compounding transmissions. As such depth of 100 mm, a 300 Hz frame rate gives a sufficient temporal resolution for microbubble tracking while enabling coherent compounding of multiple transmits (16).

The system is not physically limited in imaging depth, as the low-frequency transducer (center frequency $\sim 1 \text{ MHz}$) ensures low attenuation and allows imaging beyond 10 cm. Furthermore, the probe design can be adapted and optimized for specific organ characteristics, including size and vascular complexity.

We have included this clarification in the Discussion section;

“Thanks to its low central frequency, the multi-lens probe is not too impacted by attenuation and imaging depth could be increased in the presence of the pad. Furthermore, probe design could be adapted and optimized to accommodate specific organ features, such as their size and vascular complexity.”

•Lines 43–44: Please provide references for the statement: “...ultrasound imaging is an extremely active research field, and major technical advances throughout the past decades...”. I assume this refers to all areas of medical ultrasound imaging, not just flow imaging or ULM.

We thank the reviewer for this suggestion. As requested, we have added relevant references in the revised manuscript to support the statement regarding the continuous and significant technical advances.

The references that we added; [(Tanter, M. et al.2014, 10.1109/TUFFC.2014.2882), Ultrasound Elastography: Review of Techniques and Clinical Applications (Sigrist, R. M. S. et al.2017, 10.7150/thno.18650), A Review on Real-Time 3D Ultrasound Imaging Technology (Huang, Q. et al.2017, 10.1155/2017/6027029)

RESULTS

•Would it be possible to include a picture of the proposed probe? Since this is a new design, it would be informative. There is an image in the previous paper with only 16 elements, but how does the current version look with more elements? Are there any limitations in the setup? What is the probe’s weight—can it be handheld? Since you use a mechanical arm for positioning, is it only usable in this configuration? What about its real-time imaging capabilities?

As recommended by the reviewer, we added a photograph of the probe and the ultrasound system in Extended Fig.1.b and Fig.1.b.c in the revised manuscript, and attached below as sub-figure;

Photograph of the custom 252-element multi-lens probe and ultrasound imaging system. b. Illustration of the custom 252-element multi-lens probe, showing the aperture plane at the top and the elevation view of the probe at the bottom of the image. **c.** Photograph of the multi-lens probe connected to the 256-channel ultrasound imaging system.

The probe weighs **415.5 grams**, which makes it suitable for handheld acquisitions.

Conventional B-mode imaging is challenging with the current probe configuration. Further optimization could be required to perform real time (an approach could be the use of machine learning strategies)(Stevens, T. S. W. et al.2024, 10.1109/TMI.2024.3363460).

However, a quasi-real-time Doppler mode was used for positioning the probe. It provided sufficient anatomical details, as shown by the maximum intensity projection of power Doppler in Extended Fig.5.a included in the revised manuscript.

a, Maximum Intensity Projection in the XZ plane of Power Doppler from a porcine kidney, acquired using the custom multilens array probe.

Extra care must be taken to avoid air bubbles trapped between the probe and the skin, particularly given the large surface area of the array, which can make uniform coupling more challenging. To mitigate this, we used a custom-made degassed coupling medium, consisting of a 1:1 mixture of water and commercial ultrasound gel, which significantly reduced bubble formation and improved acoustic coupling.

Details of the Degassed gel preparation and application have been added to the Methods in Experiments subsection.

“To ensure efficient acoustic coupling and avoid air bubbles between the probe surface and the tissue, specifically, the skin in *in vivo* experiments and the heart container in *ex vivo* settings, a custom-degassed gel was employed for both *ex vivo* and *in vivo* experiments. The gel consisted of a 1:1 mixture of water and commercial ultrasound gel, which was degassed via centrifugation for 4 minutes. Following gel application to the probe surface, the probe was carefully angled and gradually applied to the tissue to eliminate air bubbles and ensure optimal acoustic coupling.”

Additionally, we have expanded the Discussion to include the use of quasi real-time Doppler imaging to facilitate optimal probe positioning and verify acoustic coupling during acquisition.

“Quasi-real-time Doppler imaging with the multi-lens probe was employed as a guidance tool to facilitate precise probe positioning and to confirm effective acoustic coupling throughout the acquisition process.”

•Figure 1 Legend: Please provide the volume size for part d. It is included for part e, but missing for d. Also, d.iv is not described in the legend.

We thank the reviewer for pointing this out. We have now added the volume size in the **fig.d** legend as “3D maps of maximum acoustic pressure field in transmission, illustrating directivity and transmit pressure performances over a volume of $60 \times 60 \times 60 \text{ mm}^3$ ”

As recommended by the reviewer, we have also added the missing description of **Fig.d.iv** in the legend of **Fig.d** as

“(d.iv) Graphs show the angular directivity [degrees] as a function of intensity threshold level, measured at a depth of 35 mm from the maximum pressure field shown in Fig. d. for three configurations: single lens with large element (green), large element without lens (blue), and small element (red)”

•Line 98: “Fig.1.d.ii” appears to refer to the small element, but it should probably be “Fig.1.d.iii”.

We thank the reviewer for catching this. We have corrected this in the revised manuscript as “Fig.1.d.iii” rather than “Fig.1.d.ii” in line 98.

•Line 99: “Fig 1.d.iii” — should this be referring to both d.ii and d.iii?

We thank the reviewer for catching this. In line 99 the reference should be “Fig.1.d.ii” rather than “Fig.1.d.iii”. We have corrected this in the revised manuscript.

•Line 101: Why is the depth of 35 mm of particular interest? Is this where all elements reach peak pressure? Please include this depth information in either the Fig. 1 legend or in the plot of d.iv.

A depth of 35 mm was selected because it lies beyond the Fresnel zone, ensuring that the measured pressure field corresponds to the far-field. Any other deeper regions could also have been selected, for evaluating angular directivity.

As recommended, we have added the information of the depth to the legend of **Fig. 1** in the revised manuscript, as follow;

“(d.iv) Graphs show the angular directivity [degrees] as a function of intensity threshold level, measured at a depth of 35 mm”

•Lines 114–116: Why did you use a SAI (Synthetic Aperture Imaging) transmission here, while in all in-vitro/ex-vivo/in-vivo acquisitions, a different transmission is used? Could the same evaluation be done using the same transmission scheme as the experimental acquisitions?

We thank the reviewer for this observation. In the simulation, we used a Synthetic Aperture Imaging (SAI) scheme consisting of 252 sequential transmissions, to evaluate the performance of the probe under optimal conditions. In contrast, for the *in-vitro*, *ex-vivo*, and *in-vivo* acquisitions, we used the same transmission approach with a reduced sequential transmission (16 sequential transmission) to perform ultrafast imaging.

As recommended by the reviewer, we have conducted in simulation using the 16-transmission configuration as used in the experimental acquisitions.

We evaluated intensity as a function of lateral dimension. It has been added to the revised manuscript (see Extended Fig.2.c and Extended Fig.2.d), and the related discussion has been included in Section Results as follow

“The simulated points spread functions (PSF) of the multi-lens array probe was performed and quantified, revealing a grating lobe level of approximately -25.50 dB, -24.57 dB, and -25.36 dB, and lateral resolution was found to be 1.15 mm, 1.17 mm, and 1.26 mm at -6 dB for imaging depths of 30 mm, 45 mm, and 60 mm, respectively. These results were obtained using synthetic aperture imaging with 252 transmit sources. Additionally, we evaluated image quality under ultrafast imaging conditions using a reduced number of transmit events (16 sources), to reflect the practical configuration used during acquisitions. The corresponding results are presented in the Extended Fig. 2.”

Simulation validation of the point spread function with the Multi-Lens Probe. **c**, Numerical 3D point spread functions (PSFs) of the multi-lens array using SAI, evaluated at the center of the medium for depths of 3 cm, 4.5 cm, and 6 cm. **d**, Quantitative comparison of the PSF intensity profiles at -6 dB for SAI (panel **d.i**) and ultrafast transmission with 16 sources (panel **d.ii**). The blue graph represents the maximum intensity projection profile of the PSF along the lateral dimension at a depth of 3 cm, while the red and yellow graphs correspond to depths of 4.5 cm and 6 cm, respectively.

•Lines 117–121: Why is only the needle tip visible? Are you thresholding the rest of the image? Also, why are 16 sources used in this acquisition instead of SAI? Please clarify this choice in the experimental methods and explain how the 16 sources were selected (e.g., location of virtual point sources, number of elements used in sub-aperture transmissions, etc.).

The needle diameter is 0.5mm which is smaller than the wavelength of the probe (wavelength of the probe ~1.5 mm). As a result, the system is unable to resolve the needle shaft. However, the needle tip generates strong echoes, making its detection and measurement possible to evaluate the PSF. No thresholding was applied to the rest of the image.

For the in-vitro experiment, we used 16 sequential transmit sources instead of the 252 sequential transmit (SAI) to image the needle, as part to evaluate and validate the same transmission configuration that was used later across all in-vitro, ex-vivo, and in-vivo acquisitions. Sources were chosen in the center of the probe, sub-aperture corresponded to a single element according to synthetic aperture scheme.

As recommended by the reviewer, we have clarified the configuration schemes, location and number of sources, as shown below and in Extended Fig.2.b, and in description of the **Experimental Methods** section in the revised manuscript, under **Ultrasound Sequences**, as follow;

“Transmit schemes were programmed to perform 3D ultrafast ultrasound imaging in order to preserve a sufficient temporal resolution. The selected configuration consisted of 16 sequential transmissions, each using a single element of the probe to emit a diverging wave. The spatial arrangement of these 16 sources is positioned in the center of the probe, and illustrated in the Extended Fig.2.b. Backscattered echoes were recorded with all element after each transmission.”

b, Schematic illustration of the ultrafast acquisition scheme using the multilens probe, showing 16 transmit apodizations (TX), where each source is activated individually on a single element during each transmission. During reception, all elements are activated after every transmit event.

•Lines 145–147: Fig. 3a looks impressive. Would the authors consider sharing a .vtk file (or other format) for improved 3D visualization of the heart?

We thank the reviewer for the positive feedback on Fig. 3b which show the MB density map of the coronary networks. We would be glad to share a 3D visualization file of the heart. Niftii files of the MB Density are shared in a Zip folder in supplementary materials.

•Lines 145–147: In the 3D motion correction section, it is mentioned that correction is applied only between blocks. Is any correction done within blocks? How much impact could intra-block motion have on image quality? Fig. 3 shows a nearly motionless acquisition (with temperature reduced to 3°C), but the stated goal of the paper is to “enable imaging and quantification of microcirculation in living organs over unprecedented large volumes.” Many ULM studies use B-mode images for motion correction—could you use yours in the same way? What do your B-mode images look like?

We thank the reviewer for this important comment. Intra-block motion was minimized on the in vivo acquisitions by inducing anaesthesia, triggering the acquisition on the low phase of respiration, setting low block duration and by fixing the probe to an articulated arm. However, residual motion due to pulsatility for instance could remain present within a block.

As suggested by the reviewer, in the revised manuscript, we further investigated intra-block motion and implemented a 3D registration strategy within blocks. Specifically, each acquisition block was divided into three sub-blocks based on time, and 3D Power Doppler volumes were computed for each sub-block. Rigid transformations were estimated between sub-blocks using the same registration framework as for inter-block correction, with tracked microbubble positions from each sub-block used as inputs. Corrected sub-blocks were then concatenated to reconstruct a motion-compensated Power Doppler volume for the full block.

This corrected Power Doppler volume, along with the corrected microbubble tracks, was then used as input for the standard inter-block motion correction. The full intra- and inter-block motion compensation pipeline is presented in Extended Fig.4, including visual and quantitative before-and-after comparisons. The observed impact of motion correction was relatively modest, likely due to the controlled experimental conditions and the optimized acquisition. These factors collectively helped minimize intra-block displacement during acquisition. However, this pipeline could be relevant in case of larger motion.

This method was added to 3D motion correction section in the revised manuscript as: “For intra-block motion correction presented in the Extended Fig.4, we implemented a 3D registration strategy within each acquisition block. Each block was divided into three temporal sub-blocks, for which 3D Power Doppler volumes were computed and rigidly registered using the same framework as inter-block correction. The corrected sub-blocks were then concatenated to form a motion-compensated volume for the full block. This corrected Power Doppler volume, along with the corrected microbubble tracks, was then used as input for the standard inter-block motion correction”

A paragraph in the discussion was also added as:

“Motion artifacts were minimized through an optimized experimental setup and acquisition strategy designed to reduce physiological motion in the data. An inter-block motion correction algorithm was applied, resulting in improved MB density and alignment of vascular structures. While B-mode-based motion correction is a common alternative in ULM literature [e.g.,(Yan, J. et al.2024, 10.1038/s41551-024-01206-6), (Dencks, S. et al.2025, 10.1109/TUFFC.2025.3543322)], its effectiveness requires a high B-mode image quality which could not be achieved by the current multi-lens probe. To assess intra-block motion, each block was subdivided into three shorter segments to compute intra-block power Doppler volumes. This analysis, detailed in the Supplemental Materials, demonstrated the presence of residual motion within blocks and the benefit of increased temporal segmentation. Further improvement could be achieved by implementing frame-by-frame intra-block motion correction, enabling more precise compensation for physiological motion such as pulsatility.”

Regarding B-mode imaging: due to the current probe design and the low number of transmissions used in 3D ultrafast imaging, high-quality B-mode images could not be acquired with the proposed system. As such, B-mode data were not used for motion correction in this study. However, we acknowledge the usefulness of B-mode-based motion correction and plan to improve B-mode imaging quality in future probe designs. This will include the use of randomized element aligned with the lenses to reduce grating lobes and advanced machine learning strategies to enhance image quality (Stevens, T. S. W. et al.2024, 10.1109/TMI.2024.3363460).

Intra-block motion correction pipeline using 3D power Doppler intensity rigid registration. a. Illustration of acquisition blocks obtained during low respiratory phases and synchronized with the cardiac cycle. Each acquisition is divided into three sub-phases, and the corresponding power Doppler volumes are reconstructed and color-coded by phase. Sub - pD_{i(x,y,z)}^l denotes the sub power doppler block used as the reference for 3D intensity-based rigid registration for the others sub power dopplers within one block. T_{i(x,y,z)} represents the transformation parameters resulting from registration between the reference sub-volume and each sub-volume to be aligned. P_{i(x,y,z)} corresponds to the microbubble (MB) tracked positions within each sub-block prior to correction, and P_{T_{i(x,y,z)}} represents the MB

positions after intra-block correction. The concatenated result of all sub-volumes after rigid transformation is denoted as $pD'_{T_{(x,y,z)}}$ which constitutes the full power Doppler volume for one cardiac cycle. At the inter-block level, $T'_{(x,y,z)}$ denotes the transformation parameters derived from intensity-based rigid registration between the reference power Doppler block and each subsequent block. $P_{corrected_{d_{(x,y,z)}}}$ represents the MB positions within each block after inter-block motion correction. **b.** Density map of the kidney cortex (blue box; $\sim 39 \times 33 \times 43 \text{ mm}^3$) before motion correction. **c.** Density map of the kidney cortex (blue box; $\sim 39 \times 33 \times 43 \text{ mm}^3$) after motion correction. **d.** Comparison of Cross-sectional density profile of vessel (i) in both panel b and c, showing the normalized amplitude as a function of distance before (red curve) and after (blue curve) motion correction.

•Lines 154 & 156 – Fig. 3f and Fig. 3h: The magnified views appear to be the same size as Fig. 3b and Fig. 3e. Could you further zoom in to provide clearer detail?

As recommended by the reviewer, we have updated **Fig. 3f** and **Fig. 3h** to **Fig3.h** and **Fig. 3.i** respectively in the revised manuscript to provide further magnification. This correction has been implemented in the revised manuscript.

•Lines 177–180: Could you provide a B-mode image from your system for Fig. 4, and for the other in-vivo figure as well? Currently, B-mode images are shown using a different system. Does this imply that the proposed probe cannot be used alone in real-time and requires another system to find the right position? Also, given the probe's size, how well does it couple to the skin? Are you using ultrasound gel, and how confident are you that no air is trapped? How much pressure is applied to ensure good contact?

We thank the reviewer for this important question.

High-quality real-time B-mode imaging could not be performed with our current prototype. To ensure proper positioning, a conventional ultrasound system was first used to acquire B-mode images for initial anatomical localization and alignment.

Subsequently, our prototype allowed for quasi real-time imaging using Doppler mode after injecting contrast agents, which enabled to adjust fine probe placement.

The probe design could be further optimized to improve B-mode image quality, for example, by introducing randomized element configurations to reduce grating lobes. Additionally, advanced machine learning strategies could be implemented to enhance image quality (Stevens, T. S. W. et al.2024, 10.1109/TMI.2024.3363460).

For acoustic coupling, we used a custom-made degassed ultrasound gel (a 1:1 mixture of water and standard ultrasound gel) to minimize bubble formation. The probe was carefully angled and gradually applied to the tissue to eliminate air bubbles and ensure optimal acoustic coupling.

These points have been clarified and added to the methods and the discussion section of the revised manuscript as.

Methods; subsection; **Experiments:** “To ensure efficient acoustic coupling and avoid air bubbles between the probe surface and the tissue, specifically, the skin in *in vivo* experiments and the heart container in *ex vivo* settings, a custom-degassed gel was employed for both *ex vivo* and *in vivo* experiments. The gel consisted of a 1:1 mixture of water and commercial ultrasound gel, which was degassed via centrifugation for 4 minutes. Following gel application to the probe surface, the probe was carefully angled and gradually applied to the tissue to eliminate air bubbles and ensure optimal acoustic coupling.”

“Real-time B-mode imaging was performed by a trained physician with a linear phased array connected to portable ultrasound scanner (Philips® Lumify, probe S4-1U) to acquire B-mode images for initial anatomical localization and alignment. Subsequently, quasi-real-time Doppler imaging mode with the multi-lens probe was employed after injected contrast agents, serving as a guidance tool to facilitate precise probe positioning and to confirm effective acoustic coupling throughout the acquisition process.”

Discussion: “In addition, for real-time B-mode imaging, B-mode based intra-block motion correction, color Doppler or elastography modes, grating lobes could remain an issue. To address this, the probe

design could be further optimized, by reducing the interelement pitch at the cost of the aperture array size. Alternatively, introducing a randomized distribution of element positions within the aperture could break the array symmetry and suppress grating lobes. In addition to hardware-level improvements, several post-processing strategies could be employed to mitigate image artifacts and enhance resolution, including deconvolution techniques [(Yang, T. C.2016, 10.1109/OCEANSAP.2016.7485545), (Lylloff, O. et al.2015, 10.1121/1.4922516)], multiply and sum beamforming techniques [(Bertolo, A. et al.2021, 10.1109/TMI.2021.3084865), (Hansen-Shearer, J. et al.2022, 10.1109/TUFFC.2021.3122094)], spatiotemporal matrix beamforming (Berthon, B. et al.2018, 10.1088/1361-6560/aaa606), or short-lag spatial coherence of backscattered echoes (Lediju, M. A. et al.2011, 10.1109/TUFFC.2011.1957) could be used. Finally, advanced machine learning based approaches may offer further enhancement of image quality by learning to suppress noise and artifacts adaptively (Stevens, T. S. W. et al.2024, 10.1109/TMI.2024.3363460).”

•Fig. 4g.i: This image could potentially be omitted, as g.ii provides more detail.

As recommended by the reviewer, we have removed Fig. 4g.i from the revised version of the manuscript.

•Fig. 5b: As with previous figures, it would be helpful to include a B-mode image from the proposed probe.

B-mode images acquired with the proposed probe were not included, as anatomical structures were not sufficiently visible. This limitation is acknowledged and discussed in the Limitations section of the revised manuscript. Improvements in B-mode image quality are feasible through further optimization of the probe design and post-processing algorithms, and this represents an active area of ongoing development. However, high-quality B-mode imaging was not the primary objective of this study. Importantly, we were still able to achieve high-quality 3D ULM results despite the lack of detailed anatomical B-mode guidance.

DISCUSSION

•Extended Fig. 2: Why does your power Doppler image show more vessels than ULM? For example, the vessel indicated by the blue arrow is not visible in the ULM image. Is the threshold too high? Also, could you include similar arrows in the ULM image?

In the initial version of the manuscript, the vessel indicated by the blue arrow was not visible in the ULM image due to the high amplitude threshold. In the revised manuscript, we lowered down the thresholding parameters to better capture such vessels.

As recommended by the reviewer, we have updated the figure to include corresponding arrows in the ULM image, enabling a direct comparison between the vascular structures depicted in Doppler and ULM modes.

It is important to note that the Power Doppler image shown in this figure primarily highlights the arterial vasculature of the kidney. This Doppler acquisition was optimized to match the renal artery angiography for comparison by only selecting the systolic phase. In contrast, in ULM figures both arterial and venous flow were shown. As a result, some venous structures present in the ULM image may obscure underlying arterial vessels due to projection effects in the rendered 3D volume.

In vivo kidney of porcine 1. **a**, Maximum Intensity Projection in the XZ plane of Power Doppler from a porcine kidney, acquired using the custom multilens array probe.; the arrows show the similarity of the trees between the panel **a** and **b**. **b**, Renal artery angiography of the porcine kidney. **c**, 3D MB density map of the porcine kidney showing the size of imaging volume ($60 \times 90 \times 30 \text{ mm}^3$) **d**, Directional flow velocity 3D map that identifies the arterial (red) and venous (blue) flow.

•Line 279: “Translation to human applications...” — Since most people associate ultrasound imaging with real-time, handheld probe, how complex is your setup compared to a conventional linear probe or existing 3D probes? This would be an important discussion point when presenting your B-mode images. You might include it around line 339, where you mention “To perform real-time B-mode imaging...” Have you tested any of these techniques?

We thank the reviewer for this relevant comment. As previously mentioned, the current prototype is lightweight enough to be handheld by a clinician. However, to achieve the highest ULM image quality, we believe that the use of an articulated or robotic arm could be important to ensure stable positioning over long acquisitions to minimize motion artifacts.

In our in vivo studies, proper positioning of the probe was successfully achieved using quasi real-time Doppler imaging after microbubble injection. It provided sufficient feedback during the acquisition to ensure adequate probe alignment with the targeted organ.

For ongoing work, we are investigating faster algorithms to enable real-time enhanced Doppler imaging, and we are also developing improved probe designs, and artificial intelligence algorithm that aim to enhance real-time B-mode performance. These improvements would significantly accelerate experimental procedures and support future clinical applications.

These points were added in the discussion as:

“Quasi-real-time Doppler imaging with the multi-lens probe was employed as a guidance tool to facilitate precise probe positioning and to confirm effective acoustic coupling throughout the acquisition process.”

“To further improve and accelerate experimental procedures, the development of a faster algorithm for real-time enhanced Doppler imaging could be investigated.”

“In addition, real-time B-mode imaging, B-mode based intra-block motion correction, color Doppler or elastography modes, grating lobes could remain an issue. To address this, the probe design could be further optimized, by reducing the interelement pitch at the cost of the aperture array size. Alternatively,

introducing a randomized distribution of element positions within the aperture could break the array symmetry and suppress grating lobes. In addition to hardware-level improvements, several post-processing strategies could be employed to mitigate image artifacts and enhance resolution, including deconvolution techniques [(Yang, T. C.2016, 10.1109/OCEANSAP.2016.7485545), (Lylloff, O. et al.2015, 10.1121/1.4922516)], multiply and sum beamforming techniques [(Bertolo, A. et al.2021, 10.1109/TMI.2021.3084865), (Hansen-Shearer, J. et al.2022, 10.1109/TUFFC.2021.3122094)], spatiotemporal matrix beamforming (Berthon, B. et al.2018, 10.1088/1361-6560/aaa606), or short-lag spatial coherence of backscattered echoes (Lediju, M. A. et al.2011, 10.1109/TUFFC.2011.1957) could be used. Finally, advanced machine learning based approaches may offer further enhancement of image quality by learning to suppress noise and artifacts adaptively (Stevens, T. S. W. et al.2024, 10.1109/TMI.2024.3363460).”

•Line 302: “The matrix array aperture is extremely large...” — As mentioned earlier, discussing the practical setup on a patient (required pressure, patient comfort, etc.) would be valuable.

We agree with the reviewer, any probe needs a certain amount of pressure applied to ensure contact with the skin. For clinical application and to improve patient comfort, we envision to use a soft deformable pad acoustically transparent in front of the probe to absorb part of the applied pressure as it is commercially proposed by some companies (Klucinec, B.1996).

It was added in the limitation section as: “The size of the array may cause patient discomfort when pressure is applied to ensure acoustic coupling. To alleviate this issue, an acoustically transparent pad could be placed in front of the probe to distribute pressure more evenly and enhance patient comfort (Klucinec, B.1996).”

•Line 351: “...leave some residual motion within the block” — This is a key limitation that should be addressed. It is well established that motion is one of the main challenges in ULM, and image quality can suffer without appropriate correction.

We thank the reviewer for highlighting this important point. We agree that motion both between and within blocks is a key challenge in ULM and can significantly affect image quality if not properly corrected.

In the revised manuscript, we addressed this limitation by implementing and evaluating an intra-block motion correction strategy, as suggested in the previous comments. Notably, the residual intra-block motion remained low, owing to experimental conditions designed to minimize it, such as the use of a fixed articulated arm, synchronization of acquisition with the low-motion phase of respiration via anaesthesia triggering, and the selection of short block durations that coincide with this low-motion phase.

We have now clearly described this process in the revised Methods section (3D motion correction) and discuss its implications in the Discussion section.

SIMULATION

•Line 411: Why was a synthetic aperture imaging sequence used? Is this the optimal transmission for this probe? How do you account for rapid motion in such scenarios? Why weren't simulations done using 16-source transmissions, as in the experimental setup? Could you show intensity and other evaluation results using that configuration?

Full synthetic aperture imaging was used for this probe because it enables to recreate a synthetic focus everywhere in the volume. Large elements tackle the classical low transmit energy limitation often associated with synthetic aperture imaging. For rapid motion, partial synthetic aperture imaging with 16 sources were defined. Other transmission schemes may have been used such as plane or diverging wave

transmits. However, large elements may limit the ability of the probe to produce waves with high angular wavevector.

As recommended by the reviewer, we have conducted in simulation using the 16-source transmission configuration as used in the experimental acquisitions.

The results show an intensity evaluation and have been added to the revised manuscript (see Extended Fig.2.d.ii), and the related discussion has been included in Section Results as

“The simulated points spread functions (PSF) of the multi-lens array probe was performed and quantified, revealing a grating lobe level of approximately -25.50 dB, -24.57 dB, and -25.36 dB, and lateral resolution was found to be 1.15 mm, 1.17 mm, and 1.26 mm at -6 dB for imaging depths of 30 mm, 45 mm, and 60 mm, respectively. These results were obtained using synthetic aperture imaging with 252 transmit sources. Additionally, we evaluated image quality under ultrafast imaging conditions using a reduced number of transmit events (16 sources), to reflect the practical configuration used during acquisitions. The corresponding results are presented in Extended Fig.2.”

Simulation validation of the point spread function with the Multi-Lens Probe. **c**, Numerical 3D point spread functions (PSFs) of the multi-lens array using SAI, evaluated at the center of the medium for depths of 3 cm, 4.5 cm, and 6 cm. **d**, Quantitative comparison of the PSF intensity profiles at -6 dB for SAI (panel **d.i**) and ultrafast transmission with 16 sources (panel **d.ii**). The blue graph represents the maximum intensity projection profile of the PSF along the lateral dimension at a depth of 3 cm, while the red and yellow graphs correspond to depths of 4.5 cm and 6 cm, respectively.

EXPERIMENTS

•Line 451: You chose to reduce heart motion for the acquisition. Have you considered comparing results with and without motion to evaluate the effectiveness of your motion correction methods? Especially given that you do not do motion correction inside each block.

We thank the reviewer for this interesting suggestion. In the present study, we aimed to minimize cardiac motion during acquisition in order to reduce the impact of intra-block and inter-block motion on localization and tracking. The inter-block motion correction approach employed here was previously introduced and validated by (Demeulenaere, O. et al.2022, 10.1016/j.jcmg.2022.02.008).

We agree that a direct comparison between datasets acquired with and without motion minimization would provide a valuable benchmark for evaluating the effectiveness of motion correction strategies. This comparison would be especially relevant given that no intra-block motion correction is applied in our current pipeline. In fact, a parallel effort in our laboratory is currently exploring this idea, using a biological model similar to the one suggested by the reviewer, to develop and validate more advanced motion correction techniques.

It was added in the discussion as:

“Modulating temperature to increase motion could provide an interesting experimental model for evaluating the accuracy and robustness of motion correction algorithms.”

3D ULM DATA PROCESSING

•Line 497: Can you specify the average MB (microbubble) velocities that are filtered out by removing between 12 and 55 of the first eigenvectors? Filtering out slow-moving MBs may reduce the number of visible small capillaries. Is that why fewer small vessels are seen in the kidney?

In our study, SVD thresholding was adapted within each block acquisition to account for varying experimental conditions, including tissue motion, microbubble (MB) concentration and clutter intensity. Specifically, the choice of thresholds (removing between the 12th and 55th eigenvectors) was optimized empirically for each dataset to balance clutter suppression and signal preservation.

We agree with the reviewer that it could be the reason why fewer vessels are seen in the kidney. This point was clarified in the revised manuscript. “Furthermore, the detection of slowly moving MBs through the small vessels could be improved using adaptative spatiotemporal SVD, or amplitude modulated SVD for ultrafast imaging (Zhang, G. et al.2025, 10.1109/TMI.2025.3565023)”

3D MOTION CORRECTION

•Line 523: Is power Doppler-based motion correction the optimal approach? Are there prior studies using this method? Do you have references? Have you explored B-mode-based motion correction? You mention correction between blocks—what about motion within each block? If no intra-block correction is applied, does that mean the method worked for the heart only because it was stopped?

Power Doppler based motion correction has been previously employed in 3D in vivo for coronary microvascular imaging (Demeulenaere, O. et al.2022, 10.1016/j.jcmg.2022.02.008), demonstrating its usability and feasibility in the presence of strong motion. In the revised manuscript, we expanded the motion correction pipeline to include intra-block motion correction, as suggested by the reviewer. We have detailed the answer above. Briefly, we implemented a 3D registration strategy within blocks. Specifically, each acquisition block was divided into three sub-blocks based on time, and 3D Power Doppler volumes were computed for each sub-block. Rigid transformations were estimated between sub-blocks using the same registration framework as for inter-block correction, with tracked microbubble positions from each sub-block used as inputs. Corrected sub-blocks were then concatenated to reconstruct a motion-compensated Power Doppler volume for the full block. This corrected Power Doppler volume, along with the corrected microbubble tracks, was then used as input for the standard inter-block motion correction. The full intra- and inter-block motion compensation pipeline is presented in Extended Fig.4, including visual and quantitative before-and-after comparisons. The observed impact of motion correction was relatively modest, likely due to the controlled experimental conditions and the optimized acquisition. These factors collectively helped minimize intra-block displacement during acquisition. However, this pipeline could be relevant in case of larger motion for the heart for instance.

Intra-block motion correction pipeline using 3D power Doppler intensity rigid registration. **a**, Illustration of acquisition blocks obtained during low respiratory phases and synchronized with the cardiac cycle. Each acquisition is divided into three sub-phases, and the corresponding power Doppler volumes are reconstructed and color-coded by phase. $sub - pD_{i(x,y,z)}^j$ denotes the sub power Doppler block used as the reference for 3D intensity-based rigid registration for the others sub power Dopplers within one block. $T_{i(x,y,z)}$ represents the transformation parameters resulting from registration between the reference sub-volume and each sub-volume to be aligned. $P_{i(x,y,z)}$ corresponds to the microbubble (MB) tracked positions within each sub-block prior to correction, and $P_{T_i(x,y,z)}$ represents the MB positions after intra-block correction. The concatenated result of all sub-volumes after rigid transformation is denoted as $pD'_{T_i(x,y,z)}$ which constitutes the full power Doppler volume for one cardiac cycle. At the inter-block level, $T'_{i(x,y,z)}$ denotes the transformation parameters derived from intensity-based rigid registration between the reference power Doppler block and each subsequent block. $P_{corrected_{i(x,y,z)}}$ represents the MB positions within each block after inter-block motion correction. **b**, Density map of the kidney cortex (blue box; $\sim 39 \times 33 \times 43 \text{ mm}^3$) before motion correction. **c**, Density map of the kidney cortex (blue box; $\sim 39 \times 33 \times 43 \text{ mm}^3$) after motion correction. **d**, Comparison of Cross-sectional density profile of vessel (i) in both panel b and c, showing the normalized amplitude as a function of distance before (red curve) and after (blue curve) motion correction.

While B-mode–based motion correction is a common alternative in ULM literature [e.g.,(Yan, J. et al.2024, 10.1038/s41551-024-01206-6), (Dencks, S. et al.2025, 10.1109/TUFFC.2025.3543322)], its effectiveness requires a high B-mode image quality.

Due to current limitations in our probe design, our B-mode images are not yet optimal for reliable motion estimation. We are actively working on enhancing probe design (e.g., randomization of elements) and exploring advanced postprocessing algorithm to make B-mode–based correction feasible in the future.

Two paragraphs were added in the discussion as:

“In addition, for real-time B-mode imaging, B-mode based intra-block motion correction, color Doppler or elastography modes, grating lobes could remain an issue. To address this, the probe design could be further optimized, by reducing the interelement pitch at the cost of the aperture array size. Alternatively, introducing a randomized distribution of element positions within the aperture could break the array symmetry and suppress grating lobes. In addition to hardware-level improvements, several post-processing strategies could be employed to mitigate image artifacts and enhance resolution, including deconvolution techniques [(Yang, T. C.2016, 10.1109/OCEANSAP.2016.7485545), (Lylloff, O. et al.2015, 10.1121/1.4922516)], multiply and sum beamforming techniques [(Bertolo, A. et al.2021, 10.1109/TMI.2021.3084865), (Hansen-Shearer, J. et al.2022, 10.1109/TUFFC.2021.3122094)], spatiotemporal matrix beamforming (Berthon, B. et al.2018, 10.1088/1361-6560/aaa606), or short-lag spatial coherence of backscattered echoes (Lediju, M. A. et al.2011, 10.1109/TUFFC.2011.1957) could be used. Finally, advanced machine learning based approaches may offer further enhancement of image quality by learning to suppress noise and artifacts adaptively (Stevens, T. S. W. et al.2024, 10.1109/TMI.2024.3363460).”

“Motion artifacts were minimized through an optimized experimental setup and acquisition strategy designed to reduce physiological motion in the data. An inter-block motion correction algorithm was applied, resulting in improved MB density and alignment of vascular structures. While B-mode–based motion correction is a common alternative in ULM literature [e.g.,(Yan, J. et al.2024, 10.1038/s41551-024-01206-6) (Dencks, S. et al.2025, 10.1109/TUFFC.2025.3543322)], its effectiveness requires a high B-mode image quality which could not be achieved by the current multi-lens probe. To assess intra-block motion, each block was subdivided into three shorter segments to compute intra-block power Doppler volumes. This analysis, detailed in the Supplemental Materials, demonstrated the presence of residual motion within blocks and the benefit of increased temporal segmentation. Further improvement could be achieved by implementing frame-by-frame intra-block motion correction, enabling more precise compensation for physiological motion such as pulsatility”

3D VISUALIZATION

•Would it be possible to share Nifti files for 3D dataset visualization? Alternatively, could you provide a 3D PDF with predefined views of key regions?

Yes, we are happy to provide a Nifti files of the MB density. The files will be shared with the reviewers in a Zip file.

Reviewer#3

Comments The authors propose a multi-lens ultrasound array for scaling 3D ultrasound localization microscopy (ULM) to large organs in vivo. Their proposed technology addresses several limitations of conventionally used ultrasound probes for ULM, which are otherwise difficult to scale to large organs, are expensive, have a limited field of view, and often require extensive computational resources. Below are some comments that may help improve the manuscript further:

We would like to thank for the positive and constructive feedback. We greatly appreciate the reviewer's recognition of the motivation behind our work and the potential of our multi-lens ultrasound array to overcome current limitations in scaling 3D ULM to large organs in vivo. We are encouraged by your comments and have carefully addressed the points raised to further improve the quality and clarity of the manuscript. Detailed responses to each comment are provided below.

1. The authors are requested to include a discussion on row-column addressable arrays, which have been proposed as alternatives to matrix probes for 3D ULM.

As recommended by the reviewer, we added a discussion on row-column addressable arrays as follow:

“Row-column addressable (RCA) arrays have emerged as a promising alternative to fully populated matrix probes for 3D ultrasound imaging (Rasmussen, M. F. et al.2013, 10.1109/ULTSYM.2013.0370) (Rasmussen, M. F. et al.2015, 10.1109/TUFFC.2014.006531), (Christiansen, T. L. et al.2015, 10.1109/TUFFC.2014.006819), including in applications such as ultrafast imaging (Flesch, M. et al.2017, 10.1088/1361-6560/aa63d9), (Sauvage, J. et al.2020, 10.1109/TMI.2019.2959833) and Ultrasound Localization Microscopy (ULM) (Tan, Q. et al.2024, 10.1109/TBME.2024.3426487). By independently addressing rows and columns, RCA arrays drastically reduce the number of active channels, simplifying hardware architecture. However, the design of RCA arrays imposes certain limitations. Specifically, achieving a large aperture suitable for imaging entire organs is challenging due to the inherently large elevational dimension, which can lead to signal decorrelation and degraded image quality. To address this, recent developments have introduced RCA arrays equipped with acoustic lenses (Audoin, M. et al.2023, 10.1109/IUS51837.2023.10306503), (Salari, A. et al.2025, 10.1109/TUFFC.2025.3526523) or mechanically curved configurations (Caudoux, M. et al.2024, 10.1109/TMI.2024.3391689) to enable diverging wave transmission and increase field of view. While the row-column approach provides higher frequencies and improved spatial resolution with a reduced number of active channels, it is better suited for superficial or medium-depth imaging. In contrast, the proposed multi-lens array was specifically designed to achieve a larger 3D field of view and greater penetration depth, which are essential for imaging deep and voluminous whole abdominal organs such as the liver, kidney, and heart. The two systems are thus complementary, each optimized for distinct anatomical targets and imaging requirements.”

2. The authors claim a resolution of 75 μm based on the statistical difference between neighboring voxels in the velocity profile. What was the inlet flow rate during this measurement? Does the resolution change at different flow rates?

The inlet flow rate for the single straight-tube phantom during the resolution measurement (75 μm) was **75 mL/h**.

We observed that the resolution did not vary with different flow rates. This is further supported by results from the newly conducted experiment using a more complex twisted-tube phantom, in which the inlet flow rate was **200 mL/h** (see Extended Fig.3.g). In both cases, a dynamic spatial resolution of 75 μm was achieved, highlighting the robustness of resolution measurement across varying flow conditions.

b, Volumetric enhanced-contrast doppler of the twisted tube. **c**, 3D MB density of the twisted tube for an imaging volume of ($38 \times 93 \times 16 \text{ mm}^3$). Top-left of (**c**): experimental setup showing the multilens probe and twisted tube in a water tank. **d**, 3D directional flow velocity map, blue indicates upward flow while red represents downward flow. **e**, Cross section of the twisted tube (white box in **c**) based on MB density, reveals the separation between the intertwined tube segments. **f**, Distance profile across the cross-section of the twisted tube shown in panel (**e**). **g**, Cross-sectional velocities in the twisted tube reveal a characteristic Poiseuille flow profile, with downward flow on the left and upward flow on the right in panel (**g**). Significant differences between the neighbouring voxels (voxel size= $75 \mu\text{m}$) of the velocity profiles were calculated using Student's t-test; $*P < 0.05$, $**P < 0.01$, $***P < 0.001$, $****P < 0.0001$; The red horizontal line indicates the median; the boxes show the 25th and 75th percentiles; whiskers extend to the most extreme data points that are not considered outliers; The red bullet points mark the regions of peak velocity.

3. What was the resolution measurement from the phantom experiment using the Fourier Shell Correlation (FSC) method, which was otherwise used to estimate spatial resolution for the in vitro and in vivo datasets? The reported spatial resolution outside phantom study is more than $75 \mu\text{m}$ which runs counter to the claims made in the paper.

The Fourier Shell Correlation (FSC) method was not performed in the in vitro phantom study, because the FSC is less reliable when applied to an isolated structure such as a phantom tube. The FSC method performed effectively in more complex vascular network, and dense spatial features to provide a robust estimation of resolution (Heel, M. van et al.2005, 10.1016/j.jsb.2005.05.009).

We thank the reviewer for this observation. The $75 \mu\text{m}$ value refers to the dynamic resolution, as defined and calculated in (Demené, C. et al.2021, 10.1038/s41551-021-00697-x), and was obtained in phantom experiments. This approach remains a robust and recognized method for evaluating the system's intrinsic dynamic resolution.

However, as suggested by the reviewer, we used FSC to avoid any confusion on the assessment of resolution. Consequently, we have removed the previously reported $75 \mu\text{m}$ resolution from both the abstract and the main discussion. Instead, we now report FSC-derived resolution values obtained in the organs.

It was added as:

“Combined to 3D ULM, it can map and quantify large vascular volumes (up to $120 \times 100 \times 82 \text{ mm}^3$) at high spatial resolution (from $125 \mu\text{m}$ to $200 \mu\text{m}$ depending on the organ of interest).”

The $75 \mu\text{m}$ resolution reported in the phantom study represents an idealized case under optimal imaging conditions, intended to demonstrate the resolution capability of the system. We agree with the reviewer that the spatial resolution achieved in more realistic ex-vivo and in-vivo settings was higher than $75 \mu\text{m}$.

4. Do the authors observe a Poiseuille flow profile in the kidney and liver datasets?

Yes, a Poiseuille profile was observed in vessels within the kidney and liver datasets. We have now included representative examples of these flow profiles in the revised figures (see Fig.4 and Fig.6) and we provided additional discussion in the Results section to highlight and interpret these observations as;

“The Fig.4.l shows a Poiseuille flow profile by analysing the velocity profile in porcine kidney within the vessel (i) illustrated in the Fig.4.e. A statistical Student’s t-test revealed significant differences in velocity values along the profile between neighbouring voxels (voxel size = 128 μm) containing tracked MB velocities (*P < 0.05)”

Analysis of the Poiseuille flow profile for kidney. I. Analysis of the Poiseuille flow profile in vessel (i) in panel e. Significant differences between the neighbouring voxels (128 μm) within the velocity profile were demonstrated using Student’s t-test; *P < 0.05; The red horizontal line indicates the median; the boxes show the 25th and 75th percentiles; whiskers extend to the most extreme data points that are not considered outliers. The red bullet points mark the regions of peak velocity.

“The Fig.6.h presents a Poiseuille flow profile by analysing the velocity distribution within the vessel (i) in porcine liver flow velocity illustrated in the Fig.6.e. A statistical Student’s t-test revealed significant differences in velocity values along the profile between neighbouring voxels (voxel size = 159 μm) containing tracked MB velocities (*P < 0.05, **<0.01).”

Analysis of the Poiseuille flow profile for liver. h. Analysis of the Poiseuille flow profile in vessel (i) in panel e. Significant differences between the neighbouring voxels (159 μm) within the velocity profile were demonstrated using Student’s t-test; *P < 0.05, **P < 0.01; The red horizontal line indicates the median; the boxes show the 25th and 75th percentiles; whiskers extend to the most extreme data points that are not considered outliers. The red bullet points mark the regions of peak velocity.

5. How are arterial and venous vessels differentiated in the kidney and liver datasets?

We thank the reviewer for this question.

Arterial and venous vessels in the kidney were differentiated based on flow axial direction (Z direction), upward vessels are considered to be the arterial vessels and downward vessels the venous system. This was also validated with anatomical comparison, we compared the vascular structures identified in our ULM velocity maps to anatomical renal artery angiography in the Extended Fig.5.b. The observed red directional flow in our ULM images closely matches the arterial vasculature shown in angiographic images, supporting the identification of these vessels as arteries.

“3D directional flow velocities reveal the arterial flow system in red and the venous flow system in blue”

In the liver dataset, arterial and venous vessels were differentiated based on flow dynamics. Specifically, the pulsatility patterns. These identifications were supported by literature-based references that characterize typical flow profiles in hepatic arteries and veins (Gallix, B. P. et al.1997, 10.2214/ajr.169.1.9207514).

Arteries generally exhibit higher velocities in the systolic phase and pulsatile flow patterns synchronized with the cardiac cycle, while hepatic veins show very specific pattern with retrograde flow between systolic and diastolic phase. These dynamic characteristics, along with anatomy, guided our interpretation and labelling of vascular structures in the liver.

It was added in the Results as:

“Dynamic velocity measurement was performed for arterial (pulsatile), vein (pulsatile/retrograde) and portal (no pulsatility) vein vessels, representing in V1, V2 and V3 respectively in the Fig.6.g”

6. In Fig. 4gii, the velocity appears to change direction from +50 to -50 mm/s. The authors are requested to explain this observation further.

We thank the reviewer for pointing this out. We believe the reviewer is referring to Fig. 5gii, which displays the velocity profile within a hepatic vein in the liver. The observed variation in velocity, ranging from approximately +50 mm/s to -50 mm/s, reflects the characteristic pulsatile flow within the hepatic venous system, as previously described by (Gallix, B. P. et al.1997, 10.2214/ajr.169.1.9207514). This flow pattern arises from the direct anatomical and functional connection between the hepatic veins and the right atrium. As a result, hepatic venous flow closely follows the cardiac cycle, exhibiting phasic variations driven by atrial contraction and relaxation dynamics.

This flow pattern was interpreted in the revised manuscript and clarified in the results section of the updated Fig.6 as;

“The Fig.6.gii, shows the velocity profile within a hepatic vein in the liver, reflecting the characteristic pulsatile flow of the hepatic venous system”

7. The statement: "To maximize energy transmission from the transducer to the biological soft tissues, the acoustic impedance of the compound lens must be as close as possible as the acoustic impedance of the biological soft tissues. Thus, the density of the compound lens must be taken into account." needs clarification. It should be noted that optimal impedance matching layer taking into account both transducer impedance and tissue impedance (not necessarily just matching the tissue impedance directly) is required to maximize energy transfer.

We thank the reviewer for this important clarification. We agree that, to maximize acoustic energy transmission, the compound lens should function as an impedance matching layer between the transducer and the biological soft tissues. It is indeed essential to consider both the acoustic impedance of the transducer and that of the tissue, rather than only matching the tissue directly.

In the current probe design, a $\lambda/4$ -thick matching layer was incorporated between the transducer and the lenses to optimize energy transmission. We have revised the original statement in the manuscript accordingly to reflect this more accurate and complete description as follows;

“To maximize energy transmission from the transducer to biological soft tissues, an impedance matching layer between the transducer and the lens must be considered. The acoustic impedance of the compound lens should be appropriately chosen to lie between that of the transducer and that of the tissue. Therefore, the density of the compound lens is a key parameter, as it directly influences its acoustic impedance”

8. For the kidney dataset, an isotropic voxel size of $\lambda/10$ (approximately 150 μm) was used; however, the minimum diameter reported in the skeletonized image is 70 μm . The authors are requested to explain this discrepancy.

In the software we used Amira 3D, the radius estimation is based on the Euclidean distance from each voxel on the vessel centerline to the closest boundary of the vessel. This measurement can be interpolated and refined by fitting a circle to the intensity gradients around the centerline. As a result, the vessel radius is calculated as the average radial distance from the centerline to the vessel boundary.

In our dataset, the 70 μm measurement corresponds to a single localized track within a vessel that was not fully reconstructed. However, due to interpolation and sub-voxel resolution estimation techniques, features smaller than the original voxel size can still be detected and estimated.

9. What concentration of microbubbles was used in the in vivo experiments?

In the in vivo experiments, the concentration of microbubbles used was as follows: a bolus of 1 mL of microbubble (MB) solution was injected intravenously through the porcine ear vein at the beginning of the acquisition, followed by intermittent bolus injections of 0.5 mL every minute for the remainder of the acquisition.

As recommended, we have revised the **Methods** section “**Experiments.**” to clearly describe the microbubble injection protocol as follows:

“An initial bolus of 1 mL of microbubble (MB) solution was injected through the intravenous system into the porcine ear. This was followed by additional intermittent bolus injections of 0.5 mL every minute for the rest of the acquisition.”

10. Since the study revolves around the transducer probe, further details about the transducer design and fabrication are warranted. What piezoelectric material was used? What were the fabrication steps? How were the two lenses attached to the piezoelectric material? What type of adhesive was employed?

We thank the reviewer for this comment.

The custom probe was designed using piezocomposite technology. The matrix of plano-convex lenses was fabricated from a single block of epoxy (thickness: 3.2 mm), with a speed of sound of 2570 m/s and an acoustic impedance of 3 MRayl.

This lens matrix was then bonded to the piezocomposite layer and matching layer using a structural adhesive. The second lens layer, composed of silicone, was molded directly onto the first assembly, forming a flat outer surface. The maximum thickness of this second lens layer was 0.375 mm at the dome of each plano-convex lens.

As recommended by reviewer, the detailed description of the custom probe has been added to the **Methods** section under sub-section “**Multi-lens array**”. In addition, schematic details of the probe have been included in Extended Fig.1.a.

“The total aperture size of the probe was $104 \times 82 \text{ mm}^2$. The probe was composed of three acoustic principal layers. The first layer consists of 252 large transducer matrix piezocomposite elements (14×18). The size of an individual element was set to $4.5 \times 4.5 \text{ mm}^2$ representing 3 times the acoustic wavelength ($3\lambda^2 \approx 4.5 \text{ mm}^2$, wavelength; $\lambda \approx 1.5 \text{ mm}$). The pitch between elements was set to 5.48 mm (corresponding 3.65λ) in both directions, this choice represents a compromise design to avoid large lens overlap while limiting grating lobe artefacts and preserving pressure field uniformity. The transducer layer was aligned with a second layer composed of diverging acoustic lenses (plano-convex lenses), also arranged in 14×18 configuration. The two layers were assembled with structural adhesive and an acoustic matching layer. A third flat lens layer (silicon plano-concave lens) was molded above the diverging lens layer, completing the assembly of a compound diverging lens. The compound diverging lens was made of an epoxy plano-convex lens of a maximum thickness of 3.2 mm, a speed of sound c_1 of 2570 m/s and an acoustic impedance of 3 MRayl. A silicon plano-concave lens was set on top with a maximum thickness of 0.375 mm, a speed of sound c_2 of 1015 m/s and an acoustic impedance of 1.1 MRayl. Other layers as backing and mechanical stiffener were added behind the piezocomposite. Technical schematic details of the custom multi-lens probe are shown in Extended Fig.1”

Technical schematics of the custom 252-element multi-lens probe. a, Transducer specifications and dimensions: (a.i) Elevation cross section view of the probe showing the layers containing; 1 represents the composite of piezoelectric elements, 2 represents the convex lens matrix layer, 3 flat lens layer which is the silicon concave lens, 4 backing layer, 5 mechanical stiffener layer. (a.ii) Magnified view of a single compound lens shown the convex lens with a thickness of 3.2 mm and the concave lens on top with a maximum thickness of 0.375 mm. (a.iii) View of the probe's aperture plane, illustrating the total aperture size (104 mm × 82 mm) and the number of transducer elements (252). (a.iv) Magnified view of the probe's aperture plane, showing the geometry of a single transducer elements (side length: 4.5 mm), pitch (5.48 mm), and lens base diameter (6.37 mm). ((a.iv) Magnified view of the probe's aperture plane, showing the geometry of a single transducer elements (side length: 4.5 mm), pitch (5.48 mm), and lens base diameter (6.37 mm).

11. How were the element size and pitch optimized? A discussion on how size and spacing affect grating lobes and sensitivity would help clarify the design choices and provide more background for the observed grating lobes.

The element size was optimized based on a prior study from our group (H. Favre et al., *Physics in Medicine & Biology*, 2022), which evaluated the angular aperture of the pressure field and acoustic gain as functions of element size. From this analysis, we selected an element size of 3λ . This choice also represented a good compromise between maintaining a compact total aperture and ensuring a sufficient number of elements to remain compatible with a standard 256-channel ultrasound system.

Regarding the pitch: each element was designed as a square, and the lens surface was sized to match the diagonal of the element, approximately 4.24λ . If a pitch equal to 3λ were used (i.e., with no spacing between elements), the lenses would overlap, degrading the lens performance and the pressure field uniformity. Conversely, increasing the pitch to 4.24λ or more would avoid overlap but introduce higher grating lobes. Therefore, a pitch of 3.65λ (corresponding to 5.48 mm) was selected as a compromise that avoids large lens overlap while limiting grating lobe artefacts and preserving pressure field uniformity.

For future designs, one strategy to reduce lens overlap and grating lobes could involve using flat circular elements that match the lens surface, and randomizing the positions of the elements aligned with lenses to have lower and suppress regular grating lobe patterns.

We added a brief clarification in the Methods section regarding how the pitch was optimized, as;

“The pitch between elements was set to 5.48 mm (corresponding 3.65λ) in both directions, this choice represents a compromise design to avoid large lens overlap while limiting grating lobe artefacts and preserving pressure field uniformity.”

12. Can authors also please comment on element to element crosstalk?

The probe design features a relatively large pitch, with an inter-element spacing of 0.66λ , which further minimizes the likelihood of element-to-element crosstalk. Additionally, in our setup, diverging waves were transmitted sequentially using only a single source positioned on one element at a time, which significantly minimizes the possibility of element-to-element crosstalk during transmission. Based on these factors, element-to-element crosstalk was not observed to be a significant issue in our configuration.

13. The temporal resolution claim of 312 Hz may be an overstatement. While the authors indeed collect volumes at 312 Hz, these volumes ultimately contribute to an accumulated ULM dataset requiring minutes of data acquisition. Moreover, even for a power Doppler sequence (not necessarily ULM as demonstrated in the paper), the volumes would typically be squared and averaged to form one image, effectively reducing the temporal resolution. In light of this, the authors should correct the following statement in the abstract and elsewhere in the manuscript “Combined to 3D ULM, it can map and quantify large vascular volumes (up to $120 \times 100 \times 82 \text{ mm}^3$) at high spatial resolution ($75 \mu\text{m}$) and high temporal resolution (312Hz) in swine organs in vivo corresponding to massive hundreds of billions of voxels per second with a low-cost technology.”

We thank the reviewer for this important observation regarding the temporal resolution. We agree that while volumes are indeed acquired at a frame rate of 312 Hz, the effective temporal resolution of the final ULM image, after microbubble localization and accumulation, is governed by the duration required to collect a sufficient number of microbubble events to generate a statistically robust map. This typically involves several seconds to minutes of acquisition, depending on the vascular flow dynamics and microbubble concentration.

The claim of "high temporal resolution (312 Hz)" in the abstract and elsewhere in the manuscript was therefore imprecise and could be misleading. We have revised the text to clarify that the system acquires volumetric data at a high frame rate of 312 Hz, which enables the capture of fast microbubble dynamics and offers robust reconstruction of vascular dynamics, such as power Doppler or short-time ULM sequences. However, we now explicitly state that the effective temporal resolution of the final ULM maps is lower, reflecting the integration time needed to achieve high spatial resolution and sufficient microbubble density. This clarification more accurately reflects the trade-off between spatial and temporal resolution inherent to ULM.

"Applied to map organs in a large animal swine model, we demonstrate ULM imaging over massive volumes (up to $120 \times 100 \times 82 \text{ mm}^3$), including ex-vivo heart, in-vivo kidney and liver with an effective temporal resolution depending on the accumulation time required for ULM reconstruction."

14. For the 3D ultrafast ultrasound sequence, was a 4×4 element matrix used for sequential transmits? How does this particular scheme compare with other 2D array transmission schemes previously employed (e.g., PMID: 35460988)

In the referenced study (e.g., PMID: 35460988; (Demeulenaere, O. et al.2022, 10.1016/j.ebiom.2022.103995)), the 3D ultrafast imaging sequence consisted of 16 plane wave transmissions, where all elements of the probe were activated simultaneously to emit a plane wave. In contrast, our study employed a different transmission strategy: we used a set of 16 centrally located elements (arranged in a 4×4 matrix) that were fired sequentially. In each sub-aperture transmission, only a single element was activated to emit a diverging wave, due to the presence of a diverging acoustic lens. Other transmission schemes may have been used such as plane or diverging wave transmits. However, large elements may limit the ability of the probe to produce waves with high angular wavevector.

To better illustrate our transmission scheme, we have added a schematic diagram in Extended Fig.2.b and attached below;

b

16 TX apodization for ultrafast ultrasound acquisition

b. Schematic illustration of the ultrafast acquisition scheme using the multilens probe, showing 16 transmit apodizations (TX), where each source is activated individually on a single element during each transmission. During reception, all elements are activated after every transmit event.

We have clarified it in the revised manuscript in section “**Ultrasound sequences.**” as follows:

“Transmit schemes were programmed to perform 3D ultrafast ultrasound imaging in order to preserve a sufficient temporal resolution. The selected configuration consisted of 16 sequential transmissions, each using a single element of the probe to emit a diverging wave. The spatial arrangement of these 16 sources is positioned in the center of the probe, and illustrated in the Extended Fig.2.b. Backscattered echoes were recorded with all element after each transmission.”

Minor comments

15. In Fig. 4l, the x-axis is labeled as "Temps" instead of "Time."

We thank the reviewer for pointing this out. The x-axis label in Fig. 4.l has been corrected from "Temps" to "Time" in the revised manuscript and updated to be Fig.4.n.

16. Fig. 1a could benefit from improved contrast for better visualization.

We thank the reviewer for the helpful suggestion. The contrast in Fig. 1a has been enhanced in the revised manuscript to improve visualization.

17. Authors are requested to provide the cost of probe and imaging system to substantiate the low cost technology claims.

The total estimated cost of the custom-designed probe is approximately €50,000. The imaging system used in this study, the Verasonics Vantage 256 research platform, costs around €150,000. While not inexpensive, this setup remains significantly more affordable than systems required for fully populated matrix arrays. For comparison, a system supporting a small-aperture matrix probe with 1024 independent channels typically exceeds €1 million. Scaling up to systems capable of handling more than 10,000 channels, required for large-aperture, high-density 3D imaging, can push total system costs above €10 million. This highlights the cost-effectiveness of our approach, which enables large-scale 3D imaging with a fraction of the hardware complexity and financial investment.

We have added in the Discussion section as follow.

“Cost of the system (less than €200k) was considered low in comparison to investment usually required for fully populated matrix arrays (~ €1M)”.

Reviewer #1 (Remarks to the Author):

The authors have addressed all of my previous comments. I only have a few minor revisions listed below. I recommend the paper for acceptance once these are addressed.

We sincerely thank the reviewer for their positive feedback and for recommending our manuscript for acceptance. We have addressed the few minor revisions listed below.

Minor points:

1- Line 477: "16 sources positioned on elements at the probe center were selected..."

Authors need to specify the position of the virtual source.

As recommended by the reviewer the position of the virtual source was specified as "For each transmission, the source was located at the physical position of the active element on the transducer surface"

2- For the Extended Fig. 5], authors need to specify if a) is the Power Doppler mode with or without MBs injection. Is it Contrast enhanced PD? Please clarify this in the text as well

Power Doppler imaging was performed using MBs injection. As recommended, it was clarified in the caption of figure 5 as "contrast-enhanced Power Doppler imaging from a porcine kidney"

Reviewer #2 (Remarks to the Author):

In this second version, the authors have provided extended information about motion compensation, the transmission scheme used, and additional details about the probe and system. Based on the feedback from the other reviewers, it appears we all had similar comments regarding these sections. I am pleased that the authors have included sufficient material on their motion compensation approach to clarify its limitations and justify their choices. The paper has been significantly improved. I still have a few minor questions and comments regarding motion compensation, but overall, I am satisfied with the current version of the manuscript. The authors have addressed all my concerns, and I look forward to seeing this probe—and similar ones based on the same principles—become available to other research groups.

We thank the reviewer for their positive feedback and are glad that the extended information on motion compensation, the transmission scheme, and the probe system details have addressed their concerns. We have carefully addressed the remaining minor questions in the section below.

Motion Compensation:

- When selecting the three shorter segments to compute intra-block power Doppler volumes, why did you choose three? How did you determine the number of frames used to generate the sub-blocks? An explanation should be provided. In your reviewer response, you wrote: "Specifically, each acquisition block was divided into three sub-blocks based on time," but no further justification or rationale was given.

The choice of dividing each block into three temporal sub-blocks was arbitrary. Further optimization could be conducted to determine the optimal number of sub-blocks required depending on motion characteristics. It was added in the text as "Each block was arbitrary divided into three temporal sub-blocks."

- At the final stage of your motion correction process, when applying the transformation matrices, do you retain all datasets? Do you perform a secondary check to assess whether motion correction was sufficient for all tracks?

At the final stage, all datasets were retained and no secondary check was performed.

- Extended Figure 4: Panels (b), (c), and (d) should match the region shown in Figure 5(a). Using the same region would allow for a fairer comparison. Additionally, in Extended Figure 4(d), please show the line profile for all motion compensation methods, including the uncorrected version. This will help evaluate whether the two-step method provides a significant improvement over the one-step method.

As recommended by the reviewer, we have updated in the manuscript the Extended Figure 4 panels (b), (c), and (d) to match the region shown in Figure 5(a). In Extended Figure 4(d), we now include the line profiles for all motion compensation methods; uncorrected, inter-block correction, and intra + inter-block correction.

b, Density map of the kidney cortex (blue box; $\sim 28 \times 45 \times 33 \text{ mm}^3$) before motion correction. **c**, Density map of the kidney cortex (blue box; $\sim 28 \times 45 \times 33 \text{ mm}^3$) after intra-block motion correction. **d**, Comparison of Cross-sectional density profile of vessel (i) in both panel b and c, showing the normalized amplitude as a function of distance before (blue curve), inter-block (red curve), and intra-block (yellow curve) motion correction.

Quasi Real-Time:

- “Subsequently, our prototype allowed for quasi real-time imaging using Doppler mode after injecting contrast agents, which enabled to adjust fine probe placement.”

In several places—both in your response and the revised manuscript—you refer to "quasi real-time." Could you please define what you mean by "quasi real-time"? (e.g., frame rate, latency, image quality?)

Quasi real-time enabled positioning. The system can generate an enhanced-contrast Doppler volume every 3 seconds at a grid reconstruction resolution of $\lambda \approx 1.5 \text{ mm}$. It was added in the text discussion as: “Quasi real-time contrast-enhanced Doppler imaging (0.3 vol/sec at a grid reconstruction resolution of $\lambda \approx 1.5 \text{ mm}$) with the multi-lens probe was employed as ...”

- In the final text, you do not mention that ultrasound contrast agents are necessary for quasi real-time Doppler imaging. For example, you wrote:

“Quasi-real-time Doppler imaging with the multi-lens probe was employed as a guidance tool to facilitate precise probe positioning and to confirm effective acoustic coupling throughout the acquisition process.” “Subsequently, quasi-real-time Doppler imaging with the multi-lens probe

was employed as a guidance tool to facilitate precise probe positioning and to confirm effective acoustic coupling throughout the acquisition process.”

This could be misleading, as Doppler imaging can be performed without contrast agents. Please clarify in the final manuscript whether contrast agents are required to achieve quasi real-time Doppler imaging with your setup.

Power Doppler imaging was performed using MBs injection. As recommended, it was clarified in the text as: “Quasi real-time contrast-enhanced Doppler imaging (0.3 vol/sec at a grid reconstruction resolution of $\lambda \approx 1.5$ mm) with the multi-lens probe was employed as a guidance tool...”

- Furthermore, if your ultimate goal is to use this approach in human subjects, injecting microbubbles to aid with positioning could limit the total allowable dosage for localization. Can you perform contrast-free Doppler imaging for guidance instead? Have you investigated contrast-free transmission strategies for this probe? I know you have mentioned it in your revised manuscript but please provide further information as it is a limitation of your probe if it is mandatory to inject microbubbles for Doppler imaging.

The large size of the probe generally allows for adequate positioning in most cases. For fine adjustments, a small contrast agent injection (0.2 mL) can be administered, which remains in the body for several minutes. We did not consider this a limitation, as it leaves up to 4.8 mL available for clinical use. Consequently, contrast-free Doppler imaging for positioning was not investigated.

Ex Vivo and In Vivo Experiments:

- Can you provide the total amount of MBs injected during your experiments? In your updated text, you state:

“An initial bolus of 1 mL of microbubble solution was injected through the intravenous system into the porcine ear. This was followed by additional intermittent bolus injections of 0.5 mL every minute for the rest of the acquisition.”

Total amount of MBs injected varied between 5mL to 15mL depending on the acquisition.

Can you discuss whether this injection protocol would be feasible for translation to human subjects?

In this study, the objective was not to develop a clinical injection protocol, but rather to demonstrate the capability of the multi-lens probe to image the vasculature of entire organs. Future work will focus on optimizing a protocol, whether based on bolus administration or continuous infusion, while respecting the maximum allowable dose. It was added in the discussion as: “Regarding the microbubble injection protocol employed in this study, it was not optimized for clinical practice. Future work will focus on improving and adapting this protocol for clinical applications.”

Other Comments:

- For all newly provided MIP Power Doppler images, please ensure the views match those of your SR images. For example, the view for the heart appears too short, and the liver image is shown from a different angle.

As recommended by the reviewer, all newly provided MIP Power Doppler images have been updated to match the corresponding SR image views, including corrected perspectives for the heart and liver datasets.

- Extended Figure 3: The image of the setup is incorrectly positioned—it overlaps with panel (b) rather than panel (c).

As recommended by the reviewer, the image of the setup was repositioned in Extended Figure 3 (c).

- Extended Figure 3(e): When showing the cross-section of the twisted tube, please overlay the Doppler imaging. Use a fair and consistent dynamic range when displaying the Doppler image.

As recommended by the reviewer, Doppler imaging has been added to Extended Figure 3(e) to overlay the cross-section of the ULM image of the twisted tube.

Cross-section of the twisted tube (white box in **c**) from the ULM image, overlaid with enhanced-contrast Doppler imaging (white box in **b**), highlighting the separation between the intertwined tube segments as resolved by ULM.

Reviewer#3

The authors have done a good job responding to the questions; however, the transducer probe is the centerpiece of this manuscript, and its fabrication details should be described in full to ensure reproducibility. Specifically, please provide:

We thank the reviewer for their positive feedback and for their interest in the fabrication details of the transducer probe. The key parameters required to understand and reproduce the concept, namely, the multi-lens design, element and lens size, pitch, lenses speed of sound and density have been fully provided in the manuscript. These represent the aspects most relevant to the scope of our study and to the majority of readers.

The detailed manufacturing process is proprietary to our industrial partner, Vermon, and is not accessible to us due to the seller–customer nature of our collaboration. This is standard in academic–industry partnerships; however, we remain happy to clarify any aspects within our knowledge and to provide all specifications necessary for others to reproduce the concept.

1. What piezocomposite was used?

The specific piezocomposite used in the probe was not disclosed to us. However, the probe was fabricated according to the specifications detailed in the manuscript. Please feel free to contact Vermon directly for further details.

2. What was the configuration, material, and radii of curvature of the plano-convex and plano-concave lenses? What epoxy material was used to fabricate them.

The layer composed of plano-convex lenses was set in a 14 by 18 arrangement. Each plano-convex lens had a maximum thickness of 3.2 mm, corresponding to a radius of curvature of 3.2 mm. The plano-convex lenses were made of epoxy with a speed of sound of 2570 m/s and an acoustic impedance of 3 MRayl (the specific epoxy formulation is not available to us). Silicone plano-concave lens had a radius of curvature of 3.2 mm. It was added in the method section. The curative radii were added in the text as “corresponding to a radius of curvature of 3.2 mm”

3. Which structural adhesive was employed? How will the adhesive properties affect the array performance?

Structural adhesive falls outside the scope of our expertise, so we did not provide specific requirements to the supplier. Additionally, Vermon considers this information proprietary and does not disclose it. Please feel free to contact them directly for further details.

4. Which backing and mechanical stiffener were used?

Backing and mechanical stiffener falls outside the scope of our expertise, so we did not provide specific requirements to the supplier. Additionally, Vermon considers this information proprietary and does not disclose it. Please feel free to contact them directly for further details.

5. What material was used for the matching layers? Was the biocompatibility of the material taken into account?

Biocompatibility was among the key specifications we requested from the supplier. However, we do not have access to the specific material composition, as Vermon treats this information as proprietary. Please feel free to contact Vermon directly for further details.

6. What rationale guided the selection of these layers? What calculation were used to optimize impedance matching?

Rationale behind the selection of these layers falls outside the scope of our expertise, so we did not provide specific requirements to the supplier. Additionally, Vermon considers this information proprietary and does not disclose it. Please feel free to contact them directly for further details.

Without these precise technical specifications—including materials, dimensions, fabrication protocols, and design rationale—other researchers cannot validate or replicate the results. Providing this information will also benefit the field by enabling further development and attracting broader interest.

We respectfully disagree. The manuscript provides all necessary specifications for researchers to define precise probe requirements. It was not intended to enable independent fabrication, as the proprietary manufacturing knowledge lies with the industrial supplier. Researchers can use the provided details to work with manufacturers experienced in producing such probes.

Comments

The authors propose a multi-lens ultrasound array for scaling 3D ultrasound localization microscopy (ULM) to large organs in vivo. Their proposed technology addresses several limitations of conventionally used ultrasound probes for ULM, which are otherwise difficult to scale to large organs, are expensive, have a limited field of view, and often require extensive computational resources. Below are some comments that may help improve the manuscript further:

1. The authors are requested to include a discussion on row-column addressable arrays, which have been proposed as alternatives to matrix probes for 3D ULM.
2. The authors claim a resolution of 75 μm based on the statistical difference between neighboring voxels in the velocity profile. What was the inlet flow rate during this measurement? Does the resolution change at different flow rates?
3. What was the resolution measurement from the phantom experiment using the Fourier Shell Correlation (FSC) method, which was otherwise used to estimate spatial resolution for the in vitro and in vivo datasets? The reported spatial resolution outside phantom study is more than 75 μm which runs counter to the claims made in the paper.
4. Do the authors observe a Poiseuille flow profile in the kidney and liver datasets?
5. How are arterial and venous vessels differentiated in the kidney and liver datasets?
6. In Fig. 4gii, the velocity appears to change direction from +50 to -50 mm/s. The authors are requested to explain this observation further.
7. The statement: *"To maximize energy transmission from the transducer to the biological soft tissues, the acoustic impedance of the compound lens must be as close as possible as the acoustic impedance of the biological soft tissues. Thus, the density of the compound lens must be taken into account."* needs clarification. It should be noted that optimal impedance matching layer taking into account both transducer impedance and tissue impedance (not necessarily just matching the tissue impedance directly) is required to maximize energy transfer.
8. For the kidney dataset, an isotropic voxel size of $\lambda/10$ (approximately 150 μm) was used; however, the minimum diameter reported in the skeletonized image is 70 μm . The authors are requested to explain this discrepancy.
9. What concentration of microbubbles was used in the in vivo experiments?
10. Since the study revolves around the transducer probe, further details about the transducer design and fabrication are warranted. What piezoelectric material was

used? What were the fabrication steps? How were the two lenses attached to the piezoelectric material? What type of adhesive was employed?

11. How were the element size and pitch optimized? A discussion on how size and spacing affect grating lobes and sensitivity would help clarify the design choices and provide more background for the observed grating lobes.
12. Can authors also please comment on element to element crosstalk?
13. The temporal resolution claim of 312 Hz may be an overstatement. While the authors indeed collect volumes at 312 Hz, these volumes ultimately contribute to an accumulated ULM dataset requiring minutes of data acquisition. Moreover, even for a power Doppler sequence (not necessarily ULM as demonstrated in the paper), the volumes would typically be squared and averaged to form one image, effectively reducing the temporal resolution. In light of this, the authors should correct the following statement in the abstract and elsewhere in the manuscript “Combined to 3D ULM, it can map and quantify large vascular volumes (up to $120 \times 100 \times 82 \text{ mm}^3$) at high spatial resolution ($75 \mu\text{m}$) and high temporal resolution (312Hz) in swine organs in vivo corresponding to massive hundreds of billions of voxels per second with a low-cost technology.”
14. For the 3D ultrafast ultrasound sequence, was a 4×4 element matrix used for sequential transmits? How does this particular scheme compare with other 2D array transmission schemes previously employed (e.g., PMID: 35460988)

Minor comments

15. In Fig. 4l, the x-axis is labeled as "Temps" instead of "Time."
16. Fig. 1a could benefit from improved contrast for better visualization.
17. Authors are requested to provide the cost of probe and imaging system to substantiate the low cost technology claims.

The authors have done a good job responding to the questions; however, the transducer probe is the centerpiece of this manuscript, and its fabrication details should be described in full to ensure reproducibility. Specifically, please provide:

1. What piezocomposite was used?
2. What was the configuration, material, and radii of curvature of the plano-convex and plano-concave lenses? What epoxy material was used to fabricate them.
3. Which structural adhesive was employed? How will the adhesive properties affect the array performance?
4. Which backing and mechanical stiffener were used?
5. What material was used for the matching layers? Was the biocompatibility of the material taken into account?
6. What rationale guided the selection of these layers? What calculation were used to optimize impedance matching?

Without these precise technical specifications—including materials, dimensions, fabrication protocols, and design rationale—other researchers cannot validate or replicate the results. Providing this information will also benefit the field by enabling further development and attracting broader interest.